# On the Existence of Consistent Adversarial Attacks in High-Dimensional Linear Classification

**Matteo Vilucchio** [1]  **Lenka Zdeborová** [2]  **Bruno Loureiro** [3]

## Abstract

What fundamentally distinguishes an adversarial attack from a misclassification due to limited model expressivity or finite data? In this work, we investigate this question in the setting of high-dimensional binary classification, where statistical effects due to limited data availability play a central role. We introduce a new error metric that precisely captures this distinction, quantifying model vulnerability to consistent adversarial attacks — perturbations that preserve the ground-truth labels. Our main technical contribution is an exact and rigorous asymptotic characterization of these metrics in both well-specified models and latent space models, revealing different vulnerability patterns compared to standard robust error measures. The theoretical results demonstrate that as models become more overparameterized, their vulnerability to label-preserving perturbations grows, offering theoretical insight into the mechanisms underlying model sensitivity to adversarial attacks.

## 1. Introduction

Machine learning models, despite their remarkable performance across various domains, remain vulnerable to adversarial examples — inputs specifically crafted to mislead models while appearing innocuous to humans. While adversarial robustness has attracted significant research attention, a critical distinction often overlooked is between *consistent* (or *proper*) and *inconsistent* (or *improper*) adversarial examples. Consistent adversarial examples maintain the

[1]Information, Learning & Physics Laboratory, École Polytechnique Fédérale de Lausanne (EPFL), Lausanne, Switzerland [2]Statistical Physics of Computation Laboratory, École Polytechnique Fédérale de Lausanne (EPFL), Lausanne, Switzerland [3]Département d'Informatique de l'École Normale Supérieure (DI ENS), CNRS, PSL University, Paris, France. Correspondence to: Matteo Vilucchio <matteo.vilucchio@epfl.ch>.

*Proceedings of the 43$^{rd}$ International Conference on Machine Learning*, Seoul, South Korea. PMLR 306, 2026. Copyright 2026 by the author(s).

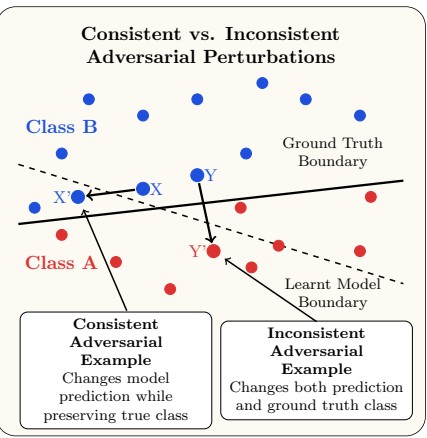

*Figure 1.* Illustration of the difference between consistent Adversarial Perturbation and inconsistent Adversarial Perturbation.

ground-truth label despite perturbations, whereas inconsistent ones change the true classification. Consider the classic panda example from Goodfellow et al. (2015): an image that, after subtle perturbations, is misclassified by a neural network but still depicts a panda to humans (consistent attack). If perturbations transformed the image to genuinely resemble a different animal, it would be inconsistent. This distinction is crucial: vulnerability to consistent attacks represents genuine model failure to capture invariant features.

The assumption that adversarial perturbations do not alter the true class (*i.e.* remain consistent) underlies most practical approaches to adversarial robustness in computer vision (Madry et al., 2018). While this assumption has been explored in theoretical works on robust generalization (Donhauser et al., 2021; Raghunathan et al., 2020), a mathematical understanding of their properties, such as existence and effectiveness in tricking even simple linear classifiers remains elusive.

Following a large body of work in high-dimensional statistics (Krogh & Hertz, 1991; Seung et al., 1992; Bean et al., 2013a; Thrampoulidis et al., 2018; Aubin et al., 2020; Vilucchio et al., 2024), we analyze this problem through the lens of exact asymptotics of linear classifiers. We develop novel metrics that precisely quantify vulnerability to both consistent and inconsistent adversarial attacks. We define and

analyze these metrics in two complementary settings: first, a *well-specified* model where all input covariates are directly available; second, a *latent space* model where the available covariates are feature transformations of underlying latent variables. For both settings, we derive closed-form expressions for the consistent robustness metrics in the high-dimensional limit—where the latent space dimension $d$, the number of features $p$, and the sample size $n$ all scale to infinity at fixed ratios. In the latent space model, we further derive exact asymptotic descriptions for the performance of robust empirical risk minimization (Goodfellow et al., 2015; Madry et al., 2018), the mostly adopted way of finding a robust model.

Furthermore, the effect of over or under-parameterization — using more or less parameters than strictly necessary to encode the data features — is still unclear in the adversarial settings. Ribeiro & Schön (2023) consider the regression case for squared loss but in the context of classification the question is still open. While overparameterization generally improves standard generalization (Goldt et al., 2022b; Hastie et al., 2022), its effects on adversarial robustness are less understood, particularly when considering consistent versus inconsistent attacks. Conventional wisdom suggests that overparameterized models might be more vulnerable to adversarial examples due to their flexibility in fitting noise. However, our analysis reveals a more nuanced picture: overparameterization can improve an attacker's ability to craft effective adversarial examples on correctly classified data points, but when considering consistent attacks on all possible inputs, increasing overparameterization leads to improvement due to its beneficial role on clean generalization. See Appendix A for a further discussion of related works.

Our **main contributions** can be summarized as

1. We establish necessary and sufficient conditions for the existence of consistent perturbations in two classes of binary classifiers: well-specified linear classifiers, and a latent variable model that accounts for misspecification and overparametrization in linear estimation, independently of the data distribution. Under a Gaussian design, this leads to an exact formula for the probability that consistent attacks exist in these models.

2. We introduce novel consistent robust error metrics quantifying the impact of consistent attacks. For the classifiers of interest, we derive an asymptotic formula that exactly characterize their high-dimensional limits under a Gaussian design assumption.

3. We study how robust empirical risk minimization can mitigate consistent attacks in this high-dimensional limit, for both the well-specified and latent variable model. For the latter, this requires a novel exact asymptotic characterization of the robust ERM estimator under misspecification, of independent interest.

Our work reveals that overparameterization plays a nuanced but crucial role in building resistance against consistent adversarial attacks. Contrary to conventional wisdom, our theoretical analysis demonstrates that higher degrees of parameterization can be beneficial for overall robustness, though this benefit must be balanced against increased vulnerability on specific subsets of inputs. These insights can provide guidance for system design, highlighting the importance of considering the consistent/inconsistent attack distinction when evaluating and optimizing model robustness.

**Notations** We denote vectors by bold letters $\boldsymbol{x} \in \mathbb{R}^d$. $\mathbb{S}^{d-1}(r) = \{\boldsymbol{x} \in \mathbb{R}^d : ||\boldsymbol{x}||_2 = r\}$ denotes the Euclidean sphere of radius $r$, and $\text{span}(\boldsymbol{x}) = \boldsymbol{x}\mathbb{R} = \{\mu\boldsymbol{x}, \mu \in \mathbb{R}\}$. For $q \geq 1$, $||\boldsymbol{x}||_q = (\sum_{j=1}^d x_j^q)^{1/q}$ denotes the $\ell^q$-norm, and $B_q(r) = \{\boldsymbol{x} \in \mathbb{R}^d : ||\boldsymbol{x}||_p \leq r\}$ the $\ell^q$-ball of radius $r > 0$. We denote by $q^\star$ the dual of $q$ in the $\ell^q$ sense: the number $q^\star$ such that $1/q + 1/q^\star = 1$. We denote by $\mathcal{N}(0,1)$ the standard normal distribution, and $\mathbb{P}[Z \leq t] = \Phi(t)$ its c.d.f. Additionally $\mathcal{P}_{f(\cdot)}$ represents the proximal operator of the function $f : \mathbb{R} \to \mathbb{R}$.

## 2. Consistent Adversarial Perturbations and How To Quantify Them

As motivated in Section 1, the key distinction between an adversarial attack and a random perturbation of the data is the underlying assumption that adversarial attacks leave the ground truth data distribution unchanged. We formalize this notion in the context of binary classifiers.

Consider a binary classification task $(\mu, f_\star)$ defined by a covariate distribution $\mu$ over $\mathbb{R}^d$ and a ground-truth classifier $f_\star : \mathbb{R}^d \to [0, 1]$, such that for a given $\boldsymbol{x} \sim \mu$, we can assign a binary label $y \in \{-1, +1\}$ with probability given by $f_\star(\boldsymbol{x}) = \mathbb{P}(y = +1|\boldsymbol{x})$.

**Definition 2.1** (Consistent attack). Let $f_\star, \hat{f} : \mathbb{R}^d \to [0, 1]$ denote two binary classifiers, refereed to as the *target* and the *model*, $\boldsymbol{x} \in \mathbb{R}^d$ a covariate and $\hat{y} : [0, 1] \to \{\pm 1\}$ a decision rule associated to $\hat{f}$. We say a perturbation $\boldsymbol{\delta} \in B_q(\varepsilon)$ of the model $\hat{f}$ is a *consistent* adversarial attack with respect to the target $f_\star$, the covariate $\boldsymbol{x} \in \mathbb{R}^d$ and the decision rule $\hat{y}$ if the following conditions hold:

- **Model deception:** $\hat{y}(\hat{f}(\boldsymbol{x})) \neq \hat{y}(\hat{f}(\boldsymbol{x} + \boldsymbol{\delta}))$.
- **Target invariance:** $f_\star(\boldsymbol{x}) = f_\star(\boldsymbol{x} + \boldsymbol{\delta})$.

If target invariance fails the attack is *inconsistent*.

See Figure 1 for an illustration of a consistent vs. inconsistent attack in the case of linear classifiers.

*Remark* 2.2. For deterministic classifiers, target invariance implies label invariance (but not conversely). For probabilistic classifiers, target invariance does not ensure label invariance for specific realizations, as labels may differ due to randomness. However, target invariance is a natural

condition in a statistical framework: it requires adversarial perturbations to preserve the conditional label distribution $\mathbb{P}(y|\boldsymbol{x})$, not individual label samples. Moreover, since the target-invariant constraint set $\{\boldsymbol{\delta} : f_\star(\boldsymbol{x}) = f_\star(\boldsymbol{x}+\boldsymbol{\delta})\}$ is always contained in the label-preserving set $\{\boldsymbol{\delta} : \hat{y}(f_\star(\boldsymbol{x})) = \hat{y}(f_\star(\boldsymbol{x}+\boldsymbol{\delta}))\}$, our consistent metrics provide a *lower bound* on the vulnerability to label-preserving attacks, while the standard robust error bounds it from above.

We introduce metrics to quantify these properties.

**Definition 2.3** (Adversarial errors)**.** Let $(\mu, f_\star)$ denote a binary classification task. Given a classifier $\hat{f} : \mathbb{R}^d \rightarrow [0, 1]$ and its associated predictor $\hat{y}(\boldsymbol{x}) = \hat{y}(\hat{f}(\boldsymbol{x}))$, we define the following three metrics

- **Robust error**: This is the standard notion of robust generalization error in the adversarial literature (Clarysse et al., 2022; Dohmatob, 2024; Dohmatob & Scetbon, 2023), and simply quantifies how vulnerable $\hat{f}$ is to arbitrary perturbations in a $\ell^q-$ball:

$$E_{\text{rob}} = \mathbb{E}_{(\boldsymbol{x},y)} \left[ \max_{\boldsymbol{\delta} \in B_q(\varepsilon)} \mathbb{1}\{y \neq \hat{y}(\boldsymbol{x} + \boldsymbol{\delta})\} \right], \quad (1)$$

- **Consistent robust error:** The standard robust error considers both consistent and inconsistent perturbations. To quantify just the role of consistent attacks, we define the *consistent robust error* by excluding inconsistent perturbations:

$$E_{\text{rob}}^{\text{cns}} = \mathbb{E}_{(\boldsymbol{x},y)} \left[ \max_{\substack{\boldsymbol{\delta} \in B_q(\varepsilon): \\ f_\star(\boldsymbol{x})=f_\star(\boldsymbol{x}+\boldsymbol{\delta})}} \mathbb{1}\{y \neq \hat{y}(\boldsymbol{x} + \boldsymbol{\delta})\} \right], \quad (2)$$

  Note that the critical difference between $E_{\text{rob}}$ and $E_{\text{rob}}^{\text{cns}}$ lies in the constraint in the inner maximization that satisfies the target invariance from Definition 2.1 (c.f. Remark 2.2).

- **Consistent boundary error:** Finally, note that the consistent robust error does not distinguish between labels that are originally misclassified by the model and labels that become misclassified under the attack perturbation. This motivates the introduction of a more nuanced metric, accounting only for labels that are misclassified due to the attack:

$$E_{\text{bnd}}^{\text{cns}} = \mathbb{E}_{(\boldsymbol{x},y)} \left[ \max_{\substack{\boldsymbol{\delta} \in B_q(\varepsilon): \\ f_\star(\boldsymbol{x})=f_\star(\boldsymbol{x}+\boldsymbol{\delta})}} \mathbb{1}\left\{ \begin{array}{c} y \neq \hat{y}(\boldsymbol{x} + \boldsymbol{\delta}) \\ y = \hat{y}(\boldsymbol{x}) \end{array} \right\} \right]. \quad (3)$$

*Remark* 2.4. Note that for any $(\boldsymbol{x}, y) \in \mathbb{R}^d \times \{-1, +1\}$, the constraint sets:

$$\begin{aligned} C_{\text{rob}} &= \{\boldsymbol{\delta} \in B_q(\varepsilon) : \hat{y}(\boldsymbol{x}) \neq \hat{y}(\boldsymbol{x} + \boldsymbol{\delta})\}, \\ C_{\text{rob}}^{\text{cns}} &= \{\boldsymbol{\delta} \in B_q(\varepsilon) : \hat{y}(\boldsymbol{x}) \neq \hat{y}(\boldsymbol{x} + \boldsymbol{\delta}) \\ &\qquad\qquad \text{and } f_\star(\boldsymbol{x}) = f_\star(\boldsymbol{x} + \boldsymbol{\delta})\}, \\ C_{\text{bnd}}^{\text{cns}} &= C_{\text{rob}}^{\text{cns}} \cap \{y = \hat{y}(\boldsymbol{x})\}, \end{aligned} \quad (4)$$

are nested $C_{\text{bnd}}^{\text{cns}} \subset C_{\text{rob}}^{\text{cns}} \subset C_{\text{rob}}$. Therefore, we generally have $0 \leq E_{\text{bnd}}^{\text{cns}} \leq E_{\text{rob}}^{\text{cns}} \leq E_{\text{rob}}$.

## 3. Consistent Attacks In Well-Specified Linear Classification

Despite an established literature studying robust training schemes, the fundamental properties of consistent attacks remain poorly understood. Our goal in the following is to fill this gap by studying their behavior in the context of high-dimensional binary linear classifiers.

**Definition 3.1** (Linear classifiers)**.** A linear binary classifier in $\mathbb{R}^d$ is a function

$$f_{\boldsymbol{w}}(\boldsymbol{x}) = \mathbb{P}_{\boldsymbol{w}}(y = +1|\boldsymbol{x}) = \varphi(\langle \boldsymbol{w}, \boldsymbol{x} \rangle), \quad (5)$$

defined by the *weight vector* $\boldsymbol{w} \in \mathbb{R}^d$ and a monotonic *link function* $\varphi : \mathbb{R} \rightarrow [0, 1]$.

The class of linear binary classifiers encompass several models of interest in statistics, such as the logit $\varphi(t) = (1 + e^{-t})^{-1}$, the probit $\varphi(t) = 1/2(\text{erf}(t) + 1)$ and the noiseless $\varphi(t) = \mathbb{1}\{t \geq 0\}$ model.

### 3.1. Geometry of Consistent Attacks

As a first step, we consider the geometry of consistent attacks in the class of linear classifiers. Let $f_{\boldsymbol{w}_\star}$ denote a reference linear classifier with weights $\boldsymbol{w}_\star \in \mathbb{S}^{d-1}(\sqrt{d})$ and link function $\varphi_\star$, which we refer to as the *ground-truth*. Since $\varphi_\star : \mathbb{R} \rightarrow [0, 1]$ is monotonic, the target invariance condition $f_{\boldsymbol{w}_\star}(\boldsymbol{x}) = f_{\boldsymbol{w}_\star}(\boldsymbol{x} + \boldsymbol{\delta})$ is equivalent to $\langle \boldsymbol{\delta}, \boldsymbol{w}_\star \rangle = 0$, i.e. the attack must be orthogonal to $\boldsymbol{w}_\star$. Therefore, the set of admissible consistent adversarial attacks with respect to the target classifier defines a hyperplane:

$$H_q(\varepsilon) := \{\boldsymbol{\delta} \in B_q(\varepsilon) : \langle \boldsymbol{w}_\star, \boldsymbol{\delta} \rangle = 0\}. \quad (6)$$

Consider a second linear classifier $f_{\hat{\boldsymbol{w}}}$ with weights $\hat{\boldsymbol{w}} \in \mathbb{R}^d$ and link function $\varphi$, which we refer to as the *model*. A successful attack should flip the predictor labels $\hat{y}(\boldsymbol{x}) \neq \hat{y}(\boldsymbol{x}+\boldsymbol{\delta})$. For the standard decision function $\hat{y}(\boldsymbol{x}) = \text{sign}(2f_{\hat{\boldsymbol{w}}}(\boldsymbol{x}) - 1)$ this is equivalent to $\langle \hat{\boldsymbol{w}}, \boldsymbol{x} \rangle(\langle \hat{\boldsymbol{w}}, \boldsymbol{x} \rangle + \langle \hat{\boldsymbol{w}}, \boldsymbol{\delta} \rangle) \leq 0$. This is the case if and only if:

$$|\langle \hat{\boldsymbol{w}}, \boldsymbol{\delta} \rangle| > |\langle \hat{\boldsymbol{w}}, \boldsymbol{x} \rangle|, \ \text{sign}(\langle \hat{\boldsymbol{w}}, \boldsymbol{\delta} \rangle) = -\text{sign}(\langle \hat{\boldsymbol{w}}, \boldsymbol{x} \rangle). \quad (7)$$

In other words, to flip the model prediction, an attacker must have an anti-parallel component to the predictor weights and exceed the prediction margin $|\langle \hat{\boldsymbol{w}}, \boldsymbol{x} \rangle|$. Combining these observations, we can derive the following geometrical characterization for the existence of consistent perturbations.

**Proposition 3.2** (Existence of consistent attack)**.** *Consider two linear classifiers defined by the weights $\boldsymbol{w}_\star \in \mathbb{S}^{d-1}(\sqrt{d})$ and $\hat{\boldsymbol{w}} \in \mathbb{R}^d$. Let $\boldsymbol{x} \in \mathbb{R}^d$ denote a covariate, and assume $\langle \hat{\boldsymbol{w}}, \boldsymbol{x} \rangle \neq 0$. Then, a consistent attack $\boldsymbol{\delta} \in B_q(\varepsilon)$ with respect to $\boldsymbol{w}_\star, \boldsymbol{x} \in \mathbb{R}^d$ and the decision function $\hat{y}(\boldsymbol{x}) = \text{sign}(2f_{\hat{\boldsymbol{w}}}(\boldsymbol{x}) - 1)$ exists if and only if:*

$$\varepsilon \, d_{q^\star}^\star(\hat{\boldsymbol{w}}_\perp) \geq |\langle \hat{\boldsymbol{w}}, \boldsymbol{x} \rangle|, \quad (8)$$

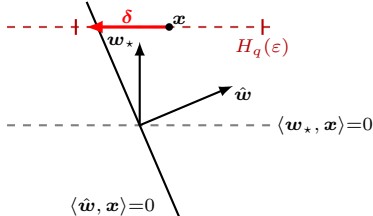

*Figure 2.* Geometry of consistent attacks (Section 3.1). Target invariance forces the perturbation to lie in the hyperplane $H_q(\varepsilon) = \{\boldsymbol{\delta} \in B_q(\varepsilon) : \langle \boldsymbol{w}_\star, \boldsymbol{\delta} \rangle = 0\}$ orthogonal to the target weights $\boldsymbol{w}_\star$ (here a segment, delimited by the two red ticks), so $\boldsymbol{\delta}$ stays parallel to the target boundary (upper dashed line). A consistent attack succeeds when it pushes $\boldsymbol{x}$ across the model boundary $\langle \hat{\boldsymbol{w}}, \boldsymbol{x} \rangle = 0$ without leaving $H_q(\varepsilon)$.

*where $\hat{\boldsymbol{w}}_\perp = \hat{\boldsymbol{w}} - 1/d\langle \boldsymbol{w}_\star, \hat{\boldsymbol{w}} \rangle \boldsymbol{w}_\star$ is the predictor components orthogonal to the target weights, $d_{q^\star}^\star(\hat{\boldsymbol{w}}_\perp) = \inf_{\mu \in \mathbb{R}} ||\hat{\boldsymbol{w}}_\perp - \mu \boldsymbol{w}_\star||_{q^\star}$ is the $\ell^{q^\star}$ distance to the $\mathrm{span}(\boldsymbol{w}_\star)$ and $q^\star$ is the dual of $q$.*

*Remark* 3.3. For $q = 2$, the infimum in the definition of $d_{q^\star}^\star$ is achieved at $\mu = 0$:

$$d_2^\star(\hat{\boldsymbol{w}}_\perp) = \inf_{\mu \in \mathbb{R}} ||\hat{\boldsymbol{w}}_\perp - \mu \boldsymbol{w}_\star||_2 = ||\hat{\boldsymbol{w}}_\perp||_2. \quad (9)$$

While $d_{q^\star}^\star(\hat{\boldsymbol{w}}_\perp) \leq ||\hat{\boldsymbol{w}}_\perp||_q$ is always an upper bound, it is not tight for $q \neq 2$, except for particular choices of $\boldsymbol{w}_\star \in \mathbb{S}^{d-1}(\sqrt{d})$, for instance $\boldsymbol{w}_\star = \sqrt{d}\boldsymbol{e}_1$. This highlights how the existence of consistent attacks depend on an interplay between the Euclidean geometry of the constraint set and the $\ell^q$ geometry of the adversarial attack.

A similar condition to Eq. (8) holds for an inconsistent attack, but without the constraint $\langle \boldsymbol{w}_\star, \boldsymbol{\delta} \rangle = 0$. Since:

$$||\hat{\boldsymbol{w}}||_{q^\star} \geq ||\hat{\boldsymbol{w}}_\perp||_{q^\star} \geq d_{q^\star}^\star(\hat{\boldsymbol{w}}_\perp), \quad (10)$$

this loosens the existence condition, as expected. In particular, the stronger the overlap between the ground-truth and the model $\langle \hat{\boldsymbol{w}}, \boldsymbol{w}_\star \rangle$, the stronger the attack needs to be in order to flip the model prediction, in contrast to inconsistent perturbations which are independent of the ground-truth weights $\boldsymbol{w}_\star$. This leads to the following corollary.

**Corollary 3.4** (Existence of inconsistent attack). *Under the same setting as Proposition 3.2, an inconsistent adversarial attack exists if and only if:*

$$\epsilon ||\hat{\boldsymbol{w}}||_{q^\star} \geq |\langle \hat{\boldsymbol{w}}, x \rangle|. \quad (11)$$

*Since $\rho = d_{q^\star}^\star(\hat{\boldsymbol{w}}_\perp)/||\hat{\boldsymbol{w}}||_{q^\star} \in [0, 1]$, this further implies :* $\rho E_{\mathrm{rob}} \leq E_{\mathrm{rob}}^{\mathrm{cns}} \leq E_{\mathrm{rob}}$.

Proposition 3.2 allow us to identify the region in $\mathbb{R}^d$ which is vulnerable to consistent attacks. Indeed, defining the ground-truth orthogonal margin

$$\kappa_q(\boldsymbol{x}) = \frac{|\langle \hat{\boldsymbol{w}}, \boldsymbol{x} \rangle|}{d_{q^\star}^\star(\hat{\boldsymbol{w}}_\perp)}, \quad (12)$$

a covariate $\boldsymbol{x} \in \mathbb{R}^d$ is vulnerable to a consistent $\boldsymbol{\delta} \in H_q(\varepsilon)$ attack if and only if $\varepsilon > \kappa_q(\boldsymbol{x})$, and the *vulnerable region* is given by $\{\boldsymbol{x} \in \mathbb{R}^d : \kappa_q(\boldsymbol{x}) < \varepsilon\} \subset \mathbb{R}^d$ — a tube around the decision hyperplane. Note that this reveals two strategies for mitigating consistent adversarial attacks by increasing $\kappa_q$: (a) aligning with the target weights; (b) reducing $d_{q^\star}^\star(\hat{\boldsymbol{w}}_\perp)$. While the first option is typically beyond the statistician's control, the second option can be achieved by explicitly regularizing the training with respect to the norm dual to the attack, which is an upper-bound to $d_{q^\star}^\star(\hat{\boldsymbol{w}}_\perp)$ — see Eq. (10). This is consistent with previous theoretical results suggesting the use of the dual norm in ERM (Yin et al., 2019; Awasthi et al., 2020; Tsilivis et al., 2024; Vilucchio et al., 2026), and will be the subject interest of Section 3.3.

Another important factor in the margin $\kappa_q(\varepsilon)$ is the interplay between the underlying Euclidean ($\ell^2$) geometry defining the classifier and the $\ell^q$-geometry of the adversarial attack. This interplay is best illustrated in the Gaussian case.

**Corollary 3.5** (Existence for Gaussian covariate with general Covariance). *In the same setting as Proposition 3.2 with Gaussian covariates $\boldsymbol{x} \sim \mathcal{N}(\boldsymbol{0}, \boldsymbol{\Sigma})$, the probability that a consistent attack exists is:*

$$\mathbb{P} \left[ \exists \text{ consistent } \boldsymbol{\delta} \in H_q(\varepsilon) \right] = 2\Phi \left( \frac{\varepsilon d_{q^\star}^\star(\hat{\boldsymbol{w}}_\perp)}{\sqrt{\hat{\boldsymbol{w}}^\top \boldsymbol{\Sigma} \hat{\boldsymbol{w}}}} \right) - 1, \quad (13)$$

*where $\Phi$ is the standard normal c.d.f.*

*Remark* 3.6. The probability in Eq. (13) is non-decreasing in $q$ (non-increasing in $q^\star$) and $d$. To build intuition, we first consider the *isotropic case* $\boldsymbol{\Sigma} = 1/d \operatorname{Id}_d$ with a random predictor $\hat{\boldsymbol{w}} = m\boldsymbol{w}_\star + \sqrt{1 - m^2}\boldsymbol{\xi}$, where $\boldsymbol{\xi} \sim \mathcal{N}(0, \operatorname{Id}_d)$ and $m = 1/d\langle \boldsymbol{w}_\star, \hat{\boldsymbol{w}} \rangle$ is the correlation with the target. Then:

$$\mathbb{P} \left[ \exists \text{ consis. } \boldsymbol{\delta} \in H_q(\varepsilon) \right] = 2\Phi \left( \varepsilon\sqrt{d} \frac{\sqrt{1 - m^2}d_{q^\star}^\star(\boldsymbol{\xi})}{||\hat{\boldsymbol{w}}||_2} \right) - 1.$$

Since $d_{q^\star}^\star(\boldsymbol{\xi}) = \Theta(d^{1/q^\star})$ as $d \to \infty$, for fixed $\varepsilon = \Theta(1)$ and $q > 1$ the probability approaches one unless $m^2 = 1$ (perfect alignment). This highlights the vulnerability of high-dimensional predictors, though sparse predictors can exhibit different behavior since $d_{q^\star}^\star(\hat{\boldsymbol{w}}_\perp)$ the part of the support which does not overlap with the target. We report how the existence probability depends on the attack geometry $q$ and the dimension $d$ in Figure 3 *(Left, Center)*.

*Remark* 3.7 (Algorithmic vs. statistical irreducibility). The probability in Remark 3.6 is strictly positive whenever $m^2 < 1$, with two contributing sources. The *algorithmic gap*, the suboptimality of a given estimator relative to the best achievable one, can be reduced by tuning the estimation of $\hat{\boldsymbol{w}}$ to maximize $m$. The *statistical gap* is the residual misalignment of the *best possible* estimator at finite $\alpha$: for the continuous weight priors considered here, even the Bayes-optimal estimator achieves only $m^2 < 1$, with generalization error decaying as $\Theta(1/\alpha)$ but never vanishing (Aubin et al., 2020). This contrasts with discrete or

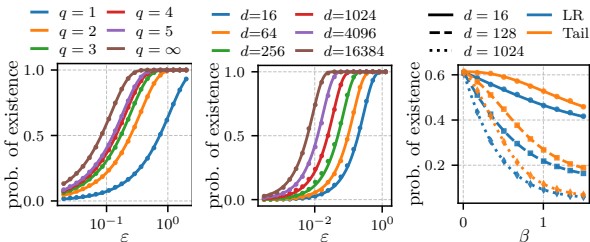

*Figure 3.* Probability of existence of consistent adversarial attacks for Gaussian covariates via Eq. (13), with $\boldsymbol{w}_\star = \sqrt{d}\boldsymbol{e}_1$ and $\hat{\boldsymbol{w}} \in \mathbb{S}^{d-1}(\sqrt{d})$ having correlation $m = 0.5$ with $\boldsymbol{w}_\star$. *(Left)* Isotropic case $\boldsymbol{\Sigma} = {}^1/_d \operatorname{Id}_d$: probability vs $\varepsilon$ for different attack geometries $q$ with $d = 10$. *(Center)* Dimensional scaling with $q = 2$, showing probability vs $\varepsilon$ for varying $d$ for $\boldsymbol{\Sigma} = {}^1/_d \operatorname{Id}_d$. *(Right)* Power law covariance $\lambda_i \sim i^{-\beta}$ for several $d$: probability vs $\beta$ at fixed $\varepsilon = 1$ for two predictor types (LR uses top $k=10$ modes; Tail uses bottom modes $(k+2){:}d$). Solid curves show theory; dots show empirical frequencies from $n = 10^4$ samples satisfying Eq. (8).

sparse priors, where a phase transition can yield exact recovery above a critical $\alpha$ (Barbier et al., 2019). Consistent attacks are thus an *irreducible* statistical phenomenon under our assumptions.

*Remark* 3.8. Consider $\boldsymbol{\Sigma}$ with *power law spectrum* with eigenvalues $\lambda_i \propto i^{-\beta}$ (for $\beta > 0$) normalized to $\sum_i \lambda_i = d$ and let $\{\boldsymbol{v}_i\}_{i=1}^d$ denote the corresponding orthonormal eigenbasis. Suppose $\boldsymbol{w}_\star = \sqrt{d}\boldsymbol{v}_1$ and that the predictor has a fixed correlation $m$ with the target. As $\beta$ increases, the spectrum concentrates on $\lambda_1$ (which grows toward $d$), reducing the attack probability for all predictors (Figure 3, *Right*). However, vulnerability also depends on where $\hat{\boldsymbol{w}}_\perp$ distributes its mass in the eigenbasis, the two cases are:

- **Low-rank (LR) alignment**: When $\hat{\boldsymbol{w}}_\perp \propto \sum_{i=2}^{k+1} \boldsymbol{v}_i$ concentrates on the top-2:$(k+1)$ eigenmodes and it benefits from eigenvalues $\lambda_2, \ldots, \lambda_{k+1}$ being large.
- **Tail alignment**: When $\hat{\boldsymbol{w}}_\perp \propto \sum_{i=k+2}^d \boldsymbol{v}_i$ concentrates on tail modes, it places norm on eigenmodes $\lambda_{k+2}, \ldots, \lambda_d$.

Figure 3 (*Right*) shows how the tail predictor is *more vulnerable* than the LR predictor by a constant logit-level factor.

### 3.2. Robust Empirical Risk Minimization

A natural question is whether robust training can effectively mitigate consistent attacks. Robust empirical risk minimization (ERM) emerged as a principled way to learn classifier rules from dataset $\mathcal{D} = \{(\boldsymbol{x}_i, y_i) \in \mathbb{R}^d \times \{-1, +1\} : i = 1, \ldots, n\}$ that are inherently robust to adversarial perturbations. From $\mathcal{D}$, the statistician estimates a classifier by optimizing the robust *empirical* (regularized) risk, defined

$$\mathcal{L}(\boldsymbol{w}) = \sum_{i=1}^n \max_{\|\boldsymbol{v}_i\|_s \leq r} \ell(y_i \langle \boldsymbol{w}, \boldsymbol{x}_i + \boldsymbol{v}_i \rangle) + \lambda \|\boldsymbol{w}\|_2^2, \quad (14)$$

where $\ell : \mathbb{R} \to \mathbb{R}_+$ is a non-increasing convex loss function and $\lambda \geq 0$ is a regularization parameter. The inner maximization over $\boldsymbol{v}_i$ models the worst-case perturbation for

each data point, constrained by the attack budget $r$ during training. The case with $r = 0$ corresponds to standard ERM while any case with $r > 0$ corresponds to robust training. Given the dataset $\mathcal{D}$, we estimate the model's weights as

$$\hat{\boldsymbol{w}} \in \operatorname*{arg\,min}_{\boldsymbol{w} \in \mathbb{R}^d} \mathcal{L}(\boldsymbol{w}). \quad (15)$$

While robust training has proven effective in practice, understanding its properties for protection to consistent attacks still requires analysis.

### 3.3. High-Dimensional Asymptotic Analysis

Motivated by Remark 3.6, we now investigate the behavior of both standard and robustly trained predictors in the high-dimensional limit where consistent adversarial attacks proliferate. More concretely, we will derive sharp asymptotic results for the case where $\hat{\boldsymbol{w}}$ is a trained predictor under the Gaussian design assumption, and discuss the benefits of robust ERM in mitigating consistent adversarial attacks. We will work under the following assumptions.

**Assumption 3.9** (Data distribution). We assume the covariates are isotropic Gaussian $\boldsymbol{x} \sim \mathcal{N}(\boldsymbol{0}, {}^1/_d \operatorname{Id}_d)$ and that labels are generated from a ground-truth linear classifier $y \sim \operatorname{Rad}(f_{\boldsymbol{w}_\star}(\boldsymbol{x}))$ where $f_{\boldsymbol{w}_\star}(\boldsymbol{x}) = \mathbb{P}(y = +1|\boldsymbol{x}) = \varphi(\langle \boldsymbol{w}_\star, \boldsymbol{x} \rangle)$ with monotonic link function $\varphi$ and ground-truth weights $\boldsymbol{w}_\star \in \mathbb{S}^{d-1}(\sqrt{d})$.

**Assumption 3.10** (Scaling of the Adversarial Strength). For a given perturbation geometry $\boldsymbol{\delta} \in B_q(\varepsilon)$ with $q > 1$, we assume that $\varepsilon = O_d(d^{-1/q^\star})$ as $d \to \infty$, where $q^\star$ is the dual. We define the rescaled radius as $\tilde{\varepsilon} = \varepsilon\, d^{1/q^\star}$.

*Remark* 3.11. As briefly discussed in Remark 3.6, Assumption 3.10 provides the right scaling for non-trivial attacks in the high-dimensional limit considered in this work: a slower scaling would result in a perturbation strength which is too weak and any faster scaling would result in a perturbation that flips any label. The same scaling was considered in previous asymptotic analyses of robust training in (Taheri et al., 2023; Javanmard et al., 2020; Tanner et al., 2025).

Robust adversarial training for well-specified linear classifiers on Gaussian covariates (Assumption 3.9) has been studied by (Vilucchio et al., 2026) in the proportional high-dimensional asymptotics where $n$ diverges with $d$ at constant ratio $\alpha = {}^n/_d = \Theta(1)$. They characterize the limiting distribution of the robust estimator $\hat{\boldsymbol{w}}$ for general $\ell^s$-robust training through a nonlinear system of equations.

For clarity of exposition, we present our result about the limiting behavior of consistent metrics for $s = 2$ and noiseless channel (for the generic form see Appendix C.3).

**Theorem 3.12** (Consistent metrics for well-specified model). *Under Assumptions 3.9 and 3.10 the metrics defined in Eqs. (2) and (3) with decision rule $\hat{y}(\boldsymbol{x}) = \operatorname{sign}(2f_{\hat{\boldsymbol{w}}}(\boldsymbol{x}) - 1)$, where $\hat{\boldsymbol{w}}$ comes from Eq. (15), concentrate in the high-dimensional limit where $n, d \to \infty$ with ${}^n/_d = \Theta(1)$ to the*

*following two dimensional integrals*

$$E_{\mathrm{rob}}^{\mathrm{cns}} = \int \mathrm{d}p(z_1, z_2) \mathbb{1}\{z_1(z_2 - \widetilde{\varepsilon}\mathcal{A}) < 0\}, \qquad (16)$$

$$E_{\mathrm{bnd}}^{\mathrm{cns}} = \int \mathrm{d}p(z_1, z_2) \mathbb{1}\{z_1(z_2 - \widetilde{\varepsilon}\mathcal{A}) < 0\}\mathbb{1}\{z_1 z_2 > 0\}, \quad (17)$$

*where* $\mathcal{A} = \sqrt{2(\tau - m^2)}\,\sqrt[q^\star]{\Gamma\left(\frac{q^\star + 1}{2}\right)/\sqrt{\pi}}$ *and* $(z_1, z_2) \sim \mathcal{N}(0, \left(\begin{smallmatrix} 1 & m \\ m & \tau \end{smallmatrix}\right))$ *with* $\tau = \frac{1}{d}\|\hat{\boldsymbol{w}}\|_2^2$ *the predictor's* $\ell_2$-*norm and* $m = \frac{1}{d}\langle \boldsymbol{w}_\star, \hat{\boldsymbol{w}} \rangle$ *its scalar product with the target.*

The proof proceeds by explicitly solving the inner maximization and then showing concentration to the previous expression in the high-dimensional limit; see Appendix C.3. The values of $\tau$ and $m$ can be obtained from the characterization of $\hat{\boldsymbol{w}}$ in (Vilucchio et al., 2026).

*Remark* 3.13 (Intuition via local fields). Equations (16) and (17) admit a *local fields* reading (Clarté et al., 2023): the Gaussian pair $(z_1, z_2) = \frac{1}{\sqrt{d}}(\langle \boldsymbol{w}_\star, \boldsymbol{x} \rangle, \langle \hat{\boldsymbol{w}}, \boldsymbol{x} \rangle)$ encodes the signed distances of $\boldsymbol{x}$ to the target and model boundaries. The first indicator fires when a consistent attack pushes $z_2$ across its boundary while $z_1$ is held fixed; the attack surface $\mathcal{A} \propto \|\hat{\boldsymbol{w}}_\perp\|_2$ only sees the predictor component orthogonal to $\boldsymbol{w}_\star$ (Proposition 3.2). The extra factor $\mathbb{1}\{z_1 z_2 > 0\}$ in $E_{\mathrm{bnd}}^{\mathrm{cns}}$ restricts to correctly classified samples.

For completeness, we report the limiting value of $E_{\mathrm{rob}}$ already derived in (Tanner et al., 2025) as being

$$E_{\mathrm{rob}} = \int \mathrm{d}p(z_1, z_2) \mathbb{1}\left\{z_1\left(z_2 - \frac{\widetilde{\varepsilon}\sqrt{2\tau}}{\pi^{\frac{1}{q^\star} + \frac{1}{2}}}\sqrt[q^\star]{\Gamma\left(\frac{q^\star + 1}{2}\right)}\right) < 0\right\}.$$

The consistent version of the errors (Eqs. (16) and (17)) depend on the quantity $\sqrt{\tau - m^2}$ while $E_{\mathrm{rob}}$ depends on $\sqrt{\tau}$. Since $\sqrt{\tau - m^2} < \sqrt{\tau}$ (as $\tau \geq m^2 > 0$), this explains why consistent attacks are less effective than inconsistent ones for the same attack strength $\varepsilon$, confirming our earlier result from Corollary 3.4 and as illustrated in Figure 4 *(Left)*. The former quantity is precisely the length of $P_\perp \hat{\boldsymbol{w}}$ appearing in Proposition 3.2. Additional experiments comparing consistent and inconsistent boundary errors are provided in Appendix B.2

Figure 4 *(Left)* shows the asymptotic dependence of the metrics in Definition 2.3 with the rescaled perturbation strength $\widetilde{\varepsilon}$ in the high-dimensional limit. This provides a quantitative measure of how strong a consistent adversarial attack needs to be to flip a certain percentage of the classifier labels: for instance, to flip $50\%$ of the labels with an $\ell_2$-attack one needs a strength of $\widetilde{\varepsilon} \approx 1$ ($\varepsilon \approx d^{-1/2}$).

Using the results from (Vilucchio et al., 2026) to find the values of $\tau$ and $m$ we can use Theorem 3.12 to analyze the consistent errors of the robustly trained $\hat{\boldsymbol{w}}$ from Eq. (15) trained on $n$ samples in dimension $d$, where both $n$ and $d$ tend to infinity with fixed ratio $\alpha = \frac{n}{d}$.

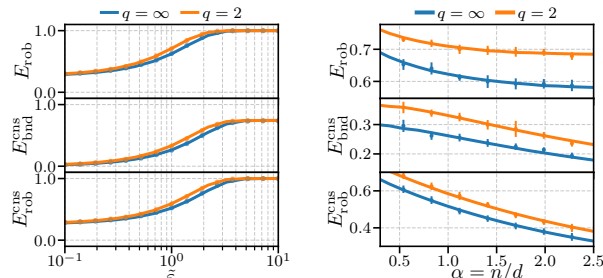

*Figure 4. (Left)* Dependence of the metrics in Eqs. (1) to (3) for the well-specified model as a function of the attackers norm. Points are 10 different simulations for $d = 500$, showing good agreement already at finite dimension. *(Right)* Performance difference for optimally regularized non-robust training under attacks constrained with different norms. The points show 10 different simulations with $d = 500$ and $n$ scaled such that $n = \alpha d$.

Figure 4 *(Right)* shows the performance of robustly trained $\hat{\boldsymbol{w}}$ as a function of $\alpha = \frac{n}{d}$, demonstrating a monotonic decrease of all metrics with the sample complexity $\alpha$ for two different attack geometries. While the errors $E_{\mathrm{rob}}$ and $E_{\mathrm{rob}}^{\mathrm{cns}}$ start from the same values, the value $E_{\mathrm{rob}}^{\mathrm{cns}}$ decreases faster with $\alpha$, indicating that with more samples the model learns more robust representations more effective against proper adversarial attacks.

We observe that $\ell^2$-attacks ($q = 2$) yield higher robust errors than $\ell^\infty$-attacks ($q = \infty$), demonstrating that attack geometry affects vulnerability beyond the difference in perturbation budget available across norms: while Assumption 3.10 ensures attacks are non-trivial, the geometric properties of each norm determine how effectively that budget translates into successful adversarial perturbations. Additional experiments are presented in Appendix B.2.

## 4. The Role of Overparameterization: Latent Variable Model

Despite many empirical works on the subject, the interplay between adversarial attacks and overparameterization remains poorly understood, with contradictory evidence on the susceptibility of large neural networks to adversarial attacks (Chen et al., 2024).

In this section, we investigate this question on a popular mathematical testbed for studying the role of overparameterization, the *latent variable model* (Hastie et al., 2022). In this model, the ground-truth classifier $f_{\boldsymbol{w}_\star}(\boldsymbol{z}) = \varphi(\langle \boldsymbol{w}_\star, \boldsymbol{z} \rangle)$ is defined in a latent space with latent covariates $\boldsymbol{z} \in \mathbb{R}^d$ and weights $\boldsymbol{w}_\star \in \mathbb{S}^{d-1}(\sqrt{d})$. Labels are generated according to the latent rule $y \sim \mathrm{Rad}(f_{\boldsymbol{w}_\star}(\boldsymbol{z}))$ as in Eq. (5). The statistician does not observe the latent covariates $\boldsymbol{z} \in \mathbb{R}^d$ directly but instead has access to a transformed representation $\boldsymbol{x} \in \mathbb{R}^p$ defined as $\boldsymbol{x} = \mathrm{F}\boldsymbol{z} + \boldsymbol{u}$, where $\mathrm{F} \in \mathbb{R}^{p \times d}$ is the *feature matrix* and $\boldsymbol{u} \sim \mathcal{N}(\boldsymbol{0}, \frac{1}{p}\mathrm{Id}_p)$ is an independent noise term.

While this model seems simplistic, recent Gaussian universality results have shown that in the proportional limit, ERM on this latent variable model is equivalent to ERM on a two-layer neural network with frozen first-layer weights (a.k.a. *random features model*) (Mei & Montanari, 2022; Goldt et al., 2022a; Gerace et al., 2020; Hu & Lu, 2023; Montanari & Saeed, 2022). This places the latent variable model as a convenient testbed to study the phenomenology associated to overparameterized networks — such as benign interpolation — in a mathematically tractable setting. In this mapping, the level of overparameterization is given by the features dimension $p$. When $p > d$, we will say the model is *overparameterized*, and when $p < d$, the model is *underparameterized*.

## 4.1. Geometry of Consistent Attacks on the Latent Space

We now discuss the geometrical properties of consistent attacks in the latent variable model. Note that in this context an adversary could either attack the latent space ($\boldsymbol{\delta} \in \mathbb{R}^d$) or feature space ($\boldsymbol{\delta} \in \mathbb{R}^p$). Considering perturbations in feature space, *i.e.* perturbations to $\boldsymbol{x}$, will result in a similar analysis as the one carried out for the model of Section 2. Therefore in the following we focus on the latter, where the conditions in Definition 2.1 translate to:

- **Target invariance:** $\boldsymbol{\delta} \in H_q(\varepsilon) = \{\boldsymbol{\delta} \in B_q(\varepsilon) : \langle \boldsymbol{w}_\star, \boldsymbol{\delta} \rangle = 0\}$.
- **Model deception:** $|\langle \hat{\boldsymbol{\theta}}, \mathrm{F}\,\boldsymbol{\delta} \rangle| > |\langle \hat{\boldsymbol{\theta}}, \mathrm{F}\,\boldsymbol{z} + \boldsymbol{u} \rangle|$ and $\mathrm{sign}(\hat{\boldsymbol{\theta}}, \mathrm{F}\,\boldsymbol{\delta}) = -\mathrm{sign}(\langle \hat{\boldsymbol{\theta}}, \mathrm{F}\,\boldsymbol{z} + \boldsymbol{u} \rangle)$

where the model weights are denoted by $\hat{\boldsymbol{\theta}} \in \mathbb{R}^p$ to avoid confusion. Adapting the argument in Section 3.1 to this case is straightforward, leading to the following characterization of consistent latent space attacks.

**Proposition 4.1** (Existence of consistent latent space attacks)**.** *Consider the setting of binary classification in the latent space model: a linear classifier defined by the weights $\boldsymbol{w}_\star \in \mathbb{S}^{d-1}(\sqrt{d})$ assign labels according to $y \sim \mathrm{Rad}(f_{\boldsymbol{w}_\star}(\boldsymbol{z}))$ where $f_{\boldsymbol{w}_\star}(\boldsymbol{z}) = \varphi_\star(\langle \boldsymbol{w}_\star, \boldsymbol{z} \rangle)$, while the statistician observes only the pairs $(\boldsymbol{x}, y) \in \mathbb{R}^p \times \{-1, +1\}$ with $\boldsymbol{x} = F\boldsymbol{z} + \boldsymbol{u} \in \mathbb{R}^d$, fitting a linear classifier $f_{\hat{\boldsymbol{\theta}}}(\boldsymbol{x}) = \varphi(\langle \hat{\boldsymbol{\theta}}, \boldsymbol{x} \rangle)$ with weights $\hat{\boldsymbol{\theta}} \in \mathbb{R}^p$. Then, a consistent attack $\boldsymbol{\delta} \in B_q(\varepsilon)$ with respect to $\boldsymbol{w}_\star \in \mathbb{S}^{d-1}(\sqrt{d})$, $\boldsymbol{z} \in \mathbb{R}^d$ and the decision function $\hat{y}(\boldsymbol{x}) = \mathrm{sign}(2f_{\hat{\boldsymbol{\theta}}}(\boldsymbol{x}) - 1)$ exists if and only if:*

$$\varepsilon d_{q^\star}^\star (P_\perp F^\top \hat{\boldsymbol{\theta}}) \geq |\langle \hat{\boldsymbol{\theta}}, F\boldsymbol{z} + \boldsymbol{u} \rangle| \qquad (18)$$

*where $P_\perp = \mathrm{Id}_d - \boldsymbol{w}_\star \boldsymbol{w}_\star^\top / d$ is the projector in the space orthogonal target weights and $q^\star$ is the dual of $q$.*

*Remark* 4.2. While in the well-specified case the vulnerable region is determined by the margin $\kappa_q(\boldsymbol{x})$ in Eq. (12), in the

latent model this is defined by the latent margin:

$$\eta_q(\boldsymbol{z}) := \frac{|\langle \hat{\boldsymbol{\theta}}, \mathrm{F}\,\boldsymbol{z} + \boldsymbol{u} \rangle|}{d_{q^\star}^\star (P_\perp F^\top \hat{\boldsymbol{\theta}})}. \qquad (19)$$

Note that this can be larger or smaller than $\kappa_q(\boldsymbol{x})$, depending on the details of the problem.

**Corollary 4.3** (Existence for Gaussian latent variables)**.** *In the case of i.i.d. Gaussian latent variables $\boldsymbol{z} \sim \mathcal{N}(0, 1/d\,\mathrm{Id}_d)$ and $\boldsymbol{u} \sim \mathcal{N}(0, 1/p\,\mathrm{Id}_p)$, the probability a consistent attack exists is given by:*

$$\mathbb{P}\left[\exists \text{ consistent } \boldsymbol{\delta} \in H_q(\varepsilon)\right] = 2\Phi\left(\frac{\sqrt{p}\varepsilon d_{q^\star}^\star (P_\perp F^\top \hat{\boldsymbol{\theta}})}{\sqrt{\|\hat{\boldsymbol{\theta}}\|_2^2 + p/d\|F^\top \hat{\boldsymbol{\theta}}\|_2^2}}\right) - 1,$$

*where $\Phi$ is the standard normal c.d.f.*

*Remark* 4.4. In the latent variable model, it is the projection of the predictor in latent space $F^\top \hat{\boldsymbol{\theta}}$ and not the predictor itself that counts for the probability of existence. In particular, the components of $\hat{\boldsymbol{\theta}} \in \mathbb{R}^p$ lying in $\mathrm{Ker}(F^\top)$ only contributes to the $\ell^2$-norm in the denominator. In other words: in the overparametrized setting $p > d$ one can reduce the probability of existence of consistent attacks by both having high alignment with the target $\langle \boldsymbol{w}_\star, F^\top \hat{\boldsymbol{\theta}} \rangle$ or by concentrating most of the norm in the $p - d$ directions in $\mathrm{Ker}(F^\top)$. This is closely related to the conditions for benign overfitting in (Bartlett et al., 2020).

The consistent and inconsistent adversarial errors associated to a predictor $\hat{y}$ in the latent space model are

$$E_{\mathrm{rob}}^{\mathrm{cns}} = \mathbb{E}\left[\max_{\substack{\boldsymbol{\delta} \in B_q(\varepsilon): \\ f_\star(\boldsymbol{x}) = f_\star(\boldsymbol{x} + \boldsymbol{\delta})}} \mathbb{1}\{y \neq \hat{y}(\mathrm{F}(\boldsymbol{z} + \boldsymbol{\delta}) + \boldsymbol{u})\}\right], \qquad (20)$$

$$E_{\mathrm{bnd}}^{\mathrm{cns}} = \mathbb{E}\left[\max_{\substack{\boldsymbol{\delta} \in B_q(\varepsilon): \\ f_\star(\boldsymbol{x}) = f_\star(\boldsymbol{x} + \boldsymbol{\delta})}} \mathbb{1}\left\{\begin{array}{c} y \neq \hat{y}(\mathrm{F}(\boldsymbol{z} + \boldsymbol{\delta}) + \boldsymbol{u}) \\ y = \hat{y}(\mathrm{F}\,\boldsymbol{z} + \boldsymbol{u}) \end{array}\right\}\right], \qquad (21)$$

$$E_{\mathrm{rob}} = \mathbb{E}\left[\max_{\boldsymbol{\delta} \in B_q(\varepsilon)} \mathbb{1}\{y \neq \hat{y}(\mathrm{F}(\boldsymbol{z} + \boldsymbol{\delta}) + \boldsymbol{u})\}\right]. \qquad (22)$$

## 4.2. High-Dimensional Asymptotics

We now move to the analysis of trained predictors in the context of the latent variable model. Consider training data $\mathcal{D} = \{(\boldsymbol{x}_i, y_i) \in \mathbb{R}^p \times \{-1, +1\} : i = 1, \ldots n\}$ independently drawn from the latent variable model. Our goal in this section is to characterize the asymptotic behavior of the estimated binary linear classifier defined by the vector $\hat{\boldsymbol{\theta}}$ estimated from $\mathcal{D}$ using Eqs. (14) and (15). Our results will hold under the following assumptions.

**Assumption 4.5** (High-dimensional limit)**.** We consider the proportional high-dimensional limit where $n, p, d \to \infty$ at fixed ratios $\alpha := n/d$ and $\psi := p/n$. For convenience, we also define $\gamma := (\alpha\psi)^{-1} = d/p$.

**Assumption 4.6** (Data Distribution Latent Space Model)**.** We assume that data $(\boldsymbol{x}, y) \in \mathbb{R}^p \times \{-1, +1\}$ is drawn from

a latent variable model with $z \sim \mathcal{N}(\mathbf{0}, {}^1\!/_d \operatorname{Id}_d)$ and ground-truth linear classifier $f_{\boldsymbol{w}_\star}(\boldsymbol{z}) = \varphi(\langle \boldsymbol{w}_\star, \boldsymbol{z} \rangle)$ with $\boldsymbol{w}_\star \in \mathbb{S}^{d-1}(\sqrt{d})$. The observed features $\boldsymbol{x} \in \mathbb{R}^p$ are generated as $\boldsymbol{x} = \mathrm{F}\,\boldsymbol{z} + \boldsymbol{u}$ with $\boldsymbol{u} \sim \mathcal{N}(0, \operatorname{Id}_p)$ independent of the other quantities and

$$\mathrm{F} = \begin{cases} \begin{bmatrix} \sqrt{\frac{p}{d}}\,\operatorname{Id}_d \\ \mathbf{0}_{(p-d)\times d} \end{bmatrix} & \text{if } p \geq d \\[2ex] \begin{bmatrix} \operatorname{Id}_p & \mathbf{0}_{p\times(d-p)} \end{bmatrix} & \text{if } p < d \end{cases}. \quad (23)$$

*Remark* 4.7. All the phenomenology that follows also hold for a random Gaussian feature matrix $F \in \mathbb{R}^{p\times d}$. The choice of feature matrix in Eq. (23) was previously considered in (Hastie et al., 2022) in the context of ridge regression. Theorem 4.8 extends the discussion to binary classification.

The main technical result for this part consists in characterizing the high dimensional behavior of the robust empirical risk minimizer $\hat{\boldsymbol{\theta}}$ as the solution of a system of self-consistent equations. For clarity of exposition we state here the result for $s = 2$ in Eqs. (14) and (15) and leave the generic statement to Appendix C.2.

**Theorem 4.8** (Self Consistent equations for Latent Space Model). *Under Assumptions 3.10, 4.5 and 4.6 the values of $m = \frac{\boldsymbol{w}_\star^\top \mathrm{F}^\top \hat{\boldsymbol{\theta}}}{d}$, $\tau = \frac{\|\mathrm{F}^\top \hat{\boldsymbol{\theta}}\|_2^2}{d} + \frac{\|\hat{\boldsymbol{\theta}}\|_2^2}{p}$ and $P = \frac{\|\hat{\boldsymbol{\theta}}\|_2^2}{p}$ concentrate in high dimension to the solution of the following system of self-consistent equations. The self-consistent equations are made of a first set*

$$\begin{cases} \bar{m} = \alpha\sqrt{\gamma}\,\mathbb{E}_{\xi\sim\mathcal{N}(0,1)}\left[\int_{\mathbb{R}} \mathrm{d}y\, \partial_\omega \mathcal{Z}_0 f_\ell\right] \\ \bar{\tau} = \alpha\gamma\,\mathbb{E}_{\xi\sim\mathcal{N}(0,1)}\left[\int_{\mathbb{R}} \mathrm{d}y\, \partial_\omega \mathcal{Z}_0 f_\ell\right] \\ \bar{V} = -\alpha\gamma\,\mathbb{E}_{\xi\sim\mathcal{N}(0,1)}\left[\int_{\mathbb{R}} \mathrm{d}y\, \partial_\omega \mathcal{Z}_0 f_\ell\right] \\ \bar{P} = 2rP^{1/s}\,\mathbb{E}_{\xi\sim\mathcal{N}(0,1)}\left[\int_{\mathbb{R}} \mathrm{d}y\, \partial_\omega \mathcal{Z}_0 f_\ell\right] \end{cases}, \quad (24)$$

*that depend on the loss function $\ell$ through $\mathcal{Z}_0 \equiv \mathcal{Z}_0(y, {}^m\!/\!\sqrt{\tau}\xi, 1 - {}^{m^2}\!/\tau)$ and $f_\ell \equiv f_\ell(y, \sqrt{\tau}\xi, V, P)$, where $\mathcal{Z}_0(y, \omega, V) = \mathbb{E}_{z\sim\mathcal{N}(\omega,V)}[\mathbb{P}(y \mid z)]$ and $f_\ell(y, \omega, V, P) = (\mathcal{P}_{V\ell(y,\cdot - y\varepsilon \sqrt[s]{P})}(\omega) - \omega)/V$.*

*The second set of equations for $\gamma \leq 1$ is*

$$\begin{cases} m = \frac{\bar{m}}{2\lambda + \bar{P} + \bar{V}(1+\frac{1}{\gamma})} \\ \tau = \frac{1}{(2\lambda + \bar{P} + \bar{V}(1+\frac{1}{\gamma}))^2}\left[\bar{m}^2\frac{1-\gamma}{\gamma} + \bar{\tau}\frac{(1+\gamma)^2 + \gamma - \gamma^2}{\gamma}\right] \\ V = \frac{1}{2\lambda + \bar{P} + \bar{V}(1+\frac{1}{\gamma})} \\ P = \frac{1}{(2\lambda + \bar{P} + \bar{V}(1+\frac{1}{\gamma}))^2}\left[\bar{m}^2 + 2\bar{\tau}\right] \end{cases} \quad (25)$$

*and for $\gamma > 1$ is*

$$\begin{cases} m = \frac{\bar{m}}{2\lambda + \bar{P} + 2\bar{V}} \\ \tau = \frac{\bar{m}^2 + 2\bar{\tau}}{(2\lambda + \bar{P} + 2\bar{V})^2} \end{cases} \quad \begin{cases} V = \frac{1}{2\lambda + \bar{P} + 2\bar{V}} \\ P = \frac{\bar{m}^2 + 2\bar{\tau}}{(2\lambda + \bar{P} + 2\bar{V})^2} \end{cases} \quad (26)$$

The proof (see Appendix C.2) uses the Gordon Min-Max theorem to characterize the minimizer of the robust risk

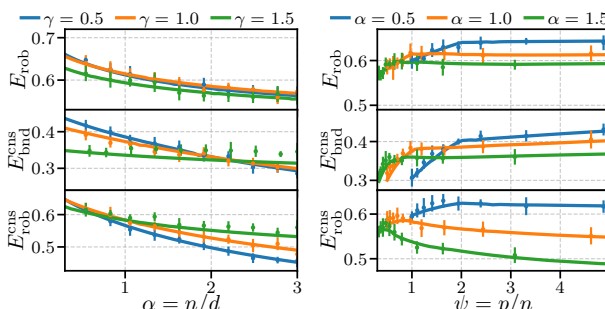

*Figure 5.* Error as a function of $\alpha$ and $\psi$ for the latent space model defined in Section 4. For both panels the lines are the exact asymptotic solution of Eqs. (24) to (26) and the error bars are average and std over 10 realizations with $d = 500$ and $p, n$ scaled accordingly. *(Left)* Robust error as a function of the number of data available during training. We see that all the metrics decrease as a function of the number of training data. *(Right)* Robust errors as a function of the number of latent space parameters. We see that while $E_{\text{rob}}$ and $E_{\text{bnd}}^{\text{cns}}$ increase with the number of features while $E_{\text{rob}}^{\text{cns}}$ decreases.

in Eq. (14) via a low-dimensional set of self-consistent equations. Such asymptotic characterizations via implicit equation systems are standard in the analysis of high-dimensional systems (Goldt et al., 2022b; Gerace et al., 2024; Cui et al., 2023; Vilucchio et al., 2025).

With the previous theorem we can characterize behavior of the proper adversarial errors in this data model.

**Theorem 4.9** (Proper Metrics for Latent Space Model). *Under the same setting of Theorem 4.8 the metrics defined in Eqs. (20) and (21) evaluated for $\hat{\boldsymbol{\theta}}$ from Eq. (15) and decision rule $\hat{y} = \text{sign}(2f_{\hat{\boldsymbol{\theta}}}(\boldsymbol{x}) - 1)$ concentrate to*

$$E_{\text{rob}}^{\text{cns}} = \int \mathrm{d}p(z_1, z_2)\, \mathbb{1}\{z_1(z_2 - \widetilde{\varepsilon}\mathcal{C}) < 0\}, \quad (27)$$

$$E_{\text{bnd}}^{\text{cns}} = \int \mathrm{d}p(z_1, z_2)\, \mathbb{1}\{z_1(z_2 - \widetilde{\varepsilon}\mathcal{C}) < 0\}\mathbb{1}\{z_1 z_2 > 0\}, \quad (28)$$

*where $\mathcal{C}$ is a complicated expression presented in Appendix C.3 and $(z_1, z_2) \sim \mathcal{N}(0, \left(\begin{smallmatrix} 1 & m \\ m & \tau \end{smallmatrix}\right))$ with $\tau, m$ the solutions of the system of equations in Theorem 4.8.*

*Remark* 4.10. The difference between the two settings appears in the constant $\mathcal{C}$, which depends on the distribution of $\hat{\boldsymbol{\theta}}$ through the parameters found by the equations in Theorem 4.8. Consequently, the local-fields interpretation of Remark 3.13 carries over, with $\mathcal{C}$ now playing the role of the available attack surface $\mathcal{A}$.

### 4.3. The Interplay Between Overparameterization and Consistent Attacks

Theorem 4.8 and 4.9 allow us to investigate the efficacy of consistent attacks on overparameterized models. We start by considering robust training with optimally tuned $\lambda$ and $r$. All the three metrics considered in this work decrease

with a function of the amount of data used in training $\alpha$, see Figure 5 *(Left)*, meaning that the more data is always beneficial. Interestingly there is a crossing for the different lines for different overparameterization level $\gamma$: the same level of overparameterization is not optimal for any amount of data availability.

In Figure 5 *(Right)* we consider the role of optimal robust training as a function of different overparameterization levels $\psi = p/n$. The three metrics present a different behavior for large $\psi$. The first two stay approximately constant, with at most a mild increase, while the third one decreases with overparmeterization and this decrease is faster and faster the more data is given to the model (greater $\alpha$).

We study the behavior of the novel metrics $E_{\mathrm{bnd}}^{\mathrm{cns}}$ and $E_{\mathrm{rob}}^{\mathrm{cns}}$ in the regieme of extreme overparameterization.

**Proposition 4.11** (Extreme overparameterization)**.** *Consider the setting of Theorems 4.8 and 4.9 with logistic loss. In the limit $\psi = p/n \to \infty$ (equivalently $\gamma = d/p \to 0^+$ at fixed $\alpha$), the order parameters admit the expansion $(m, \tau) = (m_0 \gamma, \tau_0 \gamma) + O(\gamma^2)$, where $(m_0, \tau_0)$ solve the limiting saddle-point system. Define $c := \varepsilon \sqrt{\tau_0 - m_0^2}$. Then the derivatives of the errors behave as:*

1. $\partial_\psi E_{\mathrm{rob}}^{\mathrm{cns}} = -\frac{1}{8} \frac{m_0^2}{\tau_0 \rho} \gamma^2 + O(\gamma^{5/2})$,
2. $\partial_\psi E_{\mathrm{bnd}}^{\mathrm{cns}} = \frac{1}{4\sqrt{2\pi\tau_0}} \mathcal{F}(c, \tau_0) \gamma + O(\gamma^2)$, *where*

$$\mathcal{F}(c, \tau_0) := \sqrt{\frac{\pi}{2}} \sqrt{\tau_0} \operatorname{erf}\left(\frac{c}{\sqrt{2\tau_0}}\right) - c e^{-\frac{c^2}{2\tau_0}}. \quad (29)$$

This result provides a theoretical explanation for the different behavior of the consistent error metrics in Figure 5 *(Right)*: as overparameterization increases, $E_{\mathrm{rob}}^{\mathrm{cns}}$ always decreases, while $E_{\mathrm{bnd}}^{\mathrm{cns}}$ may increase or decrease depending on the sign of Eq. (29). The proof derives the self-consistent scaling of order parameters as $\gamma \to 0^+$ and evaluates the error integrals at leading order, see Appendix C.4.

In conclusion, although overparameterized models are more vulnerable to consistent adversarial attacks, this does not *a fortiori* imply a detriment in the overall model performance, as measured for instance by $E_{\mathrm{rob}}^{\mathrm{cns}}$, since improvement of previously badly classified points can have a compensatory effect. This might provide an explanation for the contradictory observations in (Chen et al., 2024). We further verify that this dual effect is not an artifact of the linear estimator: replacing it with a trained two-layer ReLU network on the same data-generating process reproduces the same qualitative trends (Appendix B.3). Additional experiments in Appendix B.2 include ablations of other latent space models (Goldt et al., 2022a) and results on MNIST/FashionMNIST (Deng, 2012; Xiao et al., 2017).

## 5. Conclusions

We investigated the fundamental distinction between consistent and inconsistent adversarial attacks in high-dimensional binary classification, introducing the consistent robust error $E_{\mathrm{rob}}^{\mathrm{cns}}$ and the consistent boundary error $E_{\mathrm{bnd}}^{\mathrm{cns}}$ as refined metrics, and deriving their exact asymptotics for robust empirical risk minimization in both the well-specified and the latent variable settings. Curiously, we find that overparameterization has a *dual* effect on consistent robustness: the boundary error $E_{\mathrm{bnd}}^{\mathrm{cns}}$ increases—indicating heightened vulnerability among correctly classified examples—while the overall consistent robust error $E_{\mathrm{rob}}^{\mathrm{cns}}$ decreases. This counterintuitive behavior stems from the beneficial role of overparameterization on clean generalization, which compensates for the increased vulnerability of decision boundaries, offering a possible explanation for the contradictory empirical observations reported in (Chen et al., 2024).

We hope these findings, though derived from exactly-solvable models, prove valuable for the broader robust machine learning community. Rather than viewing overparameterization as inherently detrimental to adversarial robustness, one should account for the specific robustness objective at hand, as overparameterization can improve overall performance. Moreover, our characterizations provide theoretical guidance for selecting regularization parameters and attack budgets during robust training.

**Limitations and future work.** Our analysis is set in the proportional high-dimensional regime with Gaussian covariates; while this regime belongs to a broad universality class, relaxing it is an interesting direction. The latent variable model captures the random-features/NTK regime but not feature learning; extending the analysis beyond it is an important next step. Finally, target invariance $(f_\star(x + \delta) = f_\star(x))$ is strictly stronger than label preservation, so our metrics provide conservative lower bounds on label-preserving vulnerability; extending to approximate target invariance $(|f_\star(x + \delta) - f_\star(x)| < \eta)$ is a natural generalization.

## Acknowledgements

We thank the anonymous reviewers for their constructive feedback, which helped improve the paper. We thank Fanny Yang and Antonio Ribeiro for the inspiring discussions. This work was supported by the Swiss National Science Foundation under grants SNSF SMArtNet (grant number 212049) and SNSF OperaGOST (grant number 200390) and by the French government, managed by the National Research Agency (ANR), under the France 2030 program with the reference "ANR-23-IACL-0008" and the Choose France - CNRS AI Rising Talents program.

## Impact Statement

This paper presents work whose goal is to advance the field of Machine Learning. There are many potential societal consequences of our work, none which we feel must be specifically highlighted here.

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

# A. Further Related Works

**Exact Asymptotics:**    Our analysis builds upon the previous literature characterizing the properties of predictors in the high-dimensional proportional regime. This approach spans multiple theoretical frameworks: high-dimensional probability theory (Thrampoulidis et al., 2014; 2015; Vilucchio et al., 2025), statistical physics approaches (Mignacco et al., 2020; Gerace et al., 2020; Bordelon et al., 2020; Loureiro et al., 2021; Okajima et al., 2023; Adomaityte et al., 2024b;a), and random matrix theory (Bean et al., 2013b; Mai et al., 2019; Liao et al., 2020; Mei & Montanari, 2022; Xiao et al., 2022; Schröder et al., 2023; 2024; Defilippis et al., 2024; Tabanelli et al., 2025). Our work is particularly motivated by recent advances in Gaussian universality (Goldt et al., 2022a; Montanari & Saeed, 2022; Dandi et al., 2023), which demonstrate that simple Gaussian models often provide surprisingly accurate predictions for more complex data distributions in high-dimensions. This phenomenon emerges from concentration properties in high-dimensional spaces, leading to universality in generalization behavior across different covariate distributions (Tao & Vu, 2010; Donoho & Tanner, 2009; Wei et al., 2022; Dudeja et al., 2023). Our technical contribution extends these frameworks to characterize consistent adversarial attacks: while existing asymptotic analyses focus on standard generalization or robust error, we derive the first closed-form high-dimensional limits for metrics that isolate label-preserving perturbations, revealing geometric dependencies on $\sqrt{\tau - m^2}$ rather than $\sqrt{\tau}$ alone.

**Adversarial Robustness:**    Robust empirical risk minimization, commonly known as adversarial training, was pioneered in computer vision (Goodfellow et al., 2015; Madry et al., 2018) and has since evolved into a primary defense against adversarial attacks. Researchers have developed numerous approaches to improve its computational efficiency (Shafahi et al., 2019; Rice et al., 2020) and statistical properties (Zhai et al., 2019; Chen et al., 2020; Wang et al., 2023). On the theoretical front, several works have investigated the properties of robust empirical risk minimization for linear models (Raghunathan et al., 2020; Dan et al., 2020; Ribeiro et al., 2023; 2025), including sharp proportional asymptotics under different data designs (Javanmard et al., 2020; Hassani & Javanmard, 2024; Donhauser et al., 2021; Dohmatob & Scetbon, 2024; Ribeiro & Schön, 2023; Tanner et al., 2025; Peng & Yang, 2026). A common thread in most of the previous analyses is the characterization of the standard robust error $E_{\mathrm{rob}}$, treating all perturbations in $B_q(\varepsilon)$ uniformly without distinguishing whether they preserve or alter the ground-truth label. For instance, Taheri et al. (2023) and Javanmard & Soltanolkotabi (2022) derive sharp asymptotics for $E_{\mathrm{rob}}$ under adversarial training in well-specified linear models, while Clarysse et al. (2022) study adversarial perturbations directed along the signal in a noiseless setting, and Tanner et al. (2025) extend the analysis to Mahalanobis-norm attacks. Our work departs from this line by considering the consistency constraint $\langle \boldsymbol{w}_\star, \boldsymbol{\delta} \rangle = 0$, which requires a fundamentally different geometric analysis: the existence and strength of consistent attacks depend on the predictor component orthogonal to the target (Proposition 3.2 and Lemma C.10), a quantity that plays no role in unconstrained robust error. Of particular relevance is Vilucchio et al. (2026), which derives high-dimensional asymptotics for binary classification in the well-specified model and which we build upon in Section 3. Our contributions beyond their work are the geometric existence conditions (Proposition 3.2), the entire latent variable model analysis (Section 4), generalizing previous results (Mei & Montanari, 2022; Goldt et al., 2022a), and a characterization of how feature dimension $p$ affects consistent robustness differently than standard robust error (Section 4.3). Similarly, Donhauser et al. (2021) study the robust error of non-robustly trained $\ell^2$-regularized estimators in well-specified models, but do not consider adversarial training or the latent variable model. Beyond the proportional regime, Lee & Chung (2026) draw a related distinction in a feature-learning analysis of adversarial distillation, separating failures on intrinsically hard-to-learn samples from robustness loss on otherwise correctly classified points, mirroring our distinction between failures due to label-preserving perturbations and those due to limited expressivity.

**Consistent Attacks:**    The idea that adversarial attacks should be imperceptible to some metric of interest (e.g. the human eyes in vision) underlies most of the empirical literature (Szegedy et al., 2014; Goodfellow et al., 2015; Fort & Lakshminarayanan, 2024). The notion of a consistent attack in the theoretical literature was explored in Raghunathan et al. (2020); Donhauser et al. (2021); Balcan et al. (2023). These works establish that proper adversarial examples—perturbations preserving the Bayes-optimal label—can hurt robust generalization and that interpolating models may be vulnerable even without label noise. However, they do not provide: (i) geometric characterizations of when consistent attacks *exist* in high dimensions as a function of model-target alignment and attack geometry (our Propositions 3.2 and 4.1 and Corollaries 3.4 and 4.3); (ii) separate quantification of vulnerability on correctly vs. incorrectly classified points (our $E_{\mathrm{bnd}}^{\mathrm{cns}}$ metric, Eq. (3)); or (iii) exact asymptotic formulas under Gaussian designs revealing how overparameterization affects consistent robustness (our Theorems 3.12, 4.8 and 4.9). Our metrics thus enable fine-grained analysis distinguishing model failures due to label-preserving perturbations from those due to limited expressivity or sample size, providing precise predictions for the vulnerability landscape in high-dimensional linear classification.

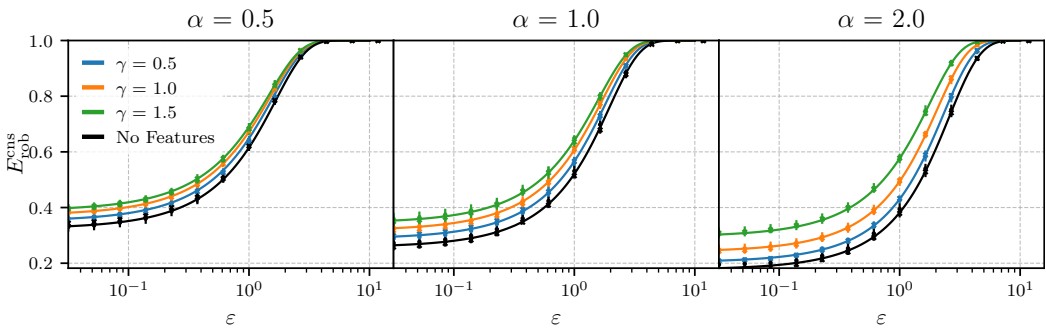

*Figure 6.* Behavior of $E_{\text{rob}}^{\text{cns}}$ for the Gaussian Features case as a function of the attack strength perturbation. The performances are for model trained as per Eqs. (14) and (15) and with the $\alpha$ and $\gamma$ specified in the figure. The error bars refer to 10 repetitions of the experiments for $d = 1024$. The metrics consider the $\hat{w}$ trained with $\lambda = 10^{-3}$, $r = 0.0$ and $s = \infty$.

## B. Additional Experiments and Figures Setting

### B.1. Setting of the Figures in the Main Text

We note that the optimization over the hyperparameters $r$ and $\lambda$ is performed with the use of the theory. In the asymptotic limit the self-consistent equations give a deterministic function of the model parameters. With gradient-free optimization techniques[1] we find the minimal values.

**Figure 3 *(Left)*** For each value of $q$, the empirical points are obtained by generating $n = 1000$ samples in $d = 10$ dimensions, with the model parameters set as $m = 0.5$. The probability is computed as the fraction of samples for which an adversarial perturbation of norm $\varepsilon$ exists, while the solid lines (theory) is computed by using Eq. (13).

**Figure 3 *(Center)*** Now we fix $q = 2$ and vary the data dimension $d$. The empirical points are again obtained by generating $n = 1000$ samples for each $d$, with model parameters $m = 0.5$ and the probability and the theoretical lines are computed as before.

**Figure 3 *(Right)*** We consider Gaussian covariates with diagonal covariance $\boldsymbol{\Sigma} = \text{diag}(\lambda_1, \ldots, \lambda_d)$ and power-law spectrum $\lambda_i \propto i^{-\beta}$, normalized so that $\sum_{i=1}^{d} \lambda_i = d$, and sweep $\beta$. We fix the target $\boldsymbol{w}_\star = \sqrt{d}\,\mathbf{e}_1$ (top eigenvector) and correlation $m = 0.5$. For each $\beta$, we construct two predictors: low-rank alignment with $\hat{\boldsymbol{w}}_\perp$ supported on the modes from 2 to $k+1$ and tail alignment with support on modes $k+2{:}d$. We set $\varepsilon = 1$, $q = 2$, $k = 10$, and evaluate $d \in \{16, 128, 1024\}$. Solid curves plot Eq. (13); empirical points are evaluated by sampling $n = 10^4$ points from $\mathcal{N}(\mathbf{0}, \boldsymbol{\Sigma})$ and report the fraction satisfying Eq. (8).

**Figure 4 *(Left)*** The curves are realized for a $w$ obtained from standard training, i.e. minimization of Eq. (15) with $r = 0$, $\lambda = 10^{-3}$ and $r = 2$. We have that the number of data is fixed at $\alpha = 1.0$. The points are produced as 10 different realizations with $d = 500$ fixed.

**Figure 4 *(Right)*** Here we show the performances of different types of attack metrics, either $L_\infty$ or $L_2$ constrained. We have that in both cases the errors correspond to optimally tuned robust estimation, $r$ and $\lambda$ chosen to have minimum errors. We have that the regularization geometry is $r = 2$ and that the geometry in adversarial training is $s = 2$. The points are produced as 10 different realizations with $d = 500$ fixed.

**Figure 5 *(Left,Right)*** We consider optimally tuned robust empirical risk minimization with $s = \infty$ and $r = 2$. The points are produced as 10 different realizations with $d = 500$ fixed and the values of $n, p$ scaled accordingly.

### B.2. Additional Experiments

**Different choice of latent space models** To test the robustness of our findings with respect to the choice of the feature matrix procedure chosen in Assumption 4.6, specifically the generation of the input data $\boldsymbol{x}$ as a function of the latent variable $\boldsymbol{z}$ we consider a different kind of latent space model. Another model used to characterize overparameterization is the hidden manifold model (Goldt et al., 2022b; Gerace et al., 2020) where the latent space covariates are still drawn from a gaussian $\boldsymbol{z} \sim \mathcal{N}(\mathbf{0}, \text{Id}_d)$ but the feature space covariates are a linear transformation $\boldsymbol{x} = \text{F}\,\boldsymbol{z}$ where $\text{F}_{ij} \sim \mathcal{N}(0, 1)$ each component independently.

---

[1]Specifically we use `scipy.optimize.minimize` with the `Nelder-Mead` algorithm.

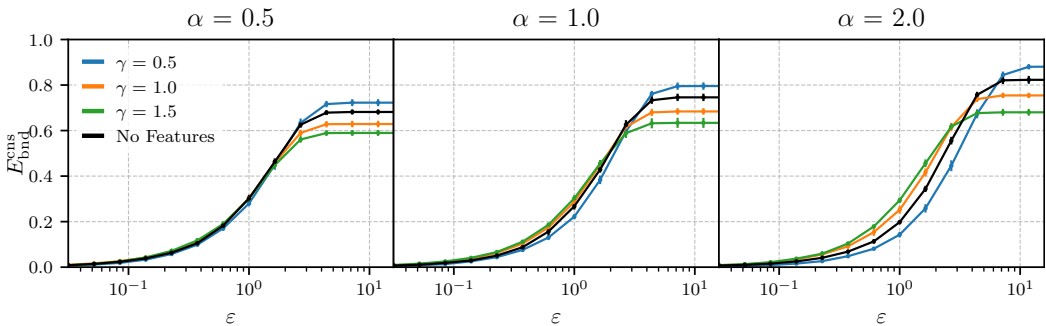

*Figure 7.* Behavior of $E_{\text{bnd}}^{\text{cns}}$ for the Gaussian Features case as a function of the attack strength perturbation. The performances are for the model trained as per Eqs. (14) and (15) with the $\alpha$ and $\gamma$ specified in the figure. The error bars refer to 10 repetitions of the experiments for $d = 1024$. The metrics consider the $\hat{w}$ trained with $\lambda = 10^{-3}$, $r = 0.0$ and $s = \infty$.

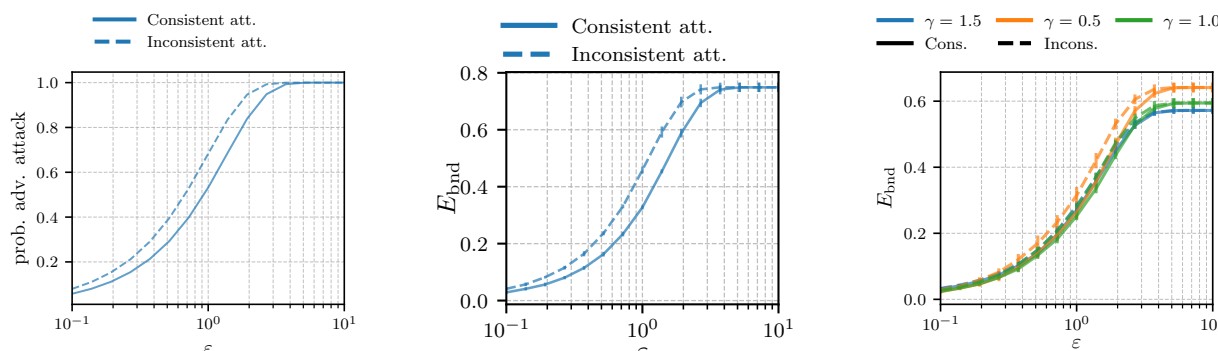

*Figure 8.* Comparison of consistent and inconsistent cases as a function of the attack strength. *(Left, Center)* Case of the well-specified model presented in Section 3. We show both the probability of existence as per Proposition 3.2 and the value of the boundary error. The error bars refer to 10 repetitions of the experiments for $d = 500$. The metrics consider the $\hat{w}$ trained with $\alpha = 1.0$, $\lambda = 10^{-2}$, $r = 0.0$ and $s = 2$. *(Right)* Case of the latent space model presented in Section 4. The error bars refer to 10 repetitions of the experiments for $d = 512$. The metrics consider the $\hat{w}$ trained with $\alpha = 1.0$, $\lambda = 10^{-2}$, $r = 0.0$ and $s = 2$.

We explore the behavior of the error metrics defined in Eqs. (20) and (21) in Figures 6 and 7. We see that the behavior is similar to the one of the model defined in the main text. The black line presented in the same figure is the performances of the well specified model in the main text. Crucially we see also in this case the metric $E_{\text{rob}}^{\text{cns}}$ equals the value of the clean generalization error in the $\varepsilon \to 0^+$ limit and it reaches one in the $\varepsilon \to \infty$ limit. We have that $E_{\text{bnd}}^{\text{cns}}$ is zero in the $\varepsilon \to 0^+$ limit.

**Errors as a function of attack strength** We additionally compare the consistent and inconsistent formulation of the boundary error as a function of the attack strength in Figure 8. The consistent boundary error is defined in Eq. (3) while the inconsistent one is defined as

$$E_{\text{bnd}} = \mathbb{E}_{(\boldsymbol{x},y)} \left[ \max_{\boldsymbol{\delta} \in B_q(\varepsilon)} \mathbb{1} \left\{ \begin{array}{c} y \neq \hat{y}(\boldsymbol{x} + \boldsymbol{\delta}) \\ y = \hat{y}(\boldsymbol{x}) \end{array} \right\} \right]. \tag{30}$$

where the crucial difference is the removal of the consistent condition $f_\star(\boldsymbol{x}) = f_\star(\boldsymbol{x} + \boldsymbol{\delta})$. We have that also in this case consistent attacks produce a milder increase in the boundary error but the qualitative behavior is the same.

In Figure 9 we show the behavior of the different attacks considered in the main text for the latent space model. The qualitative behavior of the different models (different choice of $\gamma$) is the same. We notice that again as noted in the main text for the well specified model we have that under the correct scaling of $\varepsilon$ (Assumption 3.10) we see that the $\ell_2$-bounded attack is more effective at increasing the error than the $\ell_\infty$-attack.

**Experiments on MNIST (Deng, 2012) and Fashion-MNIST (Xiao et al., 2017) datasets** In Figure 10 we show the empirical results on MNIST binary classification (digits 0 vs 1) and the Fashion-MNIST dataset for binary classification (Sandals vs Ankle Boots) using a teacher-student random features model. We load MNIST images, center them by

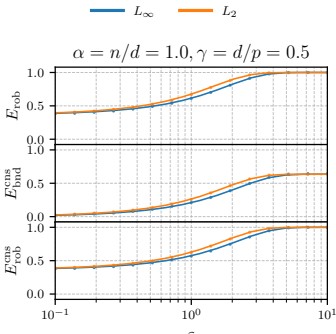
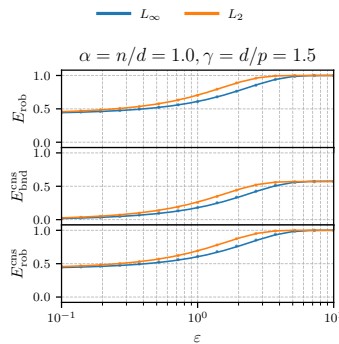

*Figure 9.* Behavior of the different adversarial errors for the Latent Space Model of Section 4. Here we show the behavior of the errors considered in the main text for different choices of $\gamma = d/p$ with $\gamma = 0.5$ for *(Left)* and $\gamma = 1.5$ for *(Right)*. We see that qualitatively the behaviors are the same in the two regimes.

subtracting the dataset mean, and denote the centered images as $\widetilde{\boldsymbol{x}}_\mu \in \mathbb{R}^{784}$. In the complex Random Fourier Features (RFF) setting, we generate a random Gaussian projection matrix $\Omega \in \mathbb{R}^{D_{\mathrm{rff}} \times 784}$ with $D_{\mathrm{rff}} = 392$ and construct latent features $\boldsymbol{z}_\mu = [\mathrm{Re}(\exp(-i\Omega\widetilde{\boldsymbol{x}}_\mu)); \mathrm{Im}(\exp(-i\Omega\widetilde{\boldsymbol{x}}_\mu))] \in \mathbb{R}^{2D_{\mathrm{rff}}}$, so that the latent dimension is $d = 2D_{\mathrm{rff}} = 784$. A teacher classifier $\boldsymbol{w}_\star \in \mathbb{R}^d$ is trained via LinearSVC (no intercept) (Pedregosa et al., 2011) on these latent features, and labels are generated through a noiseless channel. The student observes transformed features $\boldsymbol{x}_\mu = \mathrm{F}\,\boldsymbol{z}_\mu + \boldsymbol{u}_\mu$ as per Assumption 4.6. We train the student via $\ell^\infty$ ($s = \infty$) adversarial training Eqs. (14) and (15) with ridge regularization $\lambda = 10^{-2}$ and training attack budget $r = 1.0$. We evaluate the consistent adversarial metrics defined in Definition 2.3: consistent robust error $E_{\mathrm{rob}}^{\mathrm{cns}}$, and consistent boundary error $E_{\mathrm{bnd}}^{\mathrm{cns}}$ using $n_{\mathrm{gen}} = 1000$ test samples with attack budget $\varepsilon = 1.0$. Results are averaged over 10 random seed repetitions with $d = 784$ fixed, varying the sample complexity $\alpha = n/d$ and overparameterization level $\gamma = d/p$.

The results are presented in Figure 10. We see that the qualitative behavior is consistent with the Gaussian theory in Sections 4 and 4.3. The consistent boundary error $E_{\mathrm{bnd}}^{\mathrm{cns}}$ increases with the level of overparameterization while the consistent robust error $E_{\mathrm{rob}}^{\mathrm{cns}}$ increases or decreases depending on the number of data point (value of $\alpha = n/d$).

### B.3. Neural network experiment

To test whether the dual role of overparameterization predicted by our theory (Proposition 4.11) persists beyond the linear estimator, we repeat the analysis replacing the linear model by a nonlinear neural network while keeping the data-generating process identical to Assumption 4.6. This isolates the effect of the model class: the latent ground truth $\boldsymbol{w}_\star$, the labels, and the attack geometry are unchanged, and only the estimator differs.

The estimator is a two-layer ReLU multilayer perceptron with hidden dimension 256, trained to minimize the binary cross-entropy loss with Adam (learning rate $10^{-3}$, weight decay $10^{-4}$) and early stopping (patience 50 on a 20% validation split).

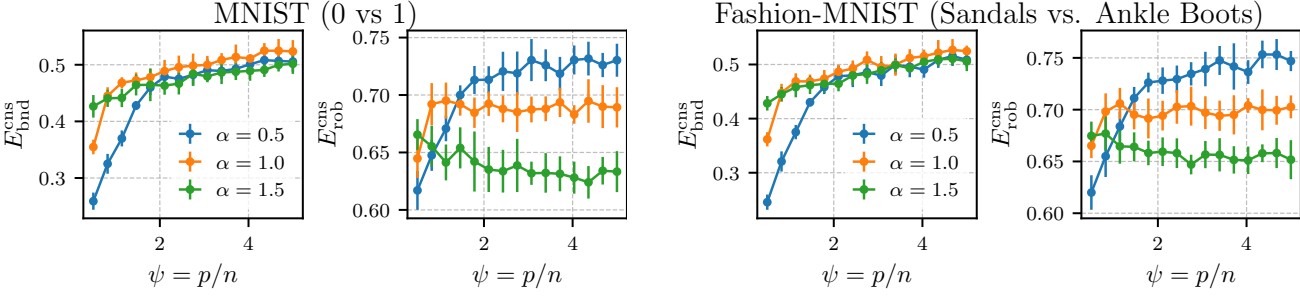

*Figure 10.* Empirical validation of the latent variable model predictions on real image datasets using complex Random Fourier Features. *(Left)* Results on MNIST binary classification (digits 0 vs 1) with $d = 784$ features. *(Right)* Results on Fashion-MNIST binary classification (Sandals vs. Ankle Boots) with $d = 784$ features. Points represent empirical averages over 10 repetitions with error bars indicating standard deviation. The qualitative behavior matches the one seen for the Gaussian model in Figure 5.

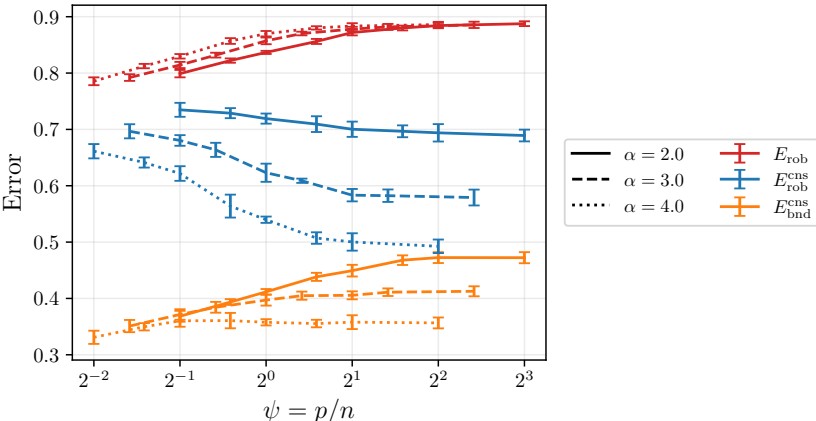

*Figure 11.* Consistent metrics for a two-layer ReLU MLP trained on the latent variable model of Assumption 4.6, as a function of the overparameterization $\psi = p/n$ ($d = 1000$, $\ell_2$ attack with $\varepsilon = 0.05$), for three sample complexities $\alpha = n/d \in \{2, 3, 4\}$. Replacing the linear estimator by a nonlinear network preserves the dual behavior predicted by the theory: $E_{\text{rob}}$ and $E_{\text{bnd}}^{\text{cns}}$ increase with $\psi$, while $E_{\text{rob}}^{\text{cns}}$ decreases. Averages over 10 seeds; error bars are one standard deviation.

We fix the latent dimension $d = 1000$ and vary the feature dimension $p$ from 1000 to 16000, so that the overparameterization ratio $\psi = p/n$ spans more than one order of magnitude. Adversarial perturbations are computed in the latent space with $\ell_2$-PGD (50 steps, 5 random restarts, $\varepsilon = 0.05$), enforcing the consistency constraint $\langle \boldsymbol{w}_\star, \boldsymbol{\delta} \rangle = 0$ by projection at each step. All metrics are evaluated on a held-out test set and averaged over 10 independent seeds.

Figure 11 reports the three metrics as a function of $p$. The qualitative predictions of the linear theory carry over to the nonlinear estimator: as overparameterization increases, the standard robust error $E_{\text{rob}}$ and the consistent boundary error $E_{\text{bnd}}^{\text{cns}}$ grow, while the overall consistent robust error $E_{\text{rob}}^{\text{cns}}$ decreases, driven by the improvement in clean generalization. This confirms that the dual effect of overparameterization is not an artifact of the linear model class.

## C. Proofs of the Results in the Main Text

The proofs in this appendix are organized according to the structure of the main results. We first present the finite-dimensional proofs in Appendix C.1, next, in Appendix C.2, we provide proofs for the high-dimensional asymptotic results, including the derivation of the self-consistent equations stated in Theorem 4.8. Finally in Appendix C.3 we detail the derivation of the limiting error metrics for both the well specified and latent space models.

### C.1. Proofs for the finite-dimensional results

In this section we gather the proofs of the finite-dimensional results presented in the main text.

*Proof of Proposition 3.2.* As discussed above, a consistent attack must satisfy the three conditions in Eqs. (6) and (7). First, note that this is only possible if $\hat{\boldsymbol{w}}_\perp \neq 0$, otherwise any admissible perturbation would *a fortiori* violate Eq. (7). Therefore, from now on we assume $\hat{\boldsymbol{w}}_\perp \neq 0$. Consider an admissible attack $\boldsymbol{\delta} \in H_q(\varepsilon)$. Since $-\boldsymbol{\delta} \in H_q(\varepsilon)$, we can always fix the sign of $\boldsymbol{\delta}$ to satisfy the constraint $\text{sign}(\langle \hat{\boldsymbol{w}}, \boldsymbol{\delta} \rangle) = -\text{sign}(\langle \hat{\boldsymbol{w}}, \boldsymbol{x} \rangle)$. The restrictive condition is the margin $|\langle \hat{\boldsymbol{w}}, \boldsymbol{\delta} \rangle| > |\langle \hat{\boldsymbol{w}}, \boldsymbol{x} \rangle|$. Since $\langle \boldsymbol{\delta}, \boldsymbol{w}_\star \rangle = 0$, we have $\langle \hat{\boldsymbol{w}}, \boldsymbol{\delta} \rangle = \langle \hat{\boldsymbol{w}}_\perp, \boldsymbol{\delta} \rangle$, and hence the margin condition is equivalent to $|\langle \hat{\boldsymbol{w}}_\perp, \boldsymbol{\delta} \rangle| > |\langle \hat{\boldsymbol{w}}, \boldsymbol{x} \rangle|$. This condition is satisfied if and only if it is satisfied by the supremum:

$$\sup_{\boldsymbol{\delta} \in H_q(\varepsilon)} |\langle \hat{\boldsymbol{w}}_\perp, \boldsymbol{\delta} \rangle| > |\langle \hat{\boldsymbol{w}}, \boldsymbol{x} \rangle| \,. \tag{31}$$

Standard results from convex analysis imply that this supremum equals $\varepsilon$ times the $\ell^{q^\star}$-distance from $\hat{\boldsymbol{w}}_\perp$ to $\text{span}(\boldsymbol{w}_\star)$ (the metric projector):

$$\sup_{\boldsymbol{\delta} \in H_q(\varepsilon)} |\langle \hat{\boldsymbol{w}}_\perp, \boldsymbol{\delta} \rangle| = \varepsilon \inf_{\mu \in \mathbb{R}} \|\hat{\boldsymbol{w}}_\perp - \mu \boldsymbol{w}_\star\|_{q^\star} =: \varepsilon \, d_{q^\star}^\star(\hat{\boldsymbol{w}}_\perp) \,, \tag{32}$$

where $q^\star$ is the Hölder conjugate of $q$: $1/q + 1/q^\star = 1$. $\qquad \square$

*Proof of Corollary 3.4.* The existence part follows the same proof as in Proposition 3.2, but without the orthogonality constraint. We then have:

$$\sup_{\boldsymbol{\delta} \in B_q(\varepsilon)} |\langle \hat{\boldsymbol{w}}, \boldsymbol{\delta} \rangle| = \varepsilon \|\hat{\boldsymbol{w}}\|_{q^\star}. \tag{33}$$

The upper-bound is immediate from Remark 2.4. The lower-bound follows from noting that $E_{\text{rob}}^{\text{cns}}(\varepsilon) = E_{\text{rob}}(\rho\varepsilon)$ and that both errors are non-decreasing functions of $\varepsilon$. $\qquad\square$

*Proof of Corollary 3.5.* Conditionally on $\hat{\boldsymbol{w}}$, we have $\langle \hat{\boldsymbol{w}}, \boldsymbol{x} \rangle \sim \mathcal{N}(0, \hat{\boldsymbol{w}}^\top \boldsymbol{\Sigma} \hat{\boldsymbol{w}})$, so $\langle \hat{\boldsymbol{w}}, \boldsymbol{x} \rangle \overset{d}{=} \sqrt{\hat{\boldsymbol{w}}^\top \boldsymbol{\Sigma} \hat{\boldsymbol{w}}} \cdot Z$ with $Z \sim \mathcal{N}(0,1)$. The condition $\varepsilon d_{q^\star}^\star(\hat{\boldsymbol{w}}_\perp) \geq |\langle \hat{\boldsymbol{w}}, \boldsymbol{x} \rangle|$ from Equation (8) yields $\mathbb{P}[|Z| \leq \varepsilon d_{q^\star}^\star(\hat{\boldsymbol{w}}_\perp)/\sqrt{\hat{\boldsymbol{w}}^\top \boldsymbol{\Sigma} \hat{\boldsymbol{w}}}] = 2\Phi(\varepsilon d_{q^\star}^\star(\hat{\boldsymbol{w}}_\perp)/\sqrt{\hat{\boldsymbol{w}}^\top \boldsymbol{\Sigma} \hat{\boldsymbol{w}}}) - 1$. $\qquad\square$

*Proof of Remark 3.8.* Assume $q = 2$ so that $q^\star = 2$ and $d_{q^\star}^\star(\cdot) = \|\cdot\|_2$. Consider the power-law spectrum

$$\lambda_i = \frac{d\, i^{-\beta}}{Z_\beta(d)}, \qquad Z_\beta(d) = \sum_{j=1}^d j^{-\beta}, \qquad (\beta > 0),$$

with orthonormal eigenbasis $\{v_i\}_{i=1}^d$, and let $\boldsymbol{w}_\star = \sqrt{d}\, v_1$. Fix a correlation level $m \in [0, 1]$ and consider predictors of the form

$$\hat{\boldsymbol{w}} = m\sqrt{d}\, v_1 + \hat{\boldsymbol{w}}_\perp, \qquad \|\hat{\boldsymbol{w}}_\perp\|_2^2 = d(1 - m^2).$$

Write $\hat{\boldsymbol{w}}_\perp = \sum_{i=2}^d c_i v_i$ with $\sum_{i=2}^d c_i^2 = d(1 - m^2)$. The Gaussian logit appearing in Eq. (13) equals

$$\gamma(\hat{\boldsymbol{w}}) := \frac{\varepsilon\|\hat{\boldsymbol{w}}_\perp\|_2}{\sqrt{\hat{\boldsymbol{w}}^\top \boldsymbol{\Sigma} \hat{\boldsymbol{w}}}} = \frac{\varepsilon\sqrt{d(1 - m^2)}}{\sqrt{\hat{\boldsymbol{w}}^\top \boldsymbol{\Sigma} \hat{\boldsymbol{w}}}}.$$

Thus, for $q = 2$, comparing predictors reduces to comparing the denominators $\sqrt{\hat{\boldsymbol{w}}^\top \boldsymbol{\Sigma} \hat{\boldsymbol{w}}}$.

Using $\boldsymbol{\Sigma} v_i = \lambda_i v_i$ and orthonormality,

$$\hat{\boldsymbol{w}}^\top \boldsymbol{\Sigma} \hat{\boldsymbol{w}} = m^2 d\, \lambda_1 + \sum_{i=2}^d c_i^2 \lambda_i = \frac{d^2}{Z_\beta(d)} \Big( m^2 + (1 - m^2) \sum_{i=2}^d p_i\, i^{-\beta} \Big), \tag{34}$$

where we introduced the weights

$$p_i := \frac{c_i^2}{d(1 - m^2)} \geq 0, \qquad \sum_{i=2}^d p_i = 1.$$

Fix $k \in \{1, \ldots, d-2\}$ and define two predictors with the same $m$ and the same $\|\hat{\boldsymbol{w}}_\perp\|_2^2 = d(1 - m^2)$ but different allocation of $\hat{\boldsymbol{w}}_\perp$:

$$\hat{\boldsymbol{w}}_\perp^{\text{LR}} := \sqrt{\frac{d(1 - m^2)}{k}} \sum_{i=2}^{k+1} \boldsymbol{v}_i, \qquad \hat{\boldsymbol{w}}_\perp^{\text{Tail}} := \sqrt{\frac{d(1 - m^2)}{d - k - 1}} \sum_{i=k+2}^d \boldsymbol{v}_i.$$

Equivalently, $p_i$ is uniform on $S_{\text{LR}} = \{2, \ldots, k+1\}$ (resp. on $S_{\text{Tail}} = \{k+2, \ldots, d\}$). Define the corresponding averages

$$\mu_{\text{LR}} := \frac{1}{k} \sum_{i=2}^{k+1} i^{-\beta}, \qquad \mu_{\text{Tail}} := \frac{1}{d - k - 1} \sum_{i=k+2}^d i^{-\beta}.$$

Then Eq. (34) yields

$$\hat{\boldsymbol{w}}^\top \boldsymbol{\Sigma} \hat{\boldsymbol{w}}\Big|_{\text{LR}} = \frac{d^2}{Z_\beta(d)} \Big( m^2 + (1 - m^2)\mu_{\text{LR}} \Big), \qquad \hat{\boldsymbol{w}}^\top \boldsymbol{\Sigma} \hat{\boldsymbol{w}}\Big|_{\text{Tail}} = \frac{d^2}{Z_\beta(d)} \Big( m^2 + (1 - m^2)\mu_{\text{Tail}} \Big).$$

Since $\beta > 0$, the sequence $a_i := i^{-\beta}$ is strictly decreasing. For every $(i, j) \in S_{\text{LR}} \times S_{\text{Tail}}$ we have $i < j$, hence $a_i > a_j$. Summing over all $k(d - k - 1)$ such pairs gives

$$(d - k - 1) \sum_{i=2}^{k+1} i^{-\beta} > k \sum_{j=k+2}^{d} j^{-\beta}.$$

Dividing by $k(d - k - 1)$ yields $\mu_{\text{LR}} > \mu_{\text{Tail}}$.

Because $m^2 + (1 - m^2)\mu$ is strictly increasing in $\mu$, the inequality $\mu_{\text{LR}} > \mu_{\text{Tail}}$ implies

$$\hat{w}^\top \Sigma \hat{w}\Big|_{\text{LR}} > \hat{w}^\top \Sigma \hat{w}\Big|_{\text{Tail}} \implies \gamma_{\text{Tail}} > \gamma_{\text{LR}}.$$

Moreover, the multiplicative separation between logits is

$$\frac{\gamma_{\text{Tail}}}{\gamma_{\text{LR}}} = \sqrt{\frac{m^2 + (1 - m^2)\mu_{\text{LR}}}{m^2 + (1 - m^2)\mu_{\text{Tail}}}} = \sqrt{\frac{m^2 + (1 - m^2)\frac{1}{k}\sum_{i=2}^{k+1} i^{-\beta}}{m^2 + (1 - m^2)\frac{1}{d-k-1}\sum_{i=k+2}^{d} i^{-\beta}}} > 1.$$

Since $\Phi$ is increasing, this implies $2\Phi(\gamma_{\text{Tail}}) - 1 > 2\Phi(\gamma_{\text{LR}}) - 1$, i.e. tail-aligned predictors are more vulnerable than LR-aligned predictors by the explicit constant factor above (at the logit level). $\qquad \square$

The proofs of Proposition 4.1 and Corollary 4.3 follow the same lines as the proofs above.

### C.2. Proof of the high-dimensional asymptotic results

In this section, we provide rigorous proofs for the theoretical results presented in the main paper, focusing on Theorem 4.8.

Central to our analysis is the Convex Gaussian MinMax Theorem (CGMT), a fundamental tool that bridges complex high-dimensional optimization problems with simpler low-dimensional counterparts. The CGMT enables us to transform our challenging primary optimization problem into a more tractable auxiliary problem, ultimately leading to the self-consistent equations presented in Equations (24) to (26).

We begin by stating the CGMT in its general form.

**Theorem C.1** (CGMT (Gordon, 1988; Thrampoulidis et al., 2014))**.** *Let $G \in \mathbb{R}^{m \times n}$ be an i.i.d. standard normal matrix and $\mathbf{g} \in \mathbb{R}^m$, $\mathbf{h} \in \mathbb{R}^n$ two i.i.d. standard normal vectors independent of each other. For compact sets $\mathcal{S}_{\boldsymbol{w}} \subset \mathbb{R}^n$ and $\mathcal{S}_{\boldsymbol{u}} \subset \mathbb{R}^n$, consider the following optimization problems with continuous function $\psi$ on $\mathcal{S}_{\boldsymbol{w}} \times \mathcal{S}_{\boldsymbol{u}}$:*

$$\mathbf{C}(G) := \min_{\boldsymbol{w} \in \mathcal{S}_{\boldsymbol{w}}} \max_{\boldsymbol{u} \in \mathcal{S}_{\boldsymbol{u}}} \boldsymbol{u}^\top G \boldsymbol{w} + \psi(\boldsymbol{w}, \boldsymbol{u}) \tag{35}$$

$$\mathcal{C}(\mathbf{g}, \mathbf{h}) := \min_{\boldsymbol{w} \in \mathcal{S}_{\boldsymbol{w}}} \max_{\boldsymbol{u} \in \mathcal{S}_{\boldsymbol{u}}} \|\boldsymbol{w}\|_2 \mathbf{g}^\top \boldsymbol{u} + \|\boldsymbol{u}\|_2 \mathbf{h}^\top \boldsymbol{w} + \psi(\boldsymbol{w}, \boldsymbol{u}) \tag{36}$$

*The following statements hold:*

1. *For all $c \in \mathbb{R}$: $\mathbb{P}(\mathbf{C}(G) < c) \leq 2\mathbb{P}(\mathcal{C}(\mathbf{g}, \mathbf{h}) \leq c)$*
2. *When $\mathcal{S}_{\boldsymbol{w}}$ and $\mathcal{S}_{\boldsymbol{u}}$ are convex sets and $\psi$ is convex-concave on $\mathcal{S}_{\boldsymbol{w}} \times \mathcal{S}_{\boldsymbol{u}}$, for all $c \in \mathbb{R}$: $\mathbb{P}(\mathbf{C}(G) > c) \leq 2\mathbb{P}(\mathcal{C}(\mathbf{g}, \mathbf{h}) \geq c)$*
3. *Consequently, for all $\mu \in \mathbb{R}$, $t > 0$: $\mathbb{P}(|\mathbf{C}(G) - \mu| > t) \leq 2\mathbb{P}(|\mathcal{C}(\mathbf{g}, \mathbf{h}) - \mu| \geq t)$*

We will utilize a specialized version of the CGMT developed by (Loureiro et al., 2021) for generalized linear models.

#### C.2.1. MATHEMATICAL PRELIMINARIES

Our analysis relies heavily on Moreau envelopes and proximal operators from convex analysis. These concepts have become essential tools in the asymptotic analysis of high-dimensional convex problems (Boyd & Vandenberghe, 2004; Parikh & Boyd, 2014). We provide key definitions below.

**Definition C.2** (Moreau Envelope)**.** For a convex function $f : \mathbb{R}^n \to \mathbb{R}$, its Moreau envelope is defined as:

$$\mathcal{M}_{Vf(\cdot)}(\boldsymbol{\omega}) = \min_{\boldsymbol{x}} \left[ \frac{1}{2V} \|\boldsymbol{x} - \boldsymbol{\omega}\|_2^2 + f(\boldsymbol{x}) \right] \tag{37}$$

where $\mathcal{M}_{Vf(\cdot)} : \mathbb{R}^n \to \mathbb{R}$.

**Definition C.3** (Proximal Operator). For a convex function $f : \mathbb{R}^n \to \mathbb{R}$, its Proximal operator is defined as:

$$\mathcal{P}_{Vf(\cdot)}(\boldsymbol{\omega}) = \arg\min_{\boldsymbol{x}} \left[ \frac{1}{2V} \|\boldsymbol{x} - \boldsymbol{\omega}\|_2^2 + f(\boldsymbol{x}) \right] \tag{38}$$

where $\mathcal{P}_{Vf(\cdot)} : \mathbb{R}^n \to \mathbb{R}^n$.

**Theorem C.4** (Gradient of Moreau Envelope (Thrampoulidis et al., 2018), Lemma D1). *For a convex function $f : \mathbb{R}^n \to \mathbb{R}$ with Moreau envelope $\mathcal{M}_{Vf(\cdot)}$ and Proximal operator $\mathcal{P}_{Vf(\cdot)}$:*

$$\boldsymbol{\nabla}_{\boldsymbol{\omega}} \mathcal{M}_{Vf(\cdot)}(\boldsymbol{\omega}) = \frac{1}{V} \left( \boldsymbol{\omega} - \mathcal{P}_{Vf(\cdot)}(\boldsymbol{\omega}) \right) \tag{39}$$

Additionally, we will use these important properties:

$$\mathcal{M}_{Vf(\cdot + \boldsymbol{u})}(\boldsymbol{\omega}) = \mathcal{M}_{Vf(\cdot)}(\boldsymbol{\omega} + \boldsymbol{u}), \quad \mathcal{P}_{Vf(\cdot + \boldsymbol{u})}(\boldsymbol{\omega}) = \boldsymbol{u} + \mathcal{P}_{Vf(\cdot)}(\boldsymbol{\omega} + \boldsymbol{u}) \tag{40}$$

which follow directly from a change of variables in the minimization.

**Definition C.5** (Dual of a number). For $a \in (1, \infty]$, we define its dual $a^\star \in [1, \infty)$ as the unique number satisfying $1/a + 1/a^\star = 1$, with the conventions $1^\star = \infty$ and $\infty^\star = 1$.

### C.2.2. ASSUMPTIONS AND PRELIMINARY DISCUSSION

We restate here all the assumptions that we make for the problem.

**Assumption C.6** (Estimation from the dataset). Given a dataset $\mathcal{D}$ made of $n$ pairs of input outputs $\{(\boldsymbol{x}_i, y_i)\}_{i=1}^n$, where $\boldsymbol{x}_i \in \mathbb{R}^d$ and $y_i \in \mathbb{R}$ we estimate the vector $\hat{\boldsymbol{w}}$ as being

$$\hat{\boldsymbol{w}} \in \arg\min_{\boldsymbol{w} \in \mathbb{R}^d} \sum_{i=1}^n \max_{\|\boldsymbol{v}_i\|_s \leq r} \ell\left( y_i \frac{\boldsymbol{w}^\top (\boldsymbol{x}_i + \boldsymbol{v}_i)}{\sqrt{d}} \right) + \lambda \|\boldsymbol{w}\|_2^2, \tag{41}$$

where $\ell : \mathbb{R} \to \mathbb{R}$ is a convex non-increasing function and where the second term is a convex regularization function whose strength can be tuned with $\lambda \in [0, \infty)$.

**Assumption C.7** (Data Distribution). We assume that data $(\boldsymbol{x}, y) \in \mathbb{R}^p \times \{-1, +1\}$ is drawn from a latent variable model with $\boldsymbol{z} \sim \mathcal{N}(\boldsymbol{0}, 1/d \, \mathrm{Id}_d)$ and ground-truth linear classifier $f_{\boldsymbol{w}_\star}(\boldsymbol{z}) = \varphi(\langle \boldsymbol{w}_\star, \boldsymbol{z} \rangle)$ with $\boldsymbol{w}_\star \in \mathbb{S}^{d-1}(\sqrt{d})$. The observed features $\boldsymbol{x} \in \mathbb{R}^p$ are generated as $\boldsymbol{x} = \mathrm{F}\,\boldsymbol{z} + \boldsymbol{u}$ with $\boldsymbol{u} \sim \mathcal{N}(0, \mathrm{Id}_p)$ independent of the other quantities and

$$\mathrm{F} = \begin{cases} \begin{bmatrix} \sqrt{\frac{p}{d}} \, \mathrm{Id}_d \\ \boldsymbol{0}_{(p-d) \times d} \end{bmatrix} & \text{if } p \geq d \\ \begin{bmatrix} \mathrm{Id}_p & \boldsymbol{0}_{p \times (d-p)} \end{bmatrix} & \text{if } p < d \end{cases} . \tag{42}$$

**Assumption C.8** (High-Dimensional Limit). We consider the proportional high-dimensional regime where both the number of training data and input dimension $n, d, p \to \infty$ at a fixed ratio $\alpha := n/d$ and $\psi = p/n$.

This setting considers most of the losses used in machine learning setups for binary classification, *e.g.* logistic, hinge, exponential losses. We additionally remark that with the given choice of regularization the whole cost function is coercive.

**Assumption C.9** (Scaling of Adversarial Norm Constraint). For a given perturbation geometry $\boldsymbol{\delta} \in B_q(r)$ with $q > 1$, we assume that $r = O_d(d^{1/q^\star + 1/2})$ as $d \to \infty$, where $q^\star$ is the dual. We define the rescaled radius as $\widetilde{\varepsilon}_t = \varepsilon/d^{1/q^\star + 1/2}$.

### C.2.3. PROBLEM SIMPLIFICATION

Recall that we start from the following optimization problem:

$$\Phi_d = \min_{\boldsymbol{w} \in \mathbb{R}^d} \sum_{i=1}^n \max_{\|\boldsymbol{v}_i\|_s \leq r} \ell\left( y_i \frac{\boldsymbol{w}^\top (\boldsymbol{x}_i + \boldsymbol{v}_i)}{\sqrt{d}} \right) + \lambda \|\boldsymbol{w}\|_2^2. \tag{43}$$

The non-increasing property of $\ell$ allows us to simplify the inner maximization, leading to an equivalent formulation

$$\Phi_d = \min_{\boldsymbol{w} \in \mathbb{R}^d} \sum_{i=1}^{n} \ell\left(y_i \frac{\boldsymbol{w}^\top \boldsymbol{x}_i}{\sqrt{d}} - \frac{r}{\sqrt{d}} \|\boldsymbol{w}\|_{s^\star}\right) + \lambda \|\boldsymbol{w}\|_2^2. \tag{44}$$

To facilitate our analysis, we introduce auxiliary variables $P = \|\boldsymbol{w}\|_{s^\star}^{s^\star}/d$ and $\hat{P}$ (the Lagrange parameter relative to this variable), which allow us to decouple the norm constraints. This leads to a min-max formulation

$$\Phi_d = \min_{\boldsymbol{w} \in \mathbb{R}^d, P} \max_{\hat{P}} \sum_{i=1}^{n} \ell\left(y_i \frac{\boldsymbol{w}^\top \boldsymbol{x}_i}{\sqrt{d}} - \frac{r}{\sqrt[s^\star]{d}} \sqrt[s^\star]{P}\right) + \lambda \|\boldsymbol{w}\|_2^2 + \hat{P} \|\boldsymbol{w}\|_{s^\star}^{s^\star} - dP\hat{P}, \tag{45}$$

where we switched the value of $r$ for its value without the scaling in $d$. This reformulation is what will allow us to apply the CGMT in subsequent steps.

It's worth noting the significance of the scaling for $r$ as detailed in Assumption C.9. In the high-dimensional limit $d \to \infty$, it's essential that all terms in $\Phi_d$ exhibit the same scaling with respect to $d$. This careful scaling ensures that our asymptotic analysis remains well-behaved and meaningful in the high-dimensional regime.

### C.2.4. SCALARIZATION AND APPLICATION OF CGMT

To facilitate our analysis, we further introduce effective regularization and loss functions, $\widetilde{r}$ and $\widetilde{\ell}$, respectively. These functions are defined as

$$\widetilde{\ell}(\boldsymbol{y}, \boldsymbol{z}) = \sum_{i=1}^{n} \ell\left(y_i z_i - \frac{r}{\sqrt[s^\star]{d}} \sqrt[s^\star]{P}\right), \quad \widetilde{r}(\boldsymbol{w}) = \|\boldsymbol{w}\|_2^2 + \hat{P} \|\boldsymbol{w}\|_{s^\star}^{s^\star}. \tag{46}$$

A crucial step in our analysis involves inverting the order of the min-max optimization. We can justify this operation by considering the minimization with respect to $\boldsymbol{w} \in \mathbb{R}^d$ at fixed values of $\hat{P}$ and $P$. This reordering is valid due to the convexity of our original problem. Specifically, the objective function is convex in $\boldsymbol{w}$ and concave in $\hat{P}$ and $P$, and the constraint sets are convex. Under these conditions, we apply Sion's minimax theorem, which guarantees the existence of a saddle point and allows us to interchange the order of minimization and maximization without affecting the optimal value.

We additionally notice that the data distribution defined in Assumption C.7 fits the framework of (Loureiro et al., 2021). Specifically, it can be seen as the case treated in Section 3.1 with the non-linearity reducing to additive Gaussian noise.

This reformulation enables us to directly apply (Loureiro et al., 2021, Lemma 11), a careful application of Theorem C.1 to scenarios involving non-separable convex regularization and loss functions. The result is a lower-dimensional equivalent of our original high-dimensional minimization problem that captures the limiting behavior of the optimum.

Throughout this subsection we use $\rho := \|\boldsymbol{w}_\star\|_2^2/d$ for the latent self-overlap of the teacher; under Assumption C.7 we have $\rho = 1$, but we keep it as a parameter to make the dependence explicit.

Consequently, our analysis now focuses on a low-dimensional functional, which takes the form

$$\widetilde{\Phi} = \min_{P, m, \eta, \tau_1} \max_{\hat{P}, \kappa, \tau_2, \nu} \left[ \frac{\kappa \tau_1}{2} - \alpha \mathcal{L}_\ell - \frac{\eta}{2\tau_2}(\nu^2 \rho + \kappa^2) - \frac{\eta \tau_2}{2} - \mathcal{L}_r + m\nu - P\hat{P} \right] \tag{47}$$

where we have restored the min max order of the problem.

In this expression, $\boldsymbol{g}$ and $\boldsymbol{h}$ are independent Gaussian vectors with i.i.d. standard normal components. The terms $\mathcal{L}_\ell$ and $\mathcal{L}_r$ represent the scaled averages of Moreau Envelopes (Eq. (37))

$$\mathcal{L}_\ell = \frac{1}{n} \mathbb{E}\left[ \mathcal{M}_{\frac{\tau_1}{\kappa} \widetilde{\ell}(\boldsymbol{y}, \cdot)}\left( \frac{m}{\sqrt{\rho}} \boldsymbol{s} + \eta \boldsymbol{h} \right) \right] \tag{48}$$

$$\mathcal{L}_r = \frac{1}{d} \mathbb{E}\left[ \mathcal{M}_{\frac{\eta}{\tau_2} \widetilde{r}(\cdot)}\left( \frac{\eta}{\tau_2}(\kappa \boldsymbol{g} + \nu \boldsymbol{w}_\star) \right) \right] \tag{49}$$

The extremization problem in Eq. (47) is related to the original optimization problem in Eq. (43) as it can be thought as the leading part in the limit $n, d \to \infty$.

This dimensional reduction is the step that allows us to study the asymptotic properties of our original high-dimensional problem through a more tractable low-dimensional optimization and thus have in the end a low dimensional set of equations to study.

It's important to note that the optimization problem $\widetilde{\Phi}$ is still implicitly defined in terms of the dimension $d$ and, consequently, as a function of the sample size $n$. We introduce two variables

$$\boldsymbol{w}_{\mathrm{eq}} = \mathcal{P}_{\frac{\eta^*}{\tau_2^*} \widetilde{r}(\cdot)} \left( \frac{\eta^*}{\tau_2^*} \left( \nu^* \mathbf{t} + \kappa^* \mathbf{g} \right) \right), \quad \boldsymbol{z}_{\mathrm{eq}} = \mathcal{P}_{\frac{\tau_1^*}{\kappa^*} \widetilde{\ell}(\cdot, \mathbf{y})} \left( \frac{m^*}{\sqrt{\rho}} \mathbf{s} + \eta^* \mathbf{h} \right) \tag{50}$$

where $(\eta^\star, \tau_2^\star, P^\star, \hat{P}^\star, \kappa^\star, \nu^\star, m^\star, \tau_1^\star)$ are the extremizer points of $\widetilde{\Phi}$.

Building upon (Loureiro et al., 2021, Theorem 5), we can establish a convergence result. Let $\hat{\boldsymbol{w}}$ be an optimal solution of the problem defined in Eq. (43), and let $\hat{\boldsymbol{z}} = \frac{1}{\sqrt{d}} \boldsymbol{X} \hat{\boldsymbol{w}}$, where $\boldsymbol{X} \in \mathbb{R}^{n \times d}$ stacks the data points by row. For any Lipschitz function $\varphi_1 : \mathbb{R}^d \to \mathbb{R}$ and any separable, pseudo-Lipschitz function $\varphi_2 : \mathbb{R}^n \to \mathbb{R}$, there exist constants $\epsilon, C, c > 0$ such that

$$\mathbb{P} \left( \left| \varphi_1 \left( \frac{\hat{\mathbf{w}}}{\sqrt{d}} \right) - \mathbb{E} \left[ \varphi_1 \left( \frac{\boldsymbol{w}_{\mathrm{eq}}}{\sqrt{d}} \right) \right] \right| \geq \epsilon \right) \leq \frac{C}{\epsilon^2} e^{-cn\epsilon^4} \tag{51}$$

$$\mathbb{P} \left( \left| \varphi_2 \left( \frac{\hat{\mathbf{z}}}{\sqrt{n}} \right) - \mathbb{E} \left[ \varphi_2 \left( \frac{\boldsymbol{z}_{\mathrm{eq}}}{\sqrt{n}} \right) \right] \right| \geq \epsilon \right) \leq \frac{C}{\epsilon^2} e^{-cn\epsilon^4} \tag{52}$$

It demonstrates that the limiting values of any function depending on $\hat{\boldsymbol{w}}$ and $\hat{\boldsymbol{z}}$ can be computed by taking the expectation of the same function evaluated at $\boldsymbol{w}_{\mathrm{eq}}$ or $\boldsymbol{z}_{\mathrm{eq}}$, respectively. This convergence property allows us to translate results from our low-dimensional proxy problem back to the original high-dimensional setting with high probability.

### C.2.5. DERIVATION OF SADDLE POINT EQUATIONS

We now want to show that extremizing the values of $m, \eta, \tau_1, P, \hat{P}, \nu, \tau_2, \kappa$ lead to the optimal value $\widetilde{\Phi}$ of Eq. (47). We are going to directly derive the saddle point equations and then argue that in the high-dimensional limit they become exactly the ones reported in the main text.

We obtain the first set of derivatives, which depend only on the loss function and the channel part, by taking the derivatives with respect to $m, \eta, \tau_1, P$:

$$
\begin{aligned}
\frac{\partial}{\partial m} &: \nu = \alpha \frac{\kappa}{n\tau_1} \mathbb{E} \left[ \left( \frac{m}{\eta\rho} \mathbf{h} - \frac{\mathbf{s}}{\sqrt{\rho}} \right)^\top \mathcal{P}_{\frac{\tau_1}{\kappa} \widetilde{\ell}(\cdot, \mathbf{y})} \left( \frac{m}{\sqrt{\rho}} \mathbf{s} + \eta\mathbf{h} \right) \right] \\
\frac{\partial}{\partial \eta} &: \tau_2 = \alpha \frac{\kappa}{\tau_1} \eta - \frac{\kappa\alpha}{\tau_1 n} \mathbb{E} \left[ \mathbf{h}^\top \mathcal{P}_{\frac{\tau_1}{\kappa} \widetilde{\ell}(\cdot, \mathbf{y})} \left( \frac{m}{\sqrt{\rho}} \mathbf{s} + \eta\mathbf{h} \right) \right] \\
\frac{\partial}{\partial \tau_1} &: \frac{\tau_1^2}{2} = \frac{1}{2} \alpha \frac{1}{n} \mathbb{E} \left[ \left\| \frac{m}{\sqrt{\rho}} \mathbf{s} + \eta\mathbf{h} - \mathcal{P}_{\frac{\tau_1}{\kappa} \widetilde{\ell}(\cdot, y)} \left( \frac{m}{\sqrt{\rho}} \mathbf{s} + \eta\mathbf{h} \right) \right\|_2^2 \right] \\
\frac{\partial}{\partial P} &: \hat{P} = \frac{\alpha}{n} \partial_P \mathbb{E} \left[ \mathcal{M}_{\frac{\tau_1}{\kappa} \widetilde{\ell}(\boldsymbol{y}, \cdot)} \left( \frac{m}{\sqrt{\rho}} \boldsymbol{s} + \eta\boldsymbol{h} \right) \right]
\end{aligned}
\tag{53}
$$

By taking the derivatives with respect to the remaining variables $\kappa, \nu, \tau_2, \hat{P}$ we obtain a set of equations depending on

regularization and prior over the teacher weights

$$
\begin{aligned}
\frac{\partial}{\partial \kappa} &: \tau_1 = \frac{1}{d}\mathbb{E}\left[\mathbf{g}^\top \mathcal{P}_{\frac{\eta}{\tau_2}\widetilde{r}(\cdot)}\left(\frac{\eta}{\tau_2}\left(\nu \boldsymbol{w}_\star + \kappa \mathbf{g}\right)\right)\right] \\
\frac{\partial}{\partial \nu} &: m = \frac{1}{d}\mathbb{E}\left[\boldsymbol{w}_\star^\top \mathcal{P}_{\frac{\eta}{\tau_2}\widetilde{r}(\cdot)}\left(\frac{\eta}{\tau_2}\left(\nu \boldsymbol{w}_\star + \kappa \mathbf{g}\right)\right)\right] \\
\frac{\partial}{\partial \tau_2} &: \frac{1}{2d}\frac{\tau_2}{\eta}\mathbb{E}\left[\left\|\frac{\eta}{\tau_2}\left(\nu \boldsymbol{w}_\star + \kappa \mathbf{g}\right) - \mathcal{P}_{\frac{\eta}{\tau_2}\widetilde{r}(\cdot)}\left(\frac{\eta}{\tau_2}\left(\nu \boldsymbol{w}_\star + \kappa \mathbf{g}\right)\right)\right\|_2^2\right] = \frac{\eta}{2\tau_2}\left(\nu^2 \rho + \kappa^2\right) - m\nu - \kappa \tau_1 + \frac{\eta \tau_2}{2} + \frac{\tau_2}{2\eta}\frac{m^2}{\rho} \\
\frac{\partial}{\partial \hat{P}} &: P = \frac{1}{d}\partial_{\hat{P}}\mathbb{E}\left[\mathcal{M}_{\frac{\eta}{\tau_2}\widetilde{r}(\cdot)}\left(\frac{\eta}{\tau_2}(\kappa \boldsymbol{g} + \nu \boldsymbol{w}_\star)\right)\right]
\end{aligned}
\tag{54}
$$

The rewriting of these equations in the desired form in Theorem 4.8 follows from the same considerations as in (Loureiro et al., 2021, Appendix C.2), specifically two changes of variables and a integration by parts.

To perform this rewriting the first ingredient we need is the following change of variables

$$
\begin{aligned}
m &\leftarrow m\,, & \tau &\leftarrow \eta^2 + \frac{m^2}{\rho}\,, & V &\leftarrow \frac{\tau_1}{\kappa}\,, & P &\leftarrow P\,, \\
\bar{V} &\leftarrow \frac{\tau_2}{\eta}\,, & \bar{\tau} &\leftarrow \kappa^2\,, & \bar{m} &\leftarrow \nu\,, & \bar{P} &\leftarrow \hat{P}\,.
\end{aligned}
\tag{55}
$$

and the use of Isserlis' theorem (Isserlis, 1918) to simplify the expectations involving the Gaussian vectors $\boldsymbol{g}$ and $\boldsymbol{h}$.

### C.2.6. REWRITING OF THE SADDLE POINT EQUATIONS

We now rewrite the zero-gradient equations in a form more amenable to numerical solution. Recently, Vilucchio et al. (2025) showed the relationship of these equations with the so-called "state evolution" for the AMP algorithm. This rewriting brings a computational advantage, since their numerical stability is improved.

To simplify the equations we first notice that, due to the separability, all terms depending on the proximal of either $\widetilde{\ell}$ or $\widetilde{r}$ simplify the $n$ or $d$ at the denominator. This cancellation is crucial as it eliminates the explicit dependence on the problem dimension, allowing us to derive dimension-independent equations.

To obtain specifically the form implied in the main text we introduce

$$
\mathcal{Z}_0(y, \omega, V) = \int \frac{\mathrm{d}x}{\sqrt{2\pi V}} e^{-\frac{1}{2V}(x-\omega)^2} \delta\left(y - f^0(x)\right)\,.
\tag{56}
$$

The function $\mathcal{Z}_0$ can be interpreted as a partition function of the conditional distribution $\mathbb{P}_{\text{out}}$ and contains all of the information about the label generating process. This brings the first set of equations to be

$$
\begin{cases}
\bar{m} = \alpha \mathbb{E}_\xi\left[\int_{\mathbb{R}} \mathrm{d}y\, \partial_\omega \mathcal{Z}_0(y, \sqrt{\frac{m^2}{\tau}}\xi, 1 - \frac{m^2}{\tau})f_\ell(\sqrt{\tau}\xi, P, r)\right] \\
\bar{\tau} = \alpha \mathbb{E}_\xi\left[\int_{\mathbb{R}} \mathrm{d}y\, \mathcal{Z}_0(y, \sqrt{\frac{m^2}{\tau}}\xi, 1 - \frac{m^2}{\tau})f_\ell^2(\sqrt{\tau}\xi, P, r)\right] \\
\bar{V} = -\alpha \mathbb{E}_\xi\left[\int_{\mathbb{R}} \mathrm{d}y\, \mathcal{Z}_0(y, \sqrt{\frac{m^2}{\tau}}\xi, 1 - \frac{m^2}{\tau})\partial_\omega f_\ell(\sqrt{\tau}\xi, P, r)\right] \\
\bar{P} = rP^{1/s}\alpha \mathbb{E}_\xi\left[\int_{\mathbb{R}} \mathrm{d}y\, \mathcal{Z}_0(y, \sqrt{\frac{m^2}{\tau}}\xi, 1 - \frac{m^2}{\tau})yf_\ell(\sqrt{\tau}\xi, P, r)\right]
\end{cases}
\tag{57}
$$

where $\xi \sim \mathcal{N}(0,1)$ and we defined for notational convenience the following

$$
f_\ell(\omega, P, r) = \frac{1}{V}\left(\mathcal{P}_{V\ell(y\cdot - r\, {}^{s_\star}\!\sqrt{P})}(\omega) - \omega\right)\,.
\tag{58}
$$

Next, we simplify the remaining set of equations by introducing

$$
\mathcal{Z}_{w_\star}(\varrho, \Lambda) = \int \mathrm{d}w\, e^{-\frac{1}{2}w^2} e^{-\frac{\Lambda}{2}w^2 + \varrho w}\,,
\tag{59}
$$

which, in turn, leads to the second set of equations

$$\begin{cases} \bar{m} = \frac{1}{\sqrt{\gamma}}\mathbb{E}_{\xi}\left[\partial_{\varrho}\mathcal{Z}_{w_{\star}}f_w^1\right] \\ \bar{\tau} = (1+\gamma)\mathbb{E}_{\xi}\left[\mathcal{Z}_{w_{\star}}(f_w^1)^2\right] + (1-\gamma)\mathbb{E}_{\xi}\left[(f_w^2)^2\right] \\ \bar{V} = (1+\gamma)\mathbb{E}_{\xi}\left[\mathcal{Z}_{w_{\star}}\partial_{\varrho}f_w^1\right] + (1-\gamma)\mathbb{E}_{\xi}\left[\partial_{\varrho}f_w^2\right] \\ \bar{P} = \gamma\mathbb{E}_{\xi}\left[\mathcal{Z}_{w_{\star}}|f_w^1|^{s^{\star}}\right] + (1-\gamma)\mathbb{E}_{\xi}\left[|f_w^2|^{s^{\star}}\right] \end{cases} \qquad \text{for } \gamma \leq 1\,, \tag{60}$$

$$\begin{cases} \bar{m} = \frac{1}{\sqrt{\gamma}}\mathbb{E}_{\xi}\left[\partial_{\varrho}\mathcal{Z}_{w_{\star}}f_w\right] \\ \bar{\tau} = 2\mathbb{E}_{\xi}\left[\mathcal{Z}_{w_{\star}}f_w\right] \\ \bar{V} = 2\mathbb{E}_{\xi}\left[\mathcal{Z}_{w_{\star}}\partial_{\varrho}f_w\right] \\ \bar{P} = \mathbb{E}_{\xi}\left[\mathcal{Z}_{w_{\star}}|f_w|^{s^{\star}}\right] \end{cases} \qquad \text{for } \gamma > 1\,, \tag{61}$$

depends on the regularization norm and the prior over the ground truth weights through $\mathcal{Z}_{w_{\star}} \equiv \mathcal{Z}_{w_{\star}}(\bar{m}\xi/\sqrt{(1+\gamma)\bar{\tau}}, \bar{m}^2/(1+\gamma)\bar{\tau})$, $f_w^1 \equiv f_w(\sqrt{\bar{\tau}}\sqrt{1+\frac{1}{\gamma}}\xi, \bar{V}(1+\frac{1}{\gamma}), \bar{P}/2)$ and $f_w^2 \equiv f_w(\sqrt{\bar{\tau}}\xi, \bar{V}, \bar{P}/2)$ for $\gamma \leq 1$ and $\mathcal{Z}_{w_{\star}} \equiv \mathcal{Z}_{w_{\star}}(\bar{m}\xi/\sqrt{2\bar{\tau}}, \bar{m}^2/2\bar{\tau})$ and $f_w \equiv f_w(\sqrt{2\bar{\tau}}\xi, 2\bar{V}, \bar{P}/2)$ for $\gamma > 1$ where

$$f_w(\varrho, \Lambda, \pi) = \arg\min_z \left[\pi|z|^{r^{\star}} + \lambda z^2 + \frac{\Lambda}{2}z^2 - \varrho z\right]. \tag{62}$$

### C.2.7. SPECIALIZATION FOR $s = 2$

To obtain the specific form of the saddle-point equations presented in Theorem 4.8, one applies the closed-form proximal operator for $s = 2$,

$$f_w(\varrho, \Lambda, \pi) = \frac{\varrho}{\lambda + \pi + \Lambda}\,. \tag{63}$$

This renders Equations (25) and (26) amenable to explicit integration: they become Gaussian integrals over the variable $\xi$. Performing these Gaussian integrals yields the form shown in the main text.

### C.3. Limiting Values for the Error Metrics

We start by proving the following lemma that will be useful in the following.

**Lemma C.10** (Consistent perturbation minimization)**.** *For any decreasing function* $g\colon \mathbb{R} \to \mathbb{R}$, *any* $y \in \{\pm 1\}$, *and any vectors* $\boldsymbol{x}, \boldsymbol{w} \in \mathbb{R}^d$,

$$\max_{\boldsymbol{\delta}:\|\boldsymbol{\delta}\|_q \leq \varepsilon, \langle\boldsymbol{w}_{\star}, \boldsymbol{\delta}\rangle=0} g\left(y\frac{\langle\boldsymbol{w}, \boldsymbol{x}+\boldsymbol{\delta}\rangle}{\sqrt{d}}\right) = \inf_{\kappa\in\mathbb{R}} g\left(y\frac{\langle\boldsymbol{w}, \boldsymbol{x}\rangle}{\sqrt{d}} - \frac{\varepsilon}{\sqrt{d}}\|\boldsymbol{w}-\kappa\boldsymbol{w}_{\star}\|_{q^{\star}}\right) \tag{64}$$

*Proof of Lemma C.10.* Since $g : \mathbb{R} \to \mathbb{R}$ in Eq. (64) is decreasing (but not necessarily continuous), it suffices to minimize its argument and then evaluate $g$ at the minimizer. Splitting the argument,

$$\min_{\boldsymbol{\delta}:\|\boldsymbol{\delta}\|_q \leq \varepsilon, \langle\boldsymbol{w}_{\star}, \boldsymbol{\delta}\rangle=0} y\langle\boldsymbol{w}, \boldsymbol{x}+\boldsymbol{\delta}\rangle = y\langle\boldsymbol{w}, \boldsymbol{x}\rangle + \min_{\boldsymbol{\delta}:\|\boldsymbol{\delta}\|_q \leq \varepsilon, \langle\boldsymbol{w}_{\star}, \boldsymbol{\delta}\rangle=0} y\langle\boldsymbol{w}, \boldsymbol{\delta}\rangle\,, \tag{65}$$

the first term does not depend on $\boldsymbol{\delta}$, so we focus on the second. Since $y \in \{+1, -1\}$, the change of variables $\boldsymbol{\delta} \to y\boldsymbol{\delta}$ leaves the constraints invariant (the $q$-norm is even and the orthogonality constraint is linear) and removes the $y$ factor from the objective. We can therefore write

$$\min_{\boldsymbol{\delta}:\|\boldsymbol{\delta}\|_q \leq \varepsilon, \langle\boldsymbol{w}_{\star}, \boldsymbol{\delta}\rangle=0} \langle\boldsymbol{w}, \boldsymbol{\delta}\rangle = \min_{\boldsymbol{\delta}:\|\boldsymbol{\delta}\|_q \leq \varepsilon} \sup_{\kappa\in\mathbb{R}} \langle\boldsymbol{w}, \boldsymbol{\delta}\rangle + \kappa\langle\boldsymbol{w}_{\star}, \boldsymbol{\delta}\rangle = \min_{\boldsymbol{\delta}:\|\boldsymbol{\delta}\|_q \leq \varepsilon} \sup_{\kappa\in\mathbb{R}} \langle\boldsymbol{w}+\kappa\boldsymbol{w}_{\star}, \boldsymbol{\delta}\rangle\,, \tag{66}$$

where we introduced a Lagrange multiplier $\kappa$ for the linear constraint. Strong duality holds for this min–sup, so we may interchange the order:

$$\sup_{\kappa\in\mathbb{R}} \min_{\boldsymbol{\delta}:\|\boldsymbol{\delta}\|_q \leq \varepsilon} \langle\boldsymbol{w}+\kappa\boldsymbol{w}_{\star}, \boldsymbol{\delta}\rangle = \sup_{\kappa\in\mathbb{R}}\left(-\varepsilon\|\boldsymbol{w}+\kappa\boldsymbol{w}_{\star}\|_{q^{\star}}\right) = -\varepsilon\inf_{\kappa\in\mathbb{R}}\|\boldsymbol{w}+\kappa\boldsymbol{w}_{\star}\|_{q^{\star}}\,, \tag{67}$$

using the duality $1/q + 1/q^\star = 1$. Reintroducing the $1/\sqrt{d}$ factor gives

$$-\frac{\varepsilon}{\sqrt{d}} \inf_{\kappa \in \mathbb{R}} \|w + \kappa w_\star\|_{q^\star}, \tag{68}$$

which is the term that appears inside $g$ on the right-hand side of Eq. (64). $\qquad\square$

### C.3.1. PROPER ERROR METRICS

We now compute the limiting values of the proper error metrics. Lemma C.10 is the basis of the proofs of Theorems 3.12 and 4.9: it tells us that, given the joint limiting distribution of $w_\star$ and $w$, one can evaluate the limiting form of the attack term as a function of $\kappa$ and then take the extremization over $\kappa$.

*Proof of Theorem 3.12.* We first recall that the limiting distribution of the trained estimator is

$$\hat{w} \sim \mathcal{P}_{\bar{V}\widetilde{r}(\cdot, \bar{P})}\left(\bar{m}w_\star + \sqrt{\bar{\tau} - \bar{m}^2}\boldsymbol{\xi}\right), \tag{69}$$

where $\bar{m}$, $\bar{\tau}$, $\bar{V}$ and $\bar{P}$ are the solution of the saddle-point equations of Vilucchio et al. (2026). In the case $s = 2$ the proximal can be evaluated in closed form, yielding $\hat{w}_i = m(w_\star)_i + \sqrt{\tau - m^2}\,\xi_i$ with $\xi \sim \mathcal{N}(0, \mathrm{Id}_d)$, where $m$ and $\tau$ are the (unbarred) order parameters determined by the saddle point.

We now evaluate the limit of the term in Eq. (68), which is the $\ell^{q^\star}$-norm of a Gaussian vector with a specific covariance. For any $p \in [1, \infty]$ the $\ell^p$-norm of a Gaussian vector concentrates around its expectation: this is an application of (Vershynin, 2018, Theorem 5.5) to the Lipschitz function $\|M \cdot\|_p$, where $M$ is the square root of the covariance of the Gaussian vector. The explicit limit for our case can be found as (Biau & Mason, 2015, Corollary 1). Under the scaling $\widetilde{\varepsilon} := \varepsilon\, \sqrt[q^\star]{d}/\sqrt{d}$ of the adversarial budget, we obtain

$$\frac{\varepsilon}{\sqrt{d}} \inf_{\kappa} \|\widetilde{w} + \kappa w_\star\|_{q^\star} \xrightarrow[d \to \infty]{} \widetilde{\varepsilon} \inf_{\kappa} \frac{\sqrt{2}}{\pi^{(2q^\star)^{-1}}} \sqrt{(m + \kappa)^2 + \tau - m^2} \sqrt[q^\star]{\Gamma\left(\frac{q^\star + 1}{2}\right)} \tag{70}$$

the previous equation is always minimized for $\kappa = -m$ and thus it leads to

$$\widetilde{\varepsilon}\sqrt{2}\sqrt{\tau - m^2} \sqrt[q^\star]{\frac{\Gamma((q^\star + 1)/2)}{\sqrt{\pi}}}. \tag{71}$$

The existence of such a closed form is a peculiarity of the case $s = 2$. In the general case the limiting value still exists because the proximal of a convex function is itself Lipschitz; applying Eq. (69) then yields

$$\mathcal{A} = \inf_{\kappa \in \mathbb{R}} \mathbb{E}_\xi\left[\int \mathrm{d}w_\star \,|f_w - \kappa w_\star|^{q^\star}\right]^{1/q^\star}. \tag{72}$$

where $f_w = \mathcal{P}_{\bar{V}\widetilde{r}(\cdot, \bar{P})}\left(\bar{m}w_\star + \sqrt{\bar{\tau} - \bar{m}^2}\xi\right)$ is the limiting distribution of the entries of $\hat{w}$.

Once the attack-surface term concentrates to its limit, the limiting distribution of the metrics Eqs. (2) and (3) can be obtained by the *local fields* method (Clarté et al., 2023): in the well-specified model the relevant pair of random variables is

$$\begin{pmatrix} \frac{\langle w_\star, x\rangle}{\sqrt{d}} \\ \frac{\langle \hat{w}, x\rangle}{\sqrt{d}} \end{pmatrix} \sim \mathcal{N}\left(\begin{pmatrix} 0 \\ 0 \end{pmatrix}, \begin{pmatrix} 1 & m \\ m & \tau \end{pmatrix}\right), \tag{73}$$

where $m$ and $\tau$ are the order parameters obtained from the saddle-point equations. $\qquad\square$

*Proof of Theorem 4.9.* The argument mirrors the proof of Theorem 3.12; only the limiting form of the attack-surface constant changes, and that change is governed by the block structure of the feature matrix F (Assumption 4.6).

*Step 1: Apply Lemma C.10 in the latent attack form.* By Proposition 4.1, a consistent latent-space perturbation $\boldsymbol{\delta} \in \mathbb{R}^d$ with $\langle w_\star, \boldsymbol{\delta}\rangle = 0$ acts on the model field through $\langle\hat{\boldsymbol{\theta}}, \mathrm{F}\boldsymbol{\delta}\rangle$. Applying Lemma C.10 with $w \leftarrow \mathrm{F}^\top\hat{\boldsymbol{\theta}}$ and $x \leftarrow z$, for any non-increasing $g$,

$$\max_{\boldsymbol{\delta} \in H_q(\varepsilon)} g\left(y\,\frac{\langle\hat{\boldsymbol{\theta}}, \mathrm{F}(z + \boldsymbol{\delta}) + u\rangle}{\sqrt{p}}\right) = \inf_{\kappa \in \mathbb{R}} g\left(y\,\frac{\langle\hat{\boldsymbol{\theta}}, \mathrm{F}z + u\rangle}{\sqrt{p}} - \frac{\varepsilon}{\sqrt{p}}\left\|\mathrm{F}^\top\hat{\boldsymbol{\theta}} - \kappa w_\star\right\|_{q^\star}\right). \tag{74}$$

Under Assumption 3.10, the attack contribution concentrates to $\widetilde{\varepsilon}\,\mathcal{C}$, with

$$\mathcal{C} \;=\; \lim_{d,p,n\to\infty}\; \frac{1}{\sqrt{p}\,d^{1/q^\star}}\, \inf_{\kappa\in\mathbb{R}} \left\|\mathrm{F}^\top\hat{\boldsymbol{\theta}} - \kappa\boldsymbol{w}_\star\right\|_{q^\star}.$$

*Step 2: Block decomposition of $\mathrm{F}^\top\hat{\boldsymbol{\theta}}$.* Assumption 4.6 prescribes

$$\mathrm{F} = \begin{cases} \left[\sqrt{\tfrac{p}{d}}\,\mathrm{Id}_d;\; \mathbf{0}_{(p-d)\times d}\right] & p \geq d, \\ \left[\mathrm{Id}_p,\; \mathbf{0}_{p\times(d-p)}\right] & p < d. \end{cases}$$

Hence, writing $\hat{\boldsymbol{\theta}} = (\hat{\boldsymbol{\theta}}^{\mathrm{s}}, \hat{\boldsymbol{\theta}}^{\mathrm{k}})$ for the block decomposition induced by $\mathrm{F}$:

- For $\gamma \leq 1$ ($p \geq d$): $\hat{\boldsymbol{\theta}}^{\mathrm{s}} = \hat{\boldsymbol{\theta}}_{1:d} \in \mathbb{R}^d$ couples to $\boldsymbol{w}_\star$ through $\mathrm{F}^\top\hat{\boldsymbol{\theta}} = \sqrt{p/d}\,\hat{\boldsymbol{\theta}}^{\mathrm{s}}$; $\hat{\boldsymbol{\theta}}^{\mathrm{k}} = \hat{\boldsymbol{\theta}}_{d+1:p} \in \mathbb{R}^{p-d}$ lies in $\ker(\mathrm{F}^\top)$ and is therefore decoupled from $\boldsymbol{w}_\star$. The two blocks contribute to empirical averages over $\hat{\boldsymbol{\theta}}$ with weights $\gamma = d/p$ and $1 - \gamma$.
- For $\gamma > 1$ ($p < d$): $\mathrm{F}^\top\hat{\boldsymbol{\theta}} = (\hat{\boldsymbol{\theta}}^\top, \mathbf{0}_{d-p}^\top)^\top$, so the single block $\hat{\boldsymbol{\theta}}^{\mathrm{s}} = \hat{\boldsymbol{\theta}}$ carries all the signal-aligned information, while the remaining $d - p$ entries of $\mathrm{F}^\top\hat{\boldsymbol{\theta}}$ are exactly zero.

The rewriting of the saddle-point equations in Appendix C.2 (Equations (25) and (26)) implements precisely this decomposition: it yields the limiting laws $f_w^1$ (signal block, $\gamma \leq 1$), $f_w^2$ (kernel block, $\gamma \leq 1$) and $f_w^3$ (single block, $\gamma > 1$) for the empirical distribution of the entries of $\hat{\boldsymbol{\theta}}$, expressed through the proximal of the regularizer.

*Step 3: Limit of the dual norm.* The $\ell^{q^\star}$-norm of a vector with i.i.d. asymptotically Gaussian entries concentrates by the Lipschitz-Gaussian inequality (Vershynin (2018, Theorem 5.5); cf. the well-specified argument). Splitting the sum that defines $\left\|\mathrm{F}^\top\hat{\boldsymbol{\theta}} - \kappa\boldsymbol{w}_\star\right\|_{q^\star}^{q^\star}$ across the blocks identified in Step 2 and using the empirical-distribution limits from Loureiro et al. (2021, Lemma 5) yields

$$\mathcal{C} \;=\; \inf_{\kappa\in\mathbb{R}} \begin{cases} \left(\gamma\,\mathbb{E}_\xi\!\int\!\mathrm{d}w_\star\,h^1(w_\star)\left|f_w^1 - \kappa w_\star\right|^{q^\star} + (1-\gamma)\,\mathbb{E}_\xi\!\int\!\mathrm{d}w_\star\,h^2(w_\star)\left|f_w^2 - \kappa w_\star\right|^{q^\star}\right)^{1/q^\star} & \gamma \leq 1, \\[2mm] \left(\mathbb{E}_\xi\!\int\!\mathrm{d}w_\star\,h^3(w_\star)\left|f_w^3 - \kappa w_\star\right|^{q^\star} + \mathbb{E}_\xi\!\int\!\mathrm{d}w_\star\,h^3(w_\star)\left|\kappa w_\star\right|^{q^\star}\right)^{1/q^\star} & \gamma > 1, \end{cases} \tag{75}$$

where the densities $h^i(w_\star)$ encode the Gaussian "channel" prior on $w_\star$ at the relevant saddle point and the $f_w^i$ are the limiting proximals of Step 2. Explicitly, with

$$h(w_\star; \varrho, \Lambda) \;=\; e^{-w_\star^2/2}\, e^{-\Lambda w_\star^2/2 + \varrho\, w_\star},$$

one has $h^1 = h(w_\star; \bar{m}\xi/\sqrt{(1+\gamma)\bar{\tau}}, \bar{m}^2/(1+\gamma)\bar{\tau})$, $h^2 = h(w_\star; 0, 0)$ and $h^3 = h(w_\star; \bar{m}\xi/\sqrt{2\bar{\tau}}, \bar{m}^2/2\bar{\tau})$. The extra $|\kappa w_\star|^{q^\star}$ term in the $\gamma > 1$ branch reflects the contribution of the $d - p$ identically-zero entries of $\mathrm{F}^\top\hat{\boldsymbol{\theta}}$, for which $f_w^3 - \kappa w_\star$ reduces to $-\kappa w_\star$.

*Step 4: Local-fields conclusion.* Substituting $\mathcal{C}$ into Eq. (74) and applying the local-fields argument from the proof of Theorem 3.12 with $z_1 = \langle\boldsymbol{w}_\star, \boldsymbol{z}\rangle/\sqrt{d}$ and $z_2 = \langle\hat{\boldsymbol{\theta}}, \boldsymbol{x}\rangle/\sqrt{p}$ (jointly Gaussian with the covariance stated in Theorem 4.9, by Loureiro et al. (2021, Lemma 5)) yields Eq. (27) and Eq. (28). $\qquad\square$

**Corollary C.11** (Euclidean simplification of $\mathcal{C}$). *Under the setting of Theorem 4.9, when the attack norm is Euclidean ($q = 2$, hence $q^\star = 2$), the attack-surface constant in Equation (75) simplifies to*

$$\mathcal{C} \;=\; \sqrt{\tau_\ell - m^2}, \tag{76}$$

*where $\tau_\ell := \left\|\mathrm{F}^\top\hat{\boldsymbol{\theta}}\right\|_2^2/d$ is the latent-space self-overlap and $m := \langle\mathrm{F}^\top\hat{\boldsymbol{\theta}}, \boldsymbol{w}_\star\rangle/d$ is the target alignment, both as in Theorem 4.8.*

*Proof.* For $q^\star = 2$, the dual norm appearing in Eq. (74) is Euclidean, and the infimum admits a one-dimensional Hilbert-space projection in closed form. For any $v \in \mathbb{R}^d$ and any nonzero $\boldsymbol{w}_\star \in \mathbb{R}^d$,

$$\inf_{\kappa\in\mathbb{R}} \|v - \kappa\boldsymbol{w}_\star\|_2^2 \;=\; \|v\|_2^2 - \frac{\langle v, \boldsymbol{w}_\star\rangle^2}{\|\boldsymbol{w}_\star\|_2^2}, \qquad \text{attained at } \kappa^\star = \frac{\langle v, \boldsymbol{w}_\star\rangle}{\|\boldsymbol{w}_\star\|_2^2}, \tag{77}$$

which is Pythagoras applied to the orthogonal decomposition $v = \frac{\langle v, \boldsymbol{w}_\star \rangle}{\|\boldsymbol{w}_\star\|^2} \boldsymbol{w}_\star + P_\perp v$ with $P_\perp = \mathrm{Id}_d - \boldsymbol{w}_\star \boldsymbol{w}_\star^\top / \|\boldsymbol{w}_\star\|^2$.

Specializing to $v = \mathrm{F}^\top \hat{\boldsymbol{\theta}}$, and using $\|\boldsymbol{w}_\star\|_2^2 = d$ from $\boldsymbol{w}_\star \in \mathbb{S}^{d-1}(\sqrt{d})$,

$$\inf_{\kappa \in \mathbb{R}} \left\| \mathrm{F}^\top \hat{\boldsymbol{\theta}} - \kappa \boldsymbol{w}_\star \right\|_2^2 = \left\| \mathrm{F}^\top \hat{\boldsymbol{\theta}} \right\|_2^2 - \frac{\langle \mathrm{F}^\top \hat{\boldsymbol{\theta}}, \boldsymbol{w}_\star \rangle^2}{d} = d\,\tau_\ell - \frac{(d\,m)^2}{d} = d\,(\tau_\ell - m^2),$$

attained at $\kappa^\star = m$. Substituting into the definition of $\mathcal{C}$ in Step 3 of the proof of Theorem 4.9 — for $q^\star = 2$ the dimensional prefactor exactly cancels the $\sqrt{d}$ inside the square root — gives Eq. (76).

The derivation does not invoke the case split between $\gamma \leq 1$ and $\gamma > 1$ that is needed in Eq. (75) for general $q^\star$. The reason is that the Euclidean norm $\|v - \kappa \boldsymbol{w}_\star\|_2^2$ depends on $v = \mathrm{F}^\top \hat{\boldsymbol{\theta}}$ only through its first two moments against $\boldsymbol{w}_\star$, namely $\|v\|^2 / d$ and $\langle v, \boldsymbol{w}_\star \rangle / d$, which are precisely the global order parameters $\tau_\ell$ and $m$. The component-by-component decomposition of $\hat{\boldsymbol{\theta}}$ into the signal block (limiting law $f_w^1$ for $\gamma \leq 1$, $f_w^3$ for $\gamma > 1$) and the kernel block (limiting law $f_w^2$, only present for $\gamma \leq 1$) collapses when the second-order overlaps are summed:

$$\gamma \cdot \mathbb{E}_\xi \int \mathrm{d}w_\star \, h^1 \, (f_w^1)^2 + (1 - \gamma) \cdot \mathbb{E}_\xi \int \mathrm{d}w_\star \, h^2 \, (f_w^2)^2 = \tau_\ell,$$

$$\gamma \cdot \mathbb{E}_\xi \int \mathrm{d}w_\star \, h^1 \, f_w^1 \, w_\star + (1 - \gamma) \cdot \mathbb{E}_\xi \int \mathrm{d}w_\star \, h^2 \, f_w^2 \, w_\star = m,$$

$$\gamma \cdot \mathbb{E}_\xi \int \mathrm{d}w_\star \, h^1 \, w_\star^2 + (1 - \gamma) \cdot \mathbb{E}_\xi \int \mathrm{d}w_\star \, h^2 \, w_\star^2 = 1,$$

by the very definitions of $\tau_\ell$, $m$, and the normalization $\|\boldsymbol{w}_\star\|^2 / d = 1$ (and analogously for $\gamma > 1$ with $f_w^3$ and $h^3$). Expanding the quadratic $\left| f_w^i - \kappa w_\star \right|^2$ in Eq. (75) and using these three identities reduces both branches of Eq. (75) to

$$\mathcal{C}^2 = \inf_{\kappa \in \mathbb{R}} \left[ \tau_\ell - 2\kappa m + \kappa^2 \right] = \tau_\ell - m^2,$$

confirming Eq. (76) directly from Eq. (75). $\qquad\square$

### C.4. Asymptotic Behavior Under Extreme Overparameterization

This subsection is devoted to the proof of Proposition 4.11. We analyze the behavior of the consistent errors in the limit $\psi \to \infty$, which corresponds to $\gamma = d/p \to 0^+$ at fixed sample complexity $\alpha$ and adversarial budget $\varepsilon$.

*Proof of Proposition 4.11.* We first establish the self-consistent scaling of the order parameters as $\gamma \to 0^+$. The key observation for characterizing this limit is that the proximal operator of the logistic loss $\ell(y, z) = \log(1 + \exp(-yz))$, which appears in Eq. (24) from Theorem 4.8, satisfies the implicit characterization from Briceño-Arias et al. (2019, Proposition 2): for any $V \in (0, \infty)$ and $v \in \mathbb{R}$,

$$\mathcal{P}_{V\ell}(v) = v + W_{\exp(-v)}\big( V \exp(-v) \big), \tag{78}$$

where $W$ is the generalized Lambert-W function defined by the relation

$$\bar{v}(\exp(\bar{v}) + r) = v \quad \Leftrightarrow \quad \bar{v} = W_r(v) \qquad (\forall \bar{v}, v \in \mathbb{R}, \forall r \in (0, +\infty)). \tag{79}$$

In the limit $\gamma \to 0^+$, we analyze the solutions of equations Eqs. (24) and (25). Following the same arguments as in Aubin et al. (2020, Appendix V.4), we postulate a $\sqrt{\gamma}$-expansion of the solution of the self-consistent equation.

First we notice that, regardless of the leading order of the variables $\bar{m}, \bar{\tau}, \bar{P}, \bar{V}$, the prior part of the self-consistent equations in Eq. (25) implies that

$$m = \frac{\bar{m}_0}{\bar{V}_0} \gamma + O(\gamma^{3/2}), \quad \tau = \frac{\bar{m}_0^2 + \bar{\tau}_0}{\bar{V}_0^2} \gamma + O(\gamma^{3/2}), \quad V = \frac{1}{\bar{V}_0} \gamma + O(\gamma^{3/2}), \quad P = O(\gamma^2), \tag{80}$$

where we used $\bar{m} = \bar{m}_0 + \bar{m}_1 \sqrt{\gamma} + O(\gamma), \bar{\tau} = \bar{\tau}_0 + \bar{\tau}_1 \sqrt{\gamma} + O(\gamma), \bar{P} = \bar{P}_0 + \bar{P}_1 \sqrt{\gamma} + O(\gamma), \bar{V} = \bar{V}_0 + \bar{V}_1 \sqrt{\gamma} + O(\gamma)$.

Define $f_{\text{out}}(v, V) := (\mathcal{P}_V \ell(v) - v)/V$ and set $u := \mathcal{P}_V \ell(v) - v$. With $r := e^{-v} > 0$, the proximal characterization yields $u = W_r(Vr)$, equivalently

$$u\big(e^u + r\big) = Vr. \tag{81}$$

Since $\partial_u[u(e^u + r)]_{u=0} = 1 + r > 0$, the implicit function theorem ensures that, for fixed $v$ (hence fixed $r$), the solution $u$ is analytic in $V$ around 0, so we may write $u = \sum_{k \geq 1} V^k a_k(v)$. Expanding $e^u = 1 + u + \frac{1}{2}u^2 + O(u^3)$ in Eq. (81) and matching powers of $V$ gives $a_1(v) = \frac{r}{1+r} = \frac{1}{1+e^v}$ and $a_2(v) = -\frac{r^2}{(1+r)^3} = -\frac{e^v}{(1+e^v)^3}$, hence

$$f_{\text{out}}(v, V) = \frac{u}{V} = \frac{1}{1 + e^v} - V \frac{e^v}{(1 + e^v)^3} + O(V^2). \tag{82}$$

In particular, if $v = v_1\gamma + v_2\gamma^{3/2} + O(\gamma^2)$ and $V = V_1\gamma + V_2\gamma^{3/2} + O(\gamma^2)$ so that $v \to 0$, then using $\frac{1}{1+e^v} = \frac{1}{2} - \frac{v}{4} + O(v^3)$ in Eq. (82) yields

$$f_{\text{out}}(v, V) = \frac{1}{2} - \left(\frac{v_1}{4} + \frac{V_1}{8}\right)\gamma - \left(\frac{v_2}{4} + \frac{V_2}{8}\right)\gamma^{3/2} + O(\gamma^2). \tag{83}$$

The first non-trivial contribution appears at zeroth order in $\gamma$, validating the scaling

$$m = O(\gamma), \quad \tau = O(\gamma), \quad V = O(\gamma), \quad P = O(\gamma^2), \tag{84}$$

while $\bar{m}, \bar{\tau}, \bar{V}, \bar{P} = O(1)$. We write the leading-order expansion as $m = m_0\gamma + O(\gamma^{3/2})$ and $\tau = \tau_0\gamma + O(\gamma^{3/2})$, where $(m_0, \tau_0)$ solve the limiting saddle-point system to leading order.

To analyze the derivatives of the adversarial errors, we start from their integral forms in Theorem 4.9. By the dominated convergence theorem, we can interchange differentiation and integration. Recalling that $\partial_\psi = -\alpha\gamma^2\partial_\gamma$, we compute the derivative with respect to $\gamma$ and then multiply by $-\alpha\gamma^2$.

With this scaling and the specific choice $s = q = 2$, the geometric quantity $\mathcal{C}$ appearing in Theorem 4.9 behaves as $\mathcal{C} = \varepsilon\sqrt{\tau_0 - m_0^2}\sqrt{\gamma} + O(\gamma^{3/2})$ at leading order. To obtain such a simple form it is a matter of carefully expanding Eq. (75) with the scaling found in Eq. (84) and only keeping the leading order terms after the integration.

For $E_{\text{rob}}^{\text{cns}}$, applying Leibniz rule to the integral and evaluating at leading order in $\gamma$ gives

$$\partial_\psi E_{\text{rob}}^{\text{cns}} = -\frac{\gamma^2 m_0^2}{8\tau_0\rho} + O\left(\gamma^{5/2}\right), \tag{85}$$

where lower-order terms cancel. The negativity follows since $m_0^2, \tau_0, \rho > 0$ for any non-trivial solution.

For $E_{\text{bnd}}^{\text{cns}}$, the same calculation yields

$$\partial_\psi E_{\text{bnd}}^{\text{cns}} = \frac{\gamma}{4\sqrt{2\pi}\tau_0^{3/2}} \left(\sqrt{\frac{\pi}{2}}\tau_0^{3/2} \text{erf}\left(\frac{c}{\sqrt{2}\sqrt{\tau_0}}\right) - c\tau_0 e^{-\frac{c^2}{2\tau_0}}\right) + O(\gamma^2), \tag{86}$$

where $c = \varepsilon\sqrt{\tau_0 - m_0^2}$. The sign of this expression is determined by the bracketed term $\mathcal{F}(c, \tau_0)$ defined in Eq. (29), thus completing the proof. $\qquad\square$

## D. Replica derivation of the self-consistent equations

In this appendix we give an alternative derivation of the asymptotic equations of Theorem 4.8 using the (heuristic) replica method from statistical physics (Mezard et al., 1987). The replica computation reproduces *exactly* the self-consistent equations Eqs. (24) to (26) obtained from the rigorous CGMT-based derivation of Appendix C.2, providing an independent cross-check. We sketch the main steps and refer to (Loureiro et al., 2021; Vilucchio et al., 2025) for the detailed manipulations, which are standard in the high-dimensional asymptotics literature.

### D.1. Gibbs measure and free energy

To the adversarial training problem Eqs. (14) and (15) we associate the Gibbs measure at inverse temperature $\beta > 0$,

$$\mu_\beta(\mathrm{d}\boldsymbol{w}) = \frac{1}{\mathcal{Z}_\beta} \exp\left\{-\beta \sum_{\mu=1}^n \ell\left(y_\mu \frac{\boldsymbol{w}^\top \boldsymbol{x}_\mu}{\sqrt{p}} - \frac{\varepsilon_t}{\sqrt{p}}\|\boldsymbol{w}\|_{q^\star}\right) - \frac{\beta\lambda}{2}\|\boldsymbol{w}\|_2^2\right\} \mathrm{d}\boldsymbol{w}, \tag{87}$$

whose partition function $\mathcal{Z}_\beta$ concentrates on the minimizers of the adversarially trained risk as $\beta \to \infty$. The quantity of interest is the dataset-averaged free-energy density

$$f := -\lim_{\beta \to \infty} \lim_{d \to \infty} \frac{1}{\beta\,d}\,\mathbb{E}_\mathcal{D} \log \mathcal{Z}_\beta\,, \tag{88}$$

which we compute via the replica trick $\mathbb{E}_\mathcal{D} \log \mathcal{Z}_\beta = \lim_{r \to 0^+} \partial_r \mathbb{E}_\mathcal{D} \mathcal{Z}_\beta^r$.

### D.2. Replicated partition function and order parameters

After averaging over the data, the $r$-replicated partition function depends on the linear combinations $\nu_\mu := \langle \boldsymbol{w}_\star, \boldsymbol{z}_\mu \rangle / \sqrt{d}$ and $\lambda_\mu^a := \langle \boldsymbol{w}^a, \boldsymbol{x}_\mu \rangle / \sqrt{p}$ (for $a = 1, \dots, r$), which are jointly Gaussian with the covariances

$$\mathbb{E}\,\nu_\mu^2 = \frac{\|\boldsymbol{w}_\star\|_2^2}{d}\,, \quad \mathbb{E}\,\lambda_\mu^a \nu_\mu = \frac{\sqrt{\gamma}}{d}\,\boldsymbol{w}_\star^\top\,\mathrm{F}^\top\,\boldsymbol{w}^a\,, \quad \mathbb{E}\,\lambda_\mu^a \lambda_\mu^b = \frac{1}{p}\,(\boldsymbol{w}^a)^\top (\mathrm{F}\,\mathrm{F}^\top + \mathrm{Id}_p)\,\boldsymbol{w}^b\,. \tag{89}$$

Introducing the order parameters $\rho, m^a, Q^{ab}, P^a$ for these covariances together with the associated Lagrange multipliers $\hat{\rho}, \hat{m}^a, \hat{Q}^{ab}, \hat{P}^a$ via Fourier representations of the constraining $\delta$-functions, the replicated partition function takes the saddle-point form

$$\mathbb{E}_\mathcal{D} \mathcal{Z}_\beta^r \;\propto\; \int \mathrm{d}\Theta\,\mathrm{d}\hat{\Theta}\;\exp\{p\,\Phi^{(r)}(\Theta, \hat{\Theta})\}\,, \qquad \Phi^{(r)} = \Psi_t + \alpha\gamma\,\Psi_y^{(r)} + \Psi_w^{(r)}\,, \tag{90}$$

where $\Psi_t$ is the Legendre-transform (trace) term, and $\Psi_y^{(r)}, \Psi_w^{(r)}$ are the channel and prior contributions, whose explicit forms follow standard manipulations (Mezard et al., 1987; Loureiro et al., 2021) and are omitted here.

### D.3. Replica-symmetric ansatz and zero-temperature limit

We adopt the replica-symmetric (RS) ansatz $m^a = m$, $Q^{aa} = V + q$, $Q^{ab} = q$ for $a < b$, $P^a = P$, together with matching scalings for the hatted variables, and take the zero-temperature limit through the standard substitutions $V \to V/\beta$, $\hat{V} \to \beta\hat{V}$, $\hat{q} \to \beta^2\hat{q}$, $\hat{m} \to \beta\hat{m}$, $\hat{P} \to \beta\hat{P}$. The RS channel term becomes

$$\widetilde{\Psi}_y = -\mathbb{E}_\xi\left[\int \mathrm{d}y\,\mathcal{Z}_0\Big(y, \tfrac{m}{\sqrt{q}}\xi, \rho - \tfrac{m^2}{q}\Big)\,\mathcal{M}_{V\ell(y,\,\cdot\,-\varepsilon_t\,\sqrt[q]{P})}\big(\sqrt{q}\,\xi\big)\right]\,, \tag{91}$$

with $\mathcal{Z}_0(y, \omega, V) = \int (2\pi V)^{-1/2}\,e^{-(x-\omega)^2/(2V)}\,P_0(y \mid x)\,\mathrm{d}x$ the conditional-channel partition function, and $\mathcal{M}_f$ the Moreau envelope of $f$. The RS prior term splits according to the block structure of F from Assumption C.7, yielding, for $\gamma \leq 1$,

$$\widetilde{\Psi}_w^{(\gamma \leq 1)} = \gamma\,\mathbb{E}_\xi\Big[\mathcal{Z}_{w_\star}^{(1)}\,\log \mathcal{Z}_w^{(1)}\Big] + (1 - \gamma)\,\mathbb{E}_\xi\Big[\log \mathcal{Z}_w^{(2)}\Big]\,, \tag{92}$$

and, for $\gamma > 1$,

$$\widetilde{\Psi}_w^{(\gamma > 1)} = \mathbb{E}_\xi\Big[\mathcal{Z}_{w_\star}^{(3)}\,\log \mathcal{Z}_w^{(3)}\Big]\,, \tag{93}$$

where the partition functions $\mathcal{Z}_{w_\star}(\varrho, \Lambda) = \int P_\star(\mathrm{d}w_\star)\,e^{-\Lambda w_\star^2/2 + \varrho w_\star}$ and $\mathcal{Z}_w(\varrho, \Lambda, \pi) = \int \mathrm{d}w\,e^{-\lambda w^2/2 - \Lambda w^2/2 - \pi|w|^{q^\star} + \varrho w}$ are evaluated at the block-specific arguments

$$
\begin{aligned}
\mathcal{Z}_{w_\star}^{(1)} &= \mathcal{Z}_{w_\star}\big(\tfrac{\hat{m}\,\xi}{\sqrt{\hat{q}\,(1+\gamma)}}, \tfrac{\hat{m}^2}{\hat{q}\,(1+\gamma)}\big), & \mathcal{Z}_w^{(2)} &= \mathcal{Z}_w\big(\sqrt{\hat{q}}\,\xi, \hat{V}, \tfrac{\hat{P}}{2}\big), \\
\mathcal{Z}_w^{(1)} &= \mathcal{Z}_w\big(\sqrt{\hat{q}\,(1 + \tfrac{1}{\gamma})}\,\xi, \hat{V}(1 + \tfrac{1}{\gamma}), \tfrac{\hat{P}}{2}\big), & \mathcal{Z}_{w_\star}^{(3)} &= \mathcal{Z}_{w_\star}\big(\tfrac{\hat{m}\,\xi}{\sqrt{2\hat{q}}}, \tfrac{\hat{m}^2}{2\hat{q}}\big), \\
& & \mathcal{Z}_w^{(3)} &= \mathcal{Z}_w\big(\sqrt{2\hat{q}}\,\xi, 2\hat{V}, \tfrac{\hat{P}}{2}\big).
\end{aligned}
$$

The block superscripts $(1), (2), (3)$ refer respectively to the signal block ($\gamma \leq 1$, dimension $d$), the kernel block ($\gamma \leq 1$, dimension $p - d$), and the single block ($\gamma > 1$, dimension $p$) of F.

## D.4. Saddle-point equations and agreement with Theorem 4.8

Extremizing $\Phi^{(0)} := \lim_{r \to 0^+} r^{-1} \Phi^{(r)}$ in the eight RS variables $(m, V, q, P, \hat{m}, \hat{V}, \hat{q}, \hat{P})$ gives the saddle-point system. The channel block, using the proximal-shifted residual $f_\ell(y, \omega, V, P) := V^{-1}\big(\mathcal{P}_{V\ell(y, \cdot - y\varepsilon_t \, {}^{q^\star}\!\sqrt{P})}(\omega) - \omega\big)$, reads

$$
\hat{m} = \alpha\sqrt{\gamma}\, \mathbb{E}_\xi \int dy \, \partial_\omega \mathcal{Z}_0 \, f_\ell, \quad \hat{q} = \alpha\gamma\, \mathbb{E}_\xi \int dy \, \mathcal{Z}_0 \, f_\ell^2,
$$

$$
\hat{V} = -\alpha\gamma\, \mathbb{E}_\xi \int dy \, \mathcal{Z}_0 \, \partial_\omega f_\ell, \quad \hat{P} = \alpha\gamma\, \varepsilon_t \, q^\star \, P^{q^\star - 1} \mathbb{E}_\xi \int dy \, y \, \mathcal{Z}_0 \, f_\ell. \tag{94}
$$

The prior block, computed from $\partial_{\hat{m}, \hat{q}, \hat{V}, \hat{P}} \widetilde{\Psi}_w$ via the identities $\partial_1 \mathcal{Z}_w = \mathcal{Z}_w f_w$ and $\partial_2 \mathcal{Z}_w = -\frac{1}{2}\big(\partial_\varrho f_w + f_w^2\big)$, with the proximal $f_w(\varrho, \Lambda, \pi) = \arg\min_z \big[\pi|z|^{q^\star} + \lambda z^2 + \frac{\Lambda}{2}z^2 - \varrho z\big]$, reproduces the prior equations Eqs. (25) and (26) with the same case split $\gamma \leq 1$ versus $\gamma > 1$ inherited from the block structure of Eq. (42).

Under the change of variables $(m, \tau, V, P, \bar{m}, \bar{\tau}, \bar{V}, \bar{P}) = \big(m, q, V, P, \hat{m}/\sqrt{\gamma}, \hat{q}/\gamma, \hat{V}, \hat{P}/(\alpha\gamma)\big)$, the channel system Eq. (94) coincides with Eq. (24), and the prior system matches Eqs. (25) and (26). The two derivations — CGMT (rigorous, Appendix C.2) and replica (heuristic, this appendix) — therefore agree, providing an independent statistical-physics check of Theorem 4.8.

# E. Numerical Details

The self-consistent equations from Theorem 4.8 are amenable to fixed-point iteration. Starting from a random initialization, we iterate through both the hat and non-hat variable equations until the maximum absolute difference between the order parameters in two successive iterations falls below a tolerance of $10^{-5}$.

To speed up convergence we use a damping scheme. Denoting by $x_i^{\text{new}}$ the value of an order parameter freshly computed at iteration $i$ and by $x_{i-1}$ its value at the previous iteration, the update reads $x_i := \mu \, x_i^{\text{new}} + (1 - \mu) \, x_{i-1}$, where $\mu \in (0, 1]$ is the damping parameter.

Once convergence is achieved for fixed $\lambda$, the hyper-parameters are optimized via a one-dimensional, gradient-free minimization.

For each iteration we evaluate the proximal operator numerically using SciPy's (Virtanen et al., 2020) Brent algorithm for univariate minimization (`scipy.optimize.minimize_scalar`). The numerical integration is handled with SciPy's adaptive quadrature (`scipy.integrate.quad`). These routines let us evaluate the equations and the integrations to the desired accuracy.

All experiments were run on consumer-grade hardware (Mac Studio M2 Ultra, 2022), and no individual run took more than one day of CPU time.

