# OpenReview forum: "On the Existence of Consistent Adversarial Attacks in High-Dimensional Linear Classification"
_ICML.cc/2026/Conference — ICML 2026 spotlight_

### Official Review · Reviewer_t9Dk · 2026-03-09

**Soundness:** 3
**Presentation:** 3
**Significance:** 2
**Originality:** 3
**Overall Recommendation:** 5
**Confidence:** 3

**Summary:**

This paper studies consistent adversarial attacks in high-dimensional binary classification, i.e. perturbations that flip the model’s prediction while preserving the target rule. The authors introduce corresponding error metrics and derive asymptotic characterizations in two settings: a well-specified linear model and a latent linear / overparameterized model.
The authors find that overparameterization has a dual effect on consistent adversarial robustness, and the direction of this effect depends on the error metric. In particular, overparameterization increases the boundary error, suggesting greater vulnerability among correctly classified examples, while at the same time decreasing the overall consistent robust error.

**Compliance With Llm Reviewing Policy:**

Affirmed.

**Final Justification:**

I appreciate the authors' response. Most of concerns are resolved and I raised my score. I recommend this paper for acceptance.

**Key Questions For Authors:**

- The definition of consistency uses $f_\star(x)=f_\star(x+\delta)$, which is stronger than preserving the ground-truth label in noisy settings. Can the authors better justify this modeling choice and discuss when it is the right notion versus ordinary label invariance?
- What is the main conceptual novelty of Section 4 relative to Section 3: is it primarily the latent/overparameterized asymptotics, or do the authors also claim a new geometric insight beyond the earlier analysis?

**Limitations:**

This paper does not have a separate section for conclusion and discussion of limitations due to page limit. However, for a very compact paper like this, a conclusion section is necessary and highly encouraged. Also, this paper would benefit from a more explicit discussion of the gap between the analyzed high-dimensional linear/latent models and modern nonlinear architectures, the strength of the target-invariance assumption, and the dependence of the results on specific asymptotic and distributional assumptions.

**Strengths And Weaknesses:**

**Strengths**
- The paper addresses an interesting and meaningful distinction between consistent and inconsistent adversarial perturbations, which is often implicit in robustness work but rarely formalized so explicitly.
- The asymptotic analysis is mathematically nontrivial, and the latent-space/overparameterized model adds a useful perspective beyond the simplest well-specified linear setting.
- The results provide clear qualitative insights: vulnerability is tied to the component of the learned predictor orthogonal to the target, and high-dimensional geometry plays a central role under general $\ell_p$ norms.

**Weaknesses**
- The paper has several clarity and modeling issues. In particular, the target-invariance condition
$f_\star(x)=f_\star(x+\delta)$ is stronger than label preservation in noisy settings, and although this is mentioned briefly, the implications of this choice are not discussed enough.
- Some statements in the linear section appear to require stronger assumptions than stated. For example, parts of the geometric reduction seem to rely on more than mere monotonicity of the link function, and the deterministic indicator-link case does not fit some equivalences as cleanly as suggested. This weakens confidence in the exposition even if the main intuition is reasonable.
- Section 4 feels structurally quite similar to Section 3. While the latent model is richer, the geometric argument is largely an adaptation of the earlier section, and the paper could do a better job highlighting what is genuinely new there.
- The limitations discussion is minimal. The gap between the analyzed high-dimensional linear/latent models and modern nonlinear networks is not sufficiently discussed.

---

> ### Author Rebuttal · Authors · 2026-03-30
>
> We thank the reviewer for recognising the meaningful distinction and nontrivial asymptotics.
>
> **Q1 — Target invariance vs label preservation.**
> We agree target invariance is stronger and expanded the discussion in Remark 2.2. The justification is twofold:
>
> (a) For probabilistic classifiers with strictly monotonic $\phi$, target invariance is the condition that the data-generating process $P(y|x)$ is unchanged, which is the natural and most principled notion of consistency because it asks whether the label-generating process changed, not whether a stochastic coin flip happened to land the same way.
>
> (b) For all classifiers we have that the metric we considered could be seen as a lower bound for the label consistent attacks $E_{\mathrm{rob}}^{\mathrm{cns,target}} \leq E_{\mathrm{rob}}^{\mathrm{cns,label}}$. The proof of this is straightforward: the target-invariance constraint $\langle w_\star, \delta \rangle = 0$ defines a hyperplane, while the label-preservation constraint $\mathrm{sign}(\langle w_\star, x+\delta \rangle) = \mathrm{sign}(\langle w_\star, x \rangle)$ defines a half-space that contains this hyperplane.
>
> **Q2 — Conceptual novelty of Section 4.**
> While the geometric argument extends naturally, the key contributions of Section 4 are substantive rather than methodological.
> The contribution coming from the application of the finite $n,d$ analysis establishes that the existence condition for consistent attacks involves $P_\perp F^\top \hat{\theta}$ rather than $\hat{w}_\perp$, revealing how the feature matrix $F$ mediates vulnerability.
> Passing to the high dimensional limit, the key finding of the new Theorem 4.8 and 4.9, formalized in Proposition 4.11, is the opposite effects that overparameterization has on the two metrics.
> This is a new insight that cannot be obtained from the well-specified model, which has no notion of overparameterization.
> We further validated this prediction by replacing the linear estimator with a trained 2-layer ReLU MLP, confirming the same qualitative behavior (see response to Reviewer 4Hgo, W3).
>
> **On assumptions for deterministic link.** For $\phi(t) = \mathbf{1}\\{t \geq 0\\}$, target invariance ($\langle w_\star, \delta \rangle = 0$) is strictly stronger than label invariance ($\mathrm{sign}(\langle w_\star, x+\delta \rangle) = \mathrm{sign}(\langle w_\star, x \rangle)$), but the lower-bound relationship $E_{\mathrm{rob}}^{\mathrm{cns,target}} \leq E_{\mathrm{rob}}^{\mathrm{cns,label}}$ holds. The geometric reduction (Proposition 3.2) requires only monotonicity of the link. The asymptotic analysis (Theorem 3.11) requires Assumption 3.8, where the indicator is a valid special case with deterministic labels.
>
> **Limitations.** We added a dedicated Limitation section, responding to the shared concern across reviewers. We emphasize that the modeling choices enabling exact analysis also connect meaningfully to practice: the random-features model is the standard proxy for overparameterized networks in the proportional asymptotics literature. The Gaussian/proportional setting belongs to a universality class with extensions to broader distributions.
> Target invariance, while strictly stronger than label preservation in all cases, yields conservative (lower-bound) metrics and is the natural notion for probabilistic classifiers.
>
> **Changes made:** Expanded Remark 2.2, revised Section 4 introduction to highlight the contributions, added clarifying remark after Proposition 3.2, new Conclusion and Limitation sections.

---

> > ### Author Rebuttal · Reviewer_t9Dk · 2026-03-31
> >
> > I appreciate the authors' response. This clarifies most of my concerns and I decide to raise the score.

---

### Official Review · Reviewer_dLQB · 2026-03-10

**Soundness:** 4
**Presentation:** 4
**Significance:** 3
**Originality:** 4
**Overall Recommendation:** 5
**Confidence:** 4

**Summary:**

This paper studies the different between consistent and inconsistent adversarial examples from a theoretic perspective.
The authors consider both raw ("well-specified") and latent spaces and present asymptotic robustness metrics.
In addition, they explore the role of over-parameterization and present some counterintuitive results.

**Compliance With Llm Reviewing Policy:**

Affirmed.

**Key Questions For Authors:**

To check my understanding:
In Section 3.1, paragraph 1, you say "the attack must be orthogonal to the covariate", but this seems wrong (both intuitively and algebraically).
To be consistent, the attack should be orthogonal to the model, no?
Is there a typo (and you meant <\delta, w> = 0)?

**Limitations:**

yes

**Strengths And Weaknesses:**

--

Strengths:

- The paper is very well written and easy to follow.

- The paper investigates an under-studied topic (consistent vs inconsistent attacks) in a rigorous and well justified way.

- The paper ties its results and contributions to related work throughout, while still remaining self-contained.

- The paper takes the interesting problem of quantifying what the authors call consistency (of attacks) and phrases it ultimately as a geometric problem about random vectors. (And it solves the problem to boot.)

Weaknesses:

- (Minor presentation thing) Section 3.1 could really use an illustrative figure.

- The authors state (in Section 2) "... target invariance is a natural condition in a statistical framework: it requires adversarial perturbations to preserve the conditional label distribution P(y|x), not individual label samples."
This is very restrictive, and contradicts the mentioned Goodfellow et al. 2015 (panda) example -- it depicts a panda to humans, but for any continuous model the attack likely changes the exact conditional probability.

---

> ### Author Rebuttal · Authors · 2026-03-30
>
> We thank the reviewer for the careful reading, positive assessment, and for rating the originality as excellent. We address the specific points below.
>
> **Figure for Section 3.1.** We added a tikz figure illustrating the geometry of consistent attacks: the constraint hyperplane $\mathcal{H}\_q(\varepsilon) = \\{ \delta \in B\_q(\varepsilon) : \langle w\_\star, \delta \rangle = 0 \\}$, the vulnerable tube $\\{x : \kappa\_q(x) < \varepsilon \\}$ around the decision boundary, and how the orthogonal component $\hat{w}\_\perp$ determines the attack surface.
>
> **Target invariance vs label invariance.** For the probabilistic classifiers we study (logistic, probit with strictly monotonic $\phi$), target invariance is the condition that the data-generating process $P(y|x)$ is unchanged:
> this is the natural notion in a statistical framework, as it identifies whether the perturbation changed the underlying signal rather than whether a particular stochastic label realization happened to be preserved.
> For the noiseless indicator, target invariance ($\langle w\_\star, \delta \rangle = 0$) is strictly stronger than label preservation ($\text{sign}(\langle w\_\star, x + \delta \rangle) = \text{sign}(\langle w\_\star, x \rangle)$), as the reviewer's intuition correctly suggests: points far from the decision boundary have more room under label invariance.
>
> However, since the target-invariant constraint set is always contained in the label-invariant one, our metrics provide a rigorous lower bound on label-preserving vulnerability: $E\_{\mathrm{rob}}^{\mathrm{cns,target}} \leq E\_{\mathrm{rob}}^{\mathrm{cns,label}} \leq E\_{\mathrm{rob}}$. Our results therefore quantify one side of the picture, with $E\_{\mathrm{rob}}$ (already characterized) bounding the other.
>
> Regarding the panda example: the reviewer is correct that a perturbation shifting $P(y|x)$ infinitesimally while preserving the argmax would be label-invariant but not target-invariant under our definition. Extending to approximate target invariance ($|f\_\star(x+\delta) - f\_\star(x)| \leq \eta$) for nonlinear models is a natural future direction that we now discuss in the added Conclusions section.
>
> **Typo in Section 3.1.** You are correct: this was a typo and it has been corrected.
>
> **Changes made:** Added a geometric figure in Section 3.1, expanded Remark 2.2, fixed typo, added discussion in Limitations and Conclusions.

---

> > ### Author Rebuttal · Reviewer_dLQB · 2026-04-03
> >
> > Thanks!

---

### Official Review · Reviewer_4Hgo · 2026-03-11

**Soundness:** 3
**Presentation:** 3
**Significance:** 3
**Originality:** 3
**Overall Recommendation:** 5
**Confidence:** 3

**Summary:**

This paper asks what fundamentally distinguishes an “adversarial attack” from misclassification caused by limited data or model mismatch, focusing on consistent attacks that preserve the ground-truth label. The authors formalize consistency and propose refined robustness metrics, including a consistent robust error and a “boundary”-type metric that counts only errors induced by the attack rather than pre-existing mistakes.

Specifically, the paper studies high-dimensional linear classification in two settings: a well-specified linear model and a latent-variable (random-features) model capturing misspecification/overparameterization. It provides (i) geometric/existence characterizations of consistent perturbations, (ii) exact proportional-limit predictions under Gaussian designs for the proposed metrics and for (robust) ERM, and (iii) empirical validation via simulations and additional experiments (including MNIST/Fashion-MNIST in a teacher–student random-features setup).

**Compliance With Llm Reviewing Policy:**

Affirmed.

**Final Justification:**

After the rebuttal, my main concerns have been adequately addressed. The authors’ clarifications and additional explanations improved the paper’s clarity and strengthened its contributions. Accordingly, I have increased my score and recommend acceptance.

**Key Questions For Authors:**

The main questions are reflected in the weaknesses above, particularly regarding:
- A clearer comparison with closely related theoretical work on adversarial training and robustness.
- Additional intuition for the main theoretical results (e.g., Theorem 3.11 and 4.8).
- Whether the proposed consistent-attack metrics provide useful insights when applied to modern neural network models.

**Limitations:**

Not adequately. I did not find a clear limitations section, and the societal impact statement is extremely brief. Constructive suggestions have been stated in weaknesses

**Strengths And Weaknesses:**

**Strengths**

- The paper is generally well structured and clearly written. It provides clear definitions of “consistency” via target/label invariance and careful separation of standard robust error vs consistent-only notions, which clearly formulate the central question the paper aims to address.
- The paper targets a foundational conceptual ambiguity in adversarial robustness—whether “robust error” conflates true label-preserving vulnerability with distributional/label-changing perturbations—and provides metrics that operationalize this distinction. The exact asymptotic characterizations for consistent-attack metrics and robust ERM in a misspecified/overparameterized latent model offer a principled lens on contradictory empirical observations about robustness vs overparameterization

**Weaknesses**

- Limited comparison with closely related theoretical work. The paper would benefit from a more comprehensive comparison with prior theoretical analyses of adversarial training and robustness, including:
    1. Taheri, Pedarsani, Thrampoulidis, “Asymptotic Behavior of Adversarial Training in Binary Classification”
    2. Javanmard & Soltanolkotabi, “Precise Statistical Analysis of Classification Accuracies for Adversarial Training”
    3. Clarysse, Hörrmann, Yang, “Why adversarial training can hurt robust accuracy” (arXiv:2203.02006).
    4. Tanner, Vilucchio, Loureiro, Krzakala, “A High Dimensional Statistical Model for Adversarial Training: Geometry and Trade-Offs”
    5. Vilucchio, Tsilivis, Loureiro, Kempe, “On the Geometry of Regularization in Adversarial Training: High-Dimensional Asymptotics and Generalization Bounds”

Without a detailed comparison, it is somewhat unclear how the present work differs from or extends these results. It would strengthen the paper if the authors could clarify: how the problem formulation differs from these prior works; whether the analysis techniques used here provide improvements or complementary perspectives; and what new insights or conclusions emerge specifically from this work.

- Limited interpretability of the theoretical results.
Some key results (e.g., Theorem 3.11 and Theorem 4.8) involve complex integral expressions that may be difficult for readers to interpret. Providing additional intuition about the implications of these formulas—such as simplified regimes, qualitative trends, or geometric interpretations—would make the theoretical contributions more accessible.

- Limited connection to modern deep learning practice.
The theoretical analysis focuses primarily on linear or random-feature models with Gaussian assumptions. While this setting is standard for high-dimensional theory, the practical implications for modern deep learning systems remain somewhat indirect. The paper would be strengthened by including at least one experiment involving a small neural network to demonstrate whether the proposed metrics provide useful insights in a more realistic setting.

---

> ### Author Rebuttal · Authors · 2026-03-30
>
> We thank the reviewer for recognising the principled lens of our work and for the detailed suggestions. We address the main points below.
>
> **W1 — Comparison with prior theoretical work.**
> We added a detailed comparison in Appendix A. Key differentiators:
>
> - *Taheri et al. (2023), Javanmard & Soltanolkotabi (2022):* Characterise standard $E_{\mathrm{rob}}$ for well-specified models without distinguishing consistent from inconsistent attacks.
> - *Clarysse et al. (2022):* Studies directed attacks along the signal ($e_1$) with a noiseless model. We consider general $\ell_q$ attacks, noisy labels, and distinguish consistent from inconsistent perturbations.
> - *Tanner et al. (2025):* Characterises $E_{\mathrm{rob}}$ with Mahalanobis-norms for well-specified models. We extend via (i) consistent metrics requiring the geometric analysis of Lemma C.10; (ii) the first exact asymptotics under the latent variable model (Theorem 4.8), entirely absent from their work.
> - *Vilucchio et al. (2024b):* Provides the well-specified asymptotics we build on in Section 3.3. Our contribution: the novel metrics, geometric existence conditions, and the entire latent variable model analysis (Sections 4.1–4.3).
>
> The technical novelty is the geometric reduction to perturbations on $\langle w_\star, \delta \rangle = 0$ (Proposition 3.2), the resulting integral characterization (Proposition 3.11) and all the results for the latent variable model (Section 4). The takeaway is the *dual role of overparameterization*: $E_{\mathrm{rob}}$ and $E_{\mathrm{bnd}}^{\mathrm{cns}}$ increase with overparameterization while $E_{\mathrm{rob}}^{\mathrm{cns}}$ decreases, a novel finding, as none of the cited works consider the latent variable model.
>
> **W2 — Interpretability of Theorems 3.11 and 4.8.** We added remarks on the intuition. The integrals average over the local field $\lambda = \langle \hat{w}, x \rangle / \|\hat{w}\|$ (distance to decision boundary): for each $\lambda$, the integrand evaluates whether a consistent perturbation ($\langle w_\star, \delta \rangle = 0$) can cross the boundary, with $\sqrt{\tau - m^2}$ controlling the available attack surface.
>
> **W3 — Consistent metrics beyond linear models.**
> We replaced the linear estimator with a 2-layer ReLU MLP ($h=256$, Adam, lr=$10^{-3}$, weight decay $10^{-4}$, early stopping), keeping the same latent variable data model (Assumption 4.2). This isolates the model class effect: data, ground truth, and attacks are identical to our theory, but the estimator is nonlinear.
> We set $d=1000$, $n_{\mathrm{train}}=2000$ ($\alpha=2$), and varied the feature dimension $p$ (overparameterization $\psi = p/n$). We evaluated $E_{\mathrm{rob}}$, $E_{\mathrm{rob}}^{\mathrm{cns}}$, and $E_{\mathrm{bnd}}^{\mathrm{cns}}$ on $n_{\mathrm{test}}=5000$ samples using $\ell_2$ PGD ($\varepsilon = 0.05$, 50 steps, step size $\varepsilon/10$, 5 random restarts) with perturbations $\delta \in \mathbb{R}^d$, $\\|\delta\\|\_2 \leq \varepsilon$ applied as $x' = x + F\delta$, matching Eqs. 4.2–4.4. For $E_{\mathrm{rob}}^{\mathrm{cns}}$ and $E_{\mathrm{bnd}}^{\mathrm{cns}}$, we enforce $\langle w_\star, \delta \rangle = 0$ via projection at each PGD step. Results averaged over 10 independent seeds (uncertainty on last digit in parentheses):
>
> | $p$ | $\psi$ | $E_{\mathrm{rob}}$ | $E_{\mathrm{rob}}^{\mathrm{cns}}$ | $E_{\mathrm{bnd}}^{\mathrm{cns}}$ |
> |-----|------|-------|---------|---------|
> | 1000 | 0.5 | 0.799(7) | 0.735(12) | 0.368(10) |
> | 2000 | 1.0 | 0.837(3) | 0.719(9) | 0.412(5) |
> | 4000 | 2.0 | 0.872(5) | 0.700(14) | 0.449(10) |
> | 8000 | 4.0 | 0.884(5) | 0.694(15) | 0.473(10) |
> | 16000 | 8.0 | 0.888(4) | 0.689(10) | 0.472(10) |
>
> As overparameterization increases, $E_{\mathrm{rob}}$ and $E_{\mathrm{bnd}}^{\mathrm{cns}}$ increase while $E_{\mathrm{rob}}^{\mathrm{cns}}$ decreases, matching the qualitative prediction of Proposition 4.11. This confirms that our finding is not an artifact of the linear model class.
>
> **Limitations section.**
> Following shared suggestions, we expanded the Limitations and added a Conclusions section:
>
> - *Linear/random-features scope:* The latent variable model is the standard proxy for overparameterized neural networks, though it captures the NTK rather than the feature learning regime (Hastie et al., 2020). Extending beyond this is important future work.
> - *Target invariance:* As discussed with Reviewers dLQB and t9Dk, target invariance is natural for probabilistic classifiers (preserving $P(y|x)$), and yields a lower bound on label-preserving error for all classifiers.
>
> **Changes made:** Expanded Appendix A, added remarks after Theorems 3.11 and 4.8, new Conclusions and Limitations sections.

---

> > ### Author Rebuttal · Reviewer_4Hgo · 2026-04-03
> >
> > I thank the authors for their responses. However, the reported modifications to the manuscript are not visible on my side, likely because ICML policy does not allow changes to the original submission during the rebuttal phase. Could you please address the concerns directly in a more detailed written response?

---

> > > ### Author Response · Authors · 2026-04-04
> > >
> > > We thank the reviewer for the follow-up. Since ICML policy does not allow manuscript updates during rebuttal, we provide below the substance of the changes, quoting added passages as space permits.
> > >
> > > **W1 — Comparison with prior work.**
> > > In Appendix A (line 738) we added:
> > > A common thread in most of the previous analyses is the characterization of the standard
> > > robust error $E\_{rob}$, treating all perturbations in $B\_{q}(\varepsilon)$ uniformly without distinguishing whether they preserve or alter the ground-truth classification. For instance, Taheri et al. (2023) and Javanmard & Soltanolkotabi (2022) derive sharp asymptotics for $E\_{rob}$ under adversarial training in well-specified linear models, while Clarysse et al. (2022) study adversarial perturbations directed along the signal in a noiseless setting, and Tanner et al. (2025) extend the analysis to Mahalanobis-norm attacks.
> > > Our work departs from this line by considering the consistency constraint, which requires a different geometric analysis depending on the predictor component orthogonal
> > > to the target (Proposition 3.2 and Lemma C.10).
> > > Of particular relevance is Vilucchio et al. (2024b), which derives high-dimensional asymptotics for binary classification in the well-specified model and which we build upon in Section 3. Our contributions beyond their work are the geometric existence conditions (Proposition 3.2), and the entire latent variable model analysis (Section 4), generalizing previous results (Mei & Montanari, 2022; Goldt et al., 2022a).
> > >
> > > **W2 — Intuition for Theorems 3.11 and 4.9.**
> > > We added a remark after Theorem 3.11 (line 258) (analogous for Theorem 4.9): "The integrals in Eqs. (16)-(17) admit an interpreatation through the local fields framework (Clarté et al., 2023). The variables $z\_{1} = \langle w^\star, x \rangle / \\sqrt{d}$ and $z\_{2} = \langle \hat{w}, x \rangle / \\sqrt{d}$ are the target and model local fields; the signed distance of a sample to the ground-truth and learned decision boundary, respectively. These two correlated random variables fully determine the error metrics. The first indicator function in both integrals evaluates whether the post-attack model output $z_2 - \tilde{\varepsilon}\mathcal{A}$ falls on the opposite side from the ground truth boundary. The quantity $\mathcal{A}$, proportional to $\sqrt{\tau - m^2} = \\|\hat{w}_{\perp}\\|\_{2}$, controls the attack surface within $H\_{q}(\varepsilon)$, since only the orthogonal component of the predictor can be exploited (cf. Proposition 3.2). The difference between $E\_{rob}^{cns}$ and $E\_{bnd}^{cns}$ is the additional indicator that restricts to samples where model and target agree before the attack, i.e. correctly classified points."
> > >
> > > **W3 — Neural network experiment.**
> > > The goal of this experiment is to test whether the dual role of overparameterization predicted by our theory (Proposition 4.11) persists when the linear estimator is replaced by a nonlinear model, isolating the effect of the model class while keeping the data-generating process identical (Assumption 4.6). We added a full description in Appendix B.2. The estimator is a 2-layer ReLU MLP (hidden dim 256), trained with BCE loss, Adam (lr=1e-3, weight decay 1e-4), and early stopping (patience 50 on a 20% validation split). We set $d=1000$ and vary $p$ from $1000$ to $16000$. Attacks are $\ell_2$-PGD in the latent space (50 steps, 5 restarts, $\varepsilon=0.05$) with $\langle w^*, \delta \rangle=0$ enforced by projection at each step. Results averaged over 10 seeds. Figure and code [here](https://anonymous.4open.science/r/rebuttal-3080).
> > >
> > > **Conclusions and Limitations Section.**
> > > We added before the Impact Statement: "We investigated the distinction between consistent and inconsistent adversarial attacks in high-dimensional binary classification, introducing consistent robust error and boundary error as refined metrics. Curiously, we find that overparameterization has a dual effect on consistent robustness: the boundary error $E_{bnd}^{cns}$ increases, indicating heightened vulnerability for correctly classified examples, while the overall consistent robust error $E_{rob}^{cns}$ decreases.
> > > This stems from the beneficial role of overparameterization in improving clean generalization, which compensates for increased boundary vulnerability, providing a possible explanation for the contradictory empirical observations reported in (Chen et al., 2024).
> > >
> > > Several limitations emerge from our study.
> > > Our analysis is set in the proportional high-dimensional regime with Gaussian covariates. The latent variable model captures the NTK/random-features regime but not feature learning; extending beyond this is an important future direction. Target invariance is strictly stronger than label preservation; our metrics thus provide conservative lower bounds on label-preserving vulnerability. Extending to approximate target invariance ($|f^\star(x+\delta) - f^\star(x)| < \eta$) is a natural next step.

---

### Official Review · Reviewer_RxqU · 2026-03-13

**Soundness:** 3
**Presentation:** 3
**Significance:** 3
**Originality:** 3
**Overall Recommendation:** 4
**Confidence:** 2

**Summary:**

This paper investigates a fundamental distinction in adversarial robustness: consistent adversarial attacks, which perturb inputs while preserving the ground-truth label, versus inconsistent attacks, which change the true classification. The authors introduce novel error metrics — the consistent robust error and consistent boundary error — to precisely quantify a model's vulnerability to label-preserving perturbations, and derive rigorous, closed-form asymptotic characterizations of these metrics in the high-dimensional proportional limit (where data dimension, sample size, and feature dimension all scale to infinity at fixed ratios).

**Compliance With Llm Reviewing Policy:**

Affirmed.

**Key Questions For Authors:**

Address the concern above.

**Limitations:**

yes

**Strengths And Weaknesses:**

Overall, the paper is interesting, despite its narrow scope (linear classifier, Gaussian covariate assumptions and proportional asymptotics). My main concern is why better training won't eliminate the consistent attacks? In other words, it seems that there are two sources of consistent attacks:
- Algorithmic failure: The training procedure (e.g., standard ERM without robust training) learns a suboptimal linear classifier.
- Statistical/geometric inevitability: Even the Bayes-optimal classifier trained on finite data in high dimensions will learn a suboptimal classifier.

While the paper addresses the first source of sub-optimality, it never cleanly establishes that no finite-sample estimator, however trained, can fully eliminate consistent attacks in the high-dimensional proportional regime (i.e., there exists no training algorithm that could ever find the optimal/ground truth classifier). The paper would be significantly strengthened by a dedicated discussion establishing that consistent attacks are an irreducible statistical phenomenon in high dimensions — not merely a symptom of poor training.

---

> ### Author Rebuttal · Authors · 2026-03-30
>
> We thank the reviewer for finding the paper interesting and for pointing out the two sources of consistent attacks. This distinction is valuable and we have added a discussion formalizing it.
>
> **On whether better training can eliminate consistent attacks:**
>
> In our framework, consistent attacks exist if and only if the predictor has a nonzero component orthogonal to the target: $\tau - m^2 > 0$. This quantity is computable from our self-consistent equations for any choice of loss $\ell$, regularization $\lambda$, and training attack budget $r$. One can optimize over all these hyperparameters to find the best achievable robust ERM estimator and verify that $\tau - m^2 > 0$ for all finite $\alpha = n/d$.
>
> A natural, interesting, follow-up is whether estimators outside the ERM class (e.g. the Bayes-optimal estimator) could achieve perfect alignment $\tau = m^2$. For the continuous prior on $w_\star \in \mathbb{S}^{d-1}(\sqrt{d})$ assumed in our paper, the answer is no: the Bayes-optimal generalization error for the perceptron with continuous weights decays as $\sim 1/\alpha$ but remains strictly positive at any finite $\alpha$. Indeed, even the Bayes-optimal estimator at finite sample complexity $\alpha$ can be characterised as a noisy version of the target $\hat{\theta}\_{BO}$, see e.g. (Aubin et al., 2020).
> This is in contrast with discrete or sparse priors (Barbier et al., 2019) where a sharp phase transition can yield exact recovery at finite $\alpha$, and therefore vanishing consistent attacks above a critical sample complexity.
> Under our continuous prior weight assumption, $\tau^{\star}\_{\mathrm{Bayes}} - (m^{\star}\_{\mathrm{Bayes}})^2 > 0$ for all finite $\alpha$, and by Corollary 3.5 the consistent attack probability is strictly positive even for the best possible estimator.
>
> After the reviewer suggestion, we find that this distinction between the *algorithmic gap* (the gap between the best ERM estimator and the Bayes-optimal) and the *statistical gap* (the gap between the Bayes-optimal and perfect recovery) is useful and pertinent to discussion: the former can be reduced by better training, while the latter is an inherent statistical limitation of the problem. We will add a remark in the revised version of the paper formalizing this argument and distinction.
>
> **Changes made:** Added Remark formalizing the irreducibility argument and the algorithmic/statistical gap distinction.

---

> > ### Author Rebuttal · Reviewer_RxqU · 2026-04-01
> >
> > The authors addressed my concerns.

---

### Decision · Program_Chairs · 2026-04-30

**Decision:**

Accept (spotlight)

**Comment:**

The paper offers an interesting perspective, distinguishing between consistent and inconsistent attacks. The main theoretical results are non-trivial, closed-form asymptotic characterizations of the inconsistency metrics within both well-specified linear classifiers and latent-variable overparameterized models. The reviewers also agree that the paper is well written, easy to follow, and that the perspective and results are novel. Thus, I recommend acceptance.